# INDEPENDENTLY-NORMALIZED SGD FOR GENERALIZED-SMOOTH NONCONVEX OPTIMIZATION

## ABSTRACT

Recent studies have shown that many nonconvex machine learning problems meet a so-called generalized-smooth condition that extends beyond traditional smooth nonconvex optimization. However, the existing algorithms designed for generalized-smooth nonconvex optimization encounter significant limitations in both their design and convergence analysis. In this work, we first study deterministic generalized-smooth nonconvex optimization and analyze the convergence of normalized gradient descent under the generalized Polyak-Łojasiewicz condition. Our results provide a comprehensive understanding of the interplay between gradient normalization and function geometry. Then, for stochastic generalized-smooth nonconvex optimization, we propose an independently-normalized stochastic gradient descent algorithm, which leverages independent sampling, gradient normalization and clipping to achieve an $\mathcal{O}(\epsilon^{-4})$ sample complexity under relaxed assumptions. Experiments demonstrate the fast convergence of our algorithm.

## 1 INTRODUCTION

In modern machine learning, the convergence of gradient-based optimization algorithms has been well studied in the standard smooth nonconvex setting. However, it has been shown recently that smoothness fails to characterize the global geometry of many nonconvex machine learning problems, including distributionally-robust optimization (DRO)(Levy et al., 2020; Jin et al., 2021), meta-learning (Nichol et al., 2018; Chayti & Jaggi, 2024) and language models (Liu et al., 2023; Zhang et al., 2019). Instead, these problems have been shown to satisfy a so-called *generalized-smooth* condition, in which the smoothness parameter can scale with the gradient norm in the optimization process (Zhang et al., 2019).

In the existing literature, various works have proposed different algorithms for solving generalized-smooth nonconvex optimization problems. Specifically, one line of work focuses on the classic stochastic gradient descent (SGD) algorithm (Li et al., 2024; Reisizadeh et al., 2023). However, the convergence of SGD either relies on adopting very large batch size or involves large constants, and the practical performance of SGD is often poor due to the ill-conditioned smoothness parameter when gradient is large. Another line of work focuses on clipped SGD, which adapts to the generalized-smooth geometry by leveraging gradient clipping and normalization (Zhang et al., 2019; 2020). However, to establish convergence guarantee, these studies rely on the strong assumption that the stochastic approximation error is bounded almost surely.

Motivated by the algorithmic and theoretical limitations discussed above, this work aims to explore the interplay between algorithm design and the geometry of generalized-smooth functions, and develop algorithms tailored for generalized-smooth nonconvex optimization. To achieve this overarching goal, we need to address several fundamental challenges. First, even in deterministic generalized-smooth nonconvex op-

timization, there is limited knowledge about how to adapt gradient-based algorithms to the geometry of generalized-smooth problems. Thus, we want to answer the following question.

- *Q1: In deterministic nonconvex optimization, how can we adapt algorithm hyperparameters to align with the Polyak-Łojasiewicz geometry of generalized-smooth problems? What are the convergence rates?*

Second, in stochastic generalized-smooth nonconvex optimization, the existing SGD-type algorithms are either impractical due to poor performance or relying on strong assumptions to establish convergence guarantee. Therefore, we aim to answer the following question.

- *Q2: Can we develop a novel algorithm tailored for stochastic generalized-smooth optimization that achieves fast convergence in practice while providing convergence guarantee under relaxed assumptions?*

In this work, we provide comprehensive answers to the above questions and develop new algorithms as well as convergence analysis in generalized-smooth nonconvex optimization. In light of the above discussions, we summarize our key contributions as following.

## 1.1 OUR CONTRIBUTION

We first consider deterministic generalized-smooth nonconvex optimization, and study the convergence of normalized gradient descent under the generalized Polyak-Łojasiewicz (PŁ) condition. Our result characterizes the algorithm convergence rate under a broad spectrum of function geometry characterized by the generalized-smooth and PŁ conditions, and provides deep insights into adapting algorithm hyper-parameters (such as learning rate and gradient normalization scale) to the underlying function geometry.

We then consider stochastic generalized-smooth nonconvex optimization, for which we propose a novel Independently-Normalized Stochastic Gradient Descent (I-NSGD) algorithm. Specifically, I-NSGD leverages normalized gradient updates with independent sampling and gradient clipping to reduce the bias and enhance stability. Consequently, we can establish convergence of I-NSGD with $\mathcal{O}(\epsilon^{-4})$ sample complexity under a relaxed assumption on the approximation error of stochastic gradient and constant-level batch size. This makes the algorithm well-suited for solving large-scale problems. We further study the convergence behavior of I-NSGD under the generalized PŁ condition.

We compare the numerical performance of our I-NSGD algorithm with other state-of-the-art stochastic algorithms in applications of nonconvex phase retrieval and nonconvex distributionally-robust optimization, both of which are generalized-smooth nonconvex problems. Our results demonstrate the efficiency of I-NSGD in solving generalized-smooth nonconvex problems and match our theoretical guidance.

## 1.2 RELATED WORK

**Generalized-Smoothness.** The concept of generalized-smoothness was introduced by Zhang et al. (2019) with the $(L_0, L_1)$-smooth condition, which allows a function to either have an affine-bounded hessian norm or be locally $L$-smooth within a specific region. This idea was extended by Chen et al. (2023), who proposed the $\mathcal{L}^*_{asym}(\alpha)$ and $\mathcal{L}^*_{sym}(\alpha)$ conditions, controlling gradient changes globally with both a constant term and a gradient-dependent term associated with power $\alpha$, thus applying more broadly. Later, Li et al. (2024) introduced $\ell$-smoothness, which use a non-decreasing sub-quadratic polynomial to control gradient differences. Also, Mishkin et al. (2024) proposed directional smoothness, which preserves $L$-smoothness along specific directions.

**Algorithms for Generalized-Smooth Optimization.** Motivated by achieving comparable lower bounds presented in Arjevani et al. (2023) under standard assumptions, algorithms for solving generalized-smooth problems can be categorized into two main series. The first series focus on adaptive methods. In deterministic non-convex settings, Zhang et al. (2019; 2020) showed that Clipped GD can achieve a rate of

$\mathcal{O}(\epsilon^{-2})$ under mild assumptions.Later, Chen et al. (2023) proposed $\beta$-GD, also achieving $\mathcal{O}(\epsilon^{-2})$ iteration complexity. Later in this year, Vankov et al. (2024b) studies clip and normalized gradient descent under mild-conditions, where they retrieve standard convergence rate in each separate cases under deterministic cases. Gorbunov et al. (2024) varies the learning rate to study smoothed gradient clipping, gradient descent with Polyak step-sizes, triangles Method under convex $(L_0, L_1)$-smooth conditions, where they also achieve standard convergence rate under convex case. In stochastic settings, when the approximation error of the stochastic gradient estimator is bounded, Zhang et al. (2019; 2020) proved clipped SGD achieves $\mathcal{O}(\epsilon^{-4})$ sample complexity. Inspired superior performance of Adagrad Duchi et al. (2011b), Wang et al. (2023); Faw et al. (2023); Hong & Lin (2024) further studied AdaGrad under generalized smooth and relaxed variance assumption with different learning rate schemes. They all attains $\tilde{\mathcal{O}}(1/\sqrt{T})$ convergence rate under mild conditions.Xie et al. (2024a) studied trust-region methods convergence under generalized-smoothness. The second series focus on SGD methods with constant learning rate. Reisizadeh et al. (2023); Li et al. (2024) proved that SGD converges with sample complexity $\mathcal{O}(\epsilon^{-4})$ under generalized-smoothness. To ensure convergence, Reisizadeh et al. (2023) adopted a large batch size of $\mathcal{O}(\epsilon^{-2})$, while Li et al. (2024) relaxed this requirement but introduces additional variables of size $\mathcal{O}(\epsilon^{-1})$. Additionally, various acceleration methods have been explored under the generalized-smoothness condition. Zhang et al. (2020) proposed a general clipping framework with momentum updates; Jin et al. (2021) studied normalized SGD with momentum Cutkosky & Mehta (2020) under parameter-dependent achieves $\mathcal{O}(\epsilon^{-4})$ sample complexity; Hübler et al. (2024) studied normalized SGD with momentum Cutkosky & Mehta (2020) associated with parameter-agnostic learning rates, which establishes $\tilde{\mathcal{O}}(\epsilon^{-4})$ convergence rate and corresponding lower bound. By adjusting batch size, Chen et al. (2023); Reisizadeh et al. (2023) demonstrated that the SPIDER algorithm (Fang et al., 2018) can reach the optimal $\mathcal{O}(\epsilon^{-3})$ sample complexity. Furthermore, Zhang et al. (2024b); Wang et al. (2024a;b); Li et al. (2023) explored the convergence of RMSprop (Hinton et al., 2012) and Adam (Kingma, 2014) under generalized-smoothness. Jiang et al. (2024) studied variance-reduced sign-SGD convergence under generalized-smoothness.

**Machine Learning Applications.** Generalized smoothness has been studied under various machine learning framework. Levy et al. (2020); Jin et al. (2021) studied the dual formulation of regularized DRO problems, where the loss function objective satisfies generalized smoothness. Chayti & Jaggi (2024) identified their meta-learning objective's smoothness constant increases with the norm of the meta-gradient. Gong et al. (2024b); Hao et al. (2024); Gong et al. (2024a); Liu et al. (2022b) explored algorithms for bi-level optimization and federated learning within the context of generalized smoothness. Zhang et al. (2024a) developed algorithms for multi-task learning problem where the objective is generalized smooth. Xie et al. (2024b) studied online mirror descent when the objective is generalized smooth. Xian et al. (2024) studied min-max optimization algorithms' convergence behavior under generalized smooth condition. There is a concurrent work (Vankov et al., 2024a) using independent sampling with Clip-SGD framework to solve variation inequality problem(SVI). Based on this idea, they also propose stochastic Korpelevich method for clip-SGD. Under generalized smooth condition, they proved almost-sure convergence in terms of distance to solution set tailored for solving stochastic SVI problems.

## 2 DETERMINISTIC GENERALIZED-SMOOTH NONCONVEX OPTIMIZATION

We first introduce generalized-smooth optimization problems. Then, we review the classic normalized gradient descent algorithm and study its complexity in the generalized-smooth and gradient-dominant setting.

We are interested in the following nonconvex optimization problem.

$$\min_{w \in \mathbf{R}^d} f(w), \tag{1}$$

where $f : \mathbf{R}^d \to \mathbf{R}$ denotes a nonconvex and differentiable function, and $w$ corresponds to the model parameters. We assume that function $f$ satisfies the following generalized-smooth condition introduced in (Jin et al., 2021; Chen et al., 2023)[1].

**Assumption 1 (Generalized-smooth)** *The objective function $f$ satisfies the following conditions.*

1. *$f$ is differentiable and bounded below, i.e., $f^* := \inf_{x \in \mathbf{R}^d} f(x) > -\infty$;*

2. *There exists constants $L_0, L_1 > 0$ and $\alpha \in [0, 1]$ such that for any $w, w' \in \mathbf{R}^d$, it holds that*

$$\left\| \nabla f(w) - \nabla f(w') \right\| \le \left( L_0 + L_1 \left\| \nabla f(w') \right\|^{\alpha} \right) \left\| w - w' \right\|. \tag{2}$$

The generalized-smooth condition in Assumption 1 is a generalization of the standard smooth condition, which corresponds to the special case of $L_1 = 0$. To elaborate, it allows the smoothness parameter to scale with the gradient norm polynomially, and therefore is able to model functions with highly irregular nonconvex geometry. In (Chen et al., 2023; Zhang et al., 2019), it has been shown that many complex machine learning problems belong to this function class with different parameter $\alpha$, including distributionally-robust optimization ($\alpha = 1$), deep neural networks and phase retrieval ($\alpha = \frac{2}{3}$), etc. Following a standard proof, it is easy to show that generalized-smooth functions satisfy the following descent lemma.

**Lemma 1** *Under Assumption 1, function $f$ satisfies, for any $w, w' \in \mathbf{R}^d$,*

$$f(w) \le f(w') + \langle \nabla f(w'), w - w' \rangle + \frac{1}{2}(L_0 + L_1 \left\| \nabla f(w') \right\|^{\alpha}) \left\| w - w' \right\|^2. \tag{3}$$

The main challenge of generalized-smooth optimization is to control the polynomial gradient norm term $\|\nabla f(w')\|^{\alpha}$ involved in the smoothness parameter. This key observation has motivated the existing studies to develop normalized gradient methods for solving generalized-smooth problems.

Chen et al. (2023) proposed a specialized normalized gradient descent (NGD) algorithm for generalized-smooth nonconvex optimization. The algorithm normalizes the gradient by its norm polynomially, i.e.,

$$\text{(NGD)} \quad w_{t+1} = w_t - \gamma \frac{\nabla f(w_t)}{\|\nabla f(w_t)\|^{\beta}}, \tag{4}$$

where $\gamma > 0$ denotes the learning rate and $\beta$ is a scaling parameter that controls the normalization scale of the gradient norm. Intuitively, when the gradient norm is large, a smaller $\beta$ would make the normalized gradient update more aggressive. Chen et al. (2023) has empirically demonstrated the effectiveness of equation 4 than gradient descent in deterministic settings and proved that, when choosing a proper $\gamma$ and setting $\beta \in [\alpha, 1]$, NGD can find an $\epsilon$-stationary point within $\mathcal{O}(\epsilon^{-2})$ number of iterations.

Moreover, from Liu et al. (2022a); Scaman et al. (2022), it has been observed that generalization of Polyak-Łojasiewicz (PŁ) condition, such as Polyak-Łojasiewicz*(PŁ)*, Kurdyka-Łojasiewicz*(KŁ*), and Separable-Łojasiewicz*(SŁ*) hold in the landscape under over-parametrized neural networks of several state-of-art losses, such as mean-squared loss and cross-entropy loss. Based on scalability of PŁ condition that can be extended to hold in deep learning, and the linear convergence rate studied in Karimi et al. (2016b). In this work, we study the convergence rate of NGD for solving generalized-smooth problems under the following generalized Polyak-Łojasiewicz (PŁ) condition (Karimi et al., 2016a).

**Assumption 2 (Generalized Polyak-Łojasiewicz Condition)** *There exists constants $\mu > 0$ and $\rho > 0$ such that $f(\cdot)$ satisfies, for all $w \in \mathbf{R}^d$,*

$$\frac{1}{2\mu} \left\| \nabla f(w) \right\|^{\rho} \ge f(w) - f^*. \tag{5}$$

---

[1]Jin et al. (2021) considered the special case $\alpha = 1$, and Chen et al. (2023) defined a symmetric version of equation 2.

Assumption 2 generalizes the standard PŁ condition (corresponds to $\rho = 2$) via flexible choice of the parameter $\rho$. In particular, some generalized-smooth functions satisfy the above generalized PŁ condition with different parameters. For example, the sigmoid-like function $f(w) = \frac{1}{2}w^2(\exp(w^2) - 1) + \frac{1}{2}w^2$ satisfies $\rho = 1, \mu = 0.1$, and the polynomial function $f(w) = w + \frac{1}{2}w^2 + w^4$ satisfies $\rho = 3, \mu = 0.1$.

We obtain the following convergence rate result of NGD, where we denote $\Delta_t := f(w_t) - f^*$.

**Theorem 1 (Convergence of NGD)** *Let Assumptions 1 and 2 hold. Choose learning rate $\gamma = \frac{(2\mu\epsilon)^{\beta/\rho}}{8(L_0+L_1)+1}$ where $\epsilon$ denotes the target accuracy, and set $\alpha \leq \beta \leq 1$. Then, the following statements hold.*

- *If $0 < \rho < 2 - \beta$, then we have*

$$\Delta_t = \mathcal{O}\Big(\big(\frac{\rho}{(2-\beta-\rho)\gamma t}\big)^{\frac{\rho}{2-\rho-\beta}}\Big). \tag{6}$$

  *Furthermore, in order to achieve $\Delta_t \leq \epsilon$, the total number of iteration satisfies $T = \mathcal{O}\big((\frac{1}{\epsilon})^{\frac{\beta}{\rho}}\big)$ if $2 - 2\beta < \rho < 2 - \beta$, and $T = \mathcal{O}\big((\frac{1}{\epsilon})^{\frac{2-\rho-\beta}{\rho}}\big)$ if $0 < \rho \leq 2 - 2\beta$.*

- *If $\rho = 2 - \beta$ and choose $\epsilon$ such that $\gamma < \frac{2}{\mu}$, then we have*

$$\Delta_t = \mathcal{O}\Big(\big(1 - \frac{\gamma\mu}{2}\big)^t\Big). \tag{7}$$

  *In order to achieve $\Delta_t \leq \epsilon$, the total number of iteration satisfies $T = \mathcal{O}\big((\frac{1}{\epsilon})^{\frac{\beta}{\rho}} \log \frac{1}{\epsilon}\big)$.*

- *If $\rho > 2 - \beta$, then there exists $T_0 \in \mathbf{N}$ such that for all $t \geq T_0$, we have*

$$\Delta_t = \mathcal{O}\Big(\big(\frac{\Delta_{T_0}}{\gamma^{\frac{\rho}{\rho+\beta-2}}}\big)^{\frac{\rho}{2-\beta}t - T_0}\Big). \tag{8}$$

  *In order to achieve $\Delta_t \leq \epsilon$, the total number of iterations after $T_0$ satisfies $T = \Omega\big(\log((\frac{1}{\epsilon})^{\frac{\beta}{\rho+\beta-2}})\big)$.*

Theorem 1 indicates that the convergence behavior of NGD depends on the parameter $\rho$ in the generalized PŁ condition. When $\rho \leq 2 - \beta$, the algorithm achieves slow sub-linear convergence rate. When $\rho > 2 - \beta$, the algorithm achieves local linear convergence rate. These results match the intuition behind the generalized PŁ condition. Namely, a large $\rho$ indicates that the gradient norm vanishes slowly when the function value gap approaches zero, corresponding to sharp geometry that leads to fast local convergence.

## 3 STOCHASTIC GENERALIZED-SMOOTH NONCONVEX OPTIMIZATION

In this section, we study the following stochastic generalized-smooth optimization problem, where $f_\xi$ corresponds to the loss function associated with data sample $\xi$, and the expected loss function $F(\cdot)$ satisfies the generalized-smooth condition in Assumption 1.

$$\min_{w \in \mathbf{R}^d} F(w) := \mathbb{E}_{\xi \sim \mathbb{P}}\big[f_\xi(w)\big]. \tag{9}$$

### 3.1 NORMALIZED SGD AND ITS LIMITATIONS

To solve the stochastic generalized-smooth problem in equation 9, one straightforward approach is to replace the full batch gradient in the NGD update in equation 4 with the stochastic gradient $\nabla f_\xi(w_t)$, resulting in the following normalized SGD (NSGD) algorithm.

$$(\text{NSGD}) \quad w_{t+1} = w_t - \gamma \frac{\nabla f_{\xi_t}(w_t)}{\|\nabla f_{\xi_t}(w_t)\|^\beta}. \tag{10}$$

NSGD-type algorithms have attracted a lot of attention recently for solving stochastic generalized-smooth problems (Zhang et al., 2019; 2020; Liu et al., 2022b). In particular, it has been proven in these works that NSGD with proper gradient clipping can achieve a sample complexity of $\mathcal{O}(\epsilon^{-4})$, which matches that of the standard SGD algorithm for solving stochastic smooth problems (Ghadimi & Lan, 2013). However, NSGD-type update has the following limitations.

1. *Biased gradient estimator:* The normalized stochastic gradient used in equation 10 is biased, i.e., $\mathbb{E}\left[\frac{\nabla f_{\xi_t}(w_t)}{\|\nabla f_{\xi_t}(w_t)\|^\beta}\right] \neq \frac{\nabla F(w_t)}{\|\nabla F(w_t)\|^\beta}$. This is due to the dependence between $\nabla f_{\xi_t}(w_t)$ and $\|\nabla f_{\xi_t}(w_t)\|^\beta$. In particular, the bias can be huge if the stochastic gradients are diverse, as illustrated in Figure 1.

2. *Strong assumption:* To control the estimation bias and establish theoretical convergence guarantee for NSGD-type algorithms in generalized-smooth nonconvex optimization, the existing studies need to adopt strong assumption. For example, Zhang et al. (2019; 2020) and Liu et al. (2022b) assume that the stochastic approximation error $\|\nabla f_\xi(w) - \nabla F(w)\|$ is bounded by a constant almost surely. In real applications, this constant can be a large numerical number if certain sample $\xi$ is an outlier.

### 3.2 INDEPENDENTLY-NORMALIZED SGD

To overcome the aforementioned limitations, we propose the independently-normalized stochastic gradient (I-NSG) estimator

$$\text{(I-NSG estimator)} \quad \frac{\nabla f_\xi(w)}{\|\nabla f_{\xi'}(w)\|^\beta}, \qquad (11)$$

where $\xi$ and $\xi'$ are samples draw *independently* from the underlying data distribution. Intuitively, the independence between $\xi$ and $\xi'$ decorrelates the denominator from the numerator, making it an unbiased stochastic gradient estimator (up to a scaling factor). Specifically, we formally have that

$$\mathbb{E}_{\xi,\xi'}\left[\frac{\nabla f_\xi(w)}{\|\nabla f_{\xi'}(w)\|^\beta}\right] = \mathbb{E}_{\xi'}\left[\frac{\mathbb{E}_\xi\left[\nabla f_\xi(w)\right]}{\|\nabla f_{\xi'}(w)\|^\beta}\right] \propto \nabla F(w). \quad (12)$$

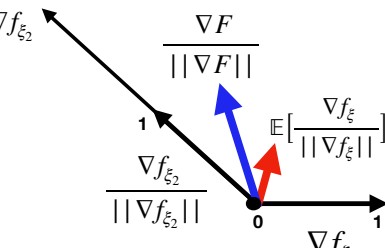

Figure 1: Comparison between normalized full gradient (blue) and expected normalized stochastic gradient (red). Here, $\xi_1$ and $\xi_2$ are sampled uniformly at random.

Moreover, as we show later under mild assumptions, the scaling factor $\mathbb{E}\left[\|\nabla f_{\xi'}(w)\|^{-\beta}\right]$ can be roughly bounded by the full gradient norm and hence resembling the full-batch NGD update. Based on this idea, we formally propose the following independently-normalized SGD (I-NSGD) algorithm, where $\nabla f_{\xi_B}(w_t)$ corresponds to the mini-batch stochastic gradient associated with a batch of samples $B$, and $B'$ denotes another independent batch.

$$\text{(I-NSGD):} \quad w_{t+1} = w_t - \gamma\frac{\nabla f_{\xi_B}(w_t)}{h_t^\beta}, \quad \text{where } h_t = \max\left\{1, (4L_1\gamma)^{\frac{1}{\beta}}\left(2\|\nabla f_{\xi_{B'}}(w_t)\| + \delta\right)\right\}. \quad (13)$$

The above I-NSGD algorithm adopts a clipping strategy for the normalization term $h_t$. This is to impose a constant lower bound on $h_t$, which helps develop the theoretical convergence analysis and avoid numerical instability in practice. We note that I-NSGD requires querying two batches of samples in every iteration. However, as we show in the experiments later, the batch size $|B'|$ can be chosen far smaller than $|B|$.

### 3.3 CONVERGENCE ANALYSIS OF I-NSGD

We adopt the following standard assumptions on the stochastic gradient.

**Assumption 3 (Unbiased stochastic gradient)** *The stochastic gradient $\nabla f_\xi(w)$ is unbiased, i.e.,* $\mathbb{E}_{\xi\sim\mathbb{P}}\left[\nabla f_\xi(w)\right] = \nabla F(w)$ *for all $w \in \mathbf{R}^d$.*

**Assumption 4 (Approximation error )** *There exists $\tau_1, \tau_2 > 0$ such that for any $w \in \mathbf{R}^d$, one has*

$$\left\|\nabla f_\xi(w) - \nabla F(w)\right\| \leq \tau_1\left\|\nabla F(w)\right\| + \tau_2 \ \ a.s. \ \ \forall \xi \sim \mathbb{P}. \tag{14}$$

We note that the above Assumption 4 is much weaker than the bounded approximation error assumption (i.e., $\tau_1 = 0$) adopted in (Zhang et al., 2019; 2020; Liu et al., 2022b). Specifically, it allows the approximation error to scale with the full gradient norm and only assumes bounded error at the stationary points. With these assumptions, we can lower bound the stochastic gradient norm with the full gradient norm as follows.

**Lemma 2** *Let Assumptions 3 and 4 hold. Consider the mini-batch stochastic gradient $\nabla f_{\xi_B}$ with batch size $B = 16\tau_1^2$, then for all $w \in \mathbf{R}^d$ we have*

$$\left\|\nabla f_{\xi_B}(w)\right\| \geq \frac{1}{2}\left\|\nabla F(w)\right\| - \frac{\tau_2}{2\tau_1}. \tag{15}$$

Lemma 2 shows that with a constant-level batch size, the stochastic gradient norm can be lower bounded the full gradient norm up to a constant. This result is very useful in our convergence analysis to effectively bound the mini-batch stochastic gradient norm used in the normalized stochastic gradient update.

We obtain the following convergence result of I-NSGD.

**Theorem 2 (Convergence of I-NSGD)** *Let Assumptions 1, 3 and 4 hold. For the I-NSGD algorithm, choose learning rate $\gamma = \min\{\frac{1}{4L_0}, \frac{1}{4L_1}, \frac{1}{\sqrt{T}}, \frac{1}{8L_1(3\tau_2/\tau_1)^\beta}\}$, batch sizes $B = 2\tau_1^2$, $B' = 16\tau_1^2$ and $\delta = \frac{\tau_2}{\tau_1}$. Denote $\Lambda := F(w_0) - F^* + \frac{1}{2}(L_0 + L_1)(1 + \tau_2^2/\tau_1^2)^2$. Then, with probability at least $\frac{1}{2}$, I-NSGD produces a sequence satisfying $\min_{t \leq T}\|\nabla F(w_t)\| \leq \epsilon$ if the total number of iteration $T$ satisfies*

$$T \geq \Lambda \max\left\{\frac{256\Lambda}{\epsilon^4}, \frac{640L_1}{\epsilon^{2-\beta}}, \frac{64(L_0 + L_1) + 128L_1(3\tau_2/\tau_1)^\beta}{\epsilon^2}\right\}. \tag{16}$$

The choices of $B, B' = \mathcal{O}(\tau_1^2)$ are mainly to simplify the symbolic operation during the proof. By deploying normalizing during data pre-processing, the value of $\tau_1$ can be approximately controlled as $\mathcal{O}(1)$ in practice. Thus, Theorem 2 indicates that I-NSGD achieves a sample complexity in the order of $\mathcal{O}(\epsilon^{-4})$ with constant-level batch sizes in generalized-smooth optimization. Compared to the existing studies on normalized/clipped SGD, this convergence result neither requires using extremely large batch sizes nor depending on the bounded error assumption. Through numerical experiments in Section 4 later, we show that it suffices to query a small number of independent samples for I-NSGD in practice.

**Proof outline and novelty:** The independent sampling strategy adopted by I-NSGD naturally decouples stochastic gradient from gradient norm normalization, making it easier to achieve the desired optimization progress in generalized-smooth optimization under relaxed conditions. By the descent lemma, we have that

$$\mathbb{E}_{\xi_B}\left[F(w_{t+1}) - F(w_t)\right] \overset{(i)}{\leq} \frac{-\gamma\left\|\nabla F(w_t)\right\|^2}{h_t^\beta} + \frac{\gamma^2(L_0 + L_1\left\|\nabla F(w_t)\right\|^\alpha)}{2h_t^{2\beta}}\mathbb{E}_{\xi_B}\left[\left\|\nabla f_{\xi_B}(w_t)\right\|^2\right]$$

$$\overset{(ii)}{\leq} \left(\frac{\gamma}{h_t^\beta}\left(-1 + \gamma\frac{L_0 + L_1\left\|\nabla F(w_t)\right\|^\alpha}{h_t^\beta}\right)\right)\left\|\nabla F(w_t)\right\|^2 + \frac{1}{2}\gamma^2\frac{L_0 + L_1\left\|\nabla F(w_t)\right\|^\alpha}{h_t^{2\beta}}\frac{\tau_2^2}{\tau_1^2}, \tag{17}$$

where the expectation (conditioned on $w_t$) in (i) is taken over $\xi_B$ only, and note that $h_t$ involves the independent mini-batch samples $\xi_{B'}$; (ii) leverages Assumption 4 to bound the second moment term $\mathbb{E}_{\xi_B}[\|\nabla f_{\xi_B}(w_t)\|^2]$ by $2\|\nabla F(w_t)\|^2 + \tau_2^2/\tau_1^2$. Then, for the first term in equation 17, we leverage the clipping structure of $h_t$ to bound the coefficient $\gamma(L_0 + L_1\|\nabla F(w)\|^\alpha)/h_t^\beta$ by $\frac{1}{2}$. For the second term in equation 17, we again leverage the clipping structure of $h_t$ and consider two complementary cases:

when $\|\nabla F(w_t)\| \leq \sqrt{1 + \tau_2^2/\tau_1^2}$, this term can be upper bounded by $\frac{1}{2}\gamma^2(L_0 + L_1)(1 + \tau_2^2/\tau_1^2)$; when $\|\nabla F(w_t)\| > \sqrt{1 + \tau_2^2/\tau_1^2}$, this term can be upper bounded by $\frac{\gamma}{4h_t^\beta}\|\nabla F(w_t)\|^2$. Summing them up gives the desired bound. We refer to Lemma 6 in the appendix for more details. Substituting these bounds into equation 17 and rearranging the terms yields that

$$\frac{\gamma}{4h_t^\beta}\left\|\nabla F(w_t)\right\|^2 \leq \mathbb{E}_{\xi_B}\left[F(w_t) - F(w_{t+1})\right] + \frac{1}{2}(L_0 + L_1)\gamma^2(1 + \frac{\tau_2^2}{\tau_1^2})^2.$$

Furthermore, by leveraging the clipping structure of $h_t^\beta$ and Assumption 4, the left hand side can be lower bounded as $\frac{\gamma\|\nabla F(w_t)\|^2}{h_t^\beta} \geq \min\{\gamma\|\nabla F(w_t)\|^2, \frac{\|\nabla F(w_t)\|^{2-\beta}}{20L_1}\}$. Finally, telescoping above inequalities over $t$ and taking expectation leads to the desired bound in equation 16.

As a comparison, in the prior work on clipped SGD (Zhang et al., 2019; 2020), their stochastic gradient and normalization term $h_t$ depend on the same mini-batch of samples, and therefore cannot be treated separately in the analysis. For example, their analysis proposed the following decomposition.

$$\mathbb{E}_{\xi_B}\frac{\|\nabla f_{\xi_B}(w_t)\|^2}{h_t^{2\beta}} = \mathbb{E}_{\xi_B}\frac{\|\nabla F(w_t)\|^2 + \|\nabla f_{\xi_B}(w_t) - \nabla F(w_t)\|^2 + 2\langle\nabla F(w_t), \nabla f_{\xi_B}(w_t) - \nabla F(w_t)\rangle}{h_t^{2\beta}}.$$

Hence their analysis need to assume a constant upper bound for the approximation error $\|\nabla f_{\xi_B}(w_t) - \nabla F(w_t)\|$ in order to obtain a comparable bound to ours.

We also analyze I-NSGD under the generalized PŁ condition by establishing a recursion similar to that proved in Theorem 1. Due to page limitation, we refer to Appendix G for more details.

## 4 EXPERIMENTS

We conduct numerical experiments to compare I-NSGD with other state-of-the-art stochastic algorithms, including the standard SGD (Ghadimi & Lan, 2013), normalized SGD, Clipped SGD (Zhang et al., 2019). The problems we consider are nonconvex phase retrieval and nonconvex distributionally-robust optimization.

### 4.1 NONCONVEX PHASE RETRIEVAL

The phase retrieval problem arises in optics, signal processing, and quantum mechanics (Drenth, 2007). The goal is to recover a signal from measurements where only the intensity is known, and the phase information is missing or difficult to measure. Specifically, denote the underlying object as $x \in \mathbf{R}^d$. Suppose we take $m$ intensity measurements $y_r = |a_r^T x|^2 + n_r$ for $r = 1 \cdots m$, where $a_r$ denotes the measurement vector and $n_r$ is the additive noise. We aim to reconstruct $x$ by solving the following regression problem.

$$\min_{z \in \mathbf{R}^d} f(z) = \frac{1}{2m}\sum_{r=1}^m \left(y_r - \left|a_r^T z\right|^2\right)^2. \tag{18}$$

Such nonconvex function is generalized-smooth with parameter $\alpha = \frac{2}{3}$ (Chen et al., 2023). In this experiment, we generate the initialization $z_0 \sim \mathcal{N}(1, 6)$ and the underlying signal $x \sim \mathcal{N}(0, 0.5)$ with dimension $d = 100$. We take $m = 3k$ measurements with $a_r \sim \mathcal{N}(0, 0.5)$ and $n_r \sim \mathcal{N}(0, 4^2)$.

We implement all the stochastic algorithms with batch size $|B| = 64$, and we choose a small independent batch size $|B'| = 8$ for I-NSGD. We use fine-tuned learning rates for all algorithms, i.e., $\gamma = 5e{-}5$ for SGD, $0.25$ for both normalized SGD and Clipped SGD, and $0.5$ for I-NSGD. We set the maximal gradient clipping constant $45$ and $\delta = 15$ for both Clipped SGD and I-NSGD. Moreover, we set the normalization parameter of I-NSGD as $\beta = \frac{2}{3}$, which matches the generalized-smoothness parameter $\alpha$ of phase retrieval.

Figure 2 (left) shows the comparison of objective value versus sample complexity. It can be seen that our I-NSGD consistently converges faster than other algorithms. This indicates that, the independently-normalized and clipped updates of I-NSGD are more adapted to the underlying generalized-smooth non-convex geometry. In Figure 2 (middle), we test the performance of I-NSGD under different choices of the normalization parameter $\beta = 1, \frac{4}{5}, \frac{2}{3}, \frac{7}{10}, \frac{13}{20}$. It can be seen that I-NSGD converges the fastest as $\beta$ matches the theoretically-suggested value $\frac{2}{3}$, demonstrating the importance of imposing a proper level of gradient normalization in generalized-smooth optimization. In Figure 2 (right), we further explore the effect of the batch size for I-NSGD's independent batch samples $B'$. Specifically, we test batch sizes $|B'| = 4, 8, 16, 32, 64$, while keeping all other hyper-parameters unchanged. The plot shows that I-NSGD can achieve both fast and stable convergence when choosing a very small batch size ($|B'| = 4$ or $8$) for the independent batch samples.

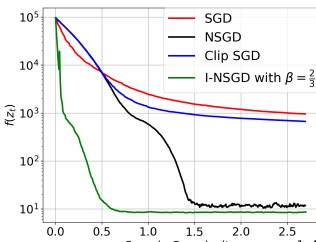 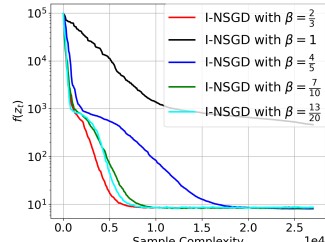 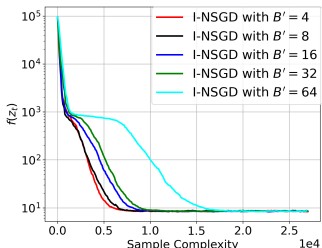

Figure 2: Left: Comparison of I-NSGD and stochastic algorithms. Middle: Performance of I-NSGD with different normalization parameters. Right: Performance of I-NSGD with different independent batch sizes.

## 4.2 DISTRIBUTIONALLY-ROBUST OPTIMIZATION

Distributionally-robust optimization (DRO) is a popular approach to enhance robustness against data distribution shift. We consider the regularized DRO problem $\min_{w \in \mathcal{W}} f(w) = \sup_{\mathbb{Q}} \left\{ \mathbb{E}_{\xi \sim \mathbb{Q}}[\ell_\xi(w)] - \lambda \Psi(\mathbb{P}; \mathbb{Q}) \right\}$, where $\mathbb{Q}, \mathbb{P}$ represents the underlying distribution and the nominal distribution respectively. $\lambda$ denotes a regularization hyper-parameter and $\Psi$ denotes a divergence metric. Under mild technical assumptions, Jin et al. (2021) showed that such a problem has the following equivalent dual formulation

$$\min_{w \in \mathcal{W}} L(w, \eta) = \lambda \mathbb{E}_{\xi \sim P} \Psi^* \left( \frac{\ell_\xi(w) - \eta}{\lambda} \right) + \eta, \tag{19}$$

where $\Psi^*$ denotes the conjugate function of $\Psi$ and $\eta$ is a dual variable. In particular, such dual objective function is generalized-smooth with parameter $\alpha = 1$ (Jin et al., 2021; Chen et al., 2023). In this experiment, we use the life expectancy data (Arshi, 2017). We set $\lambda = 0.01$ and select $\Psi^*(t) = \frac{1}{4}(t+2)_+^2 - 1$, i.e., the conjugate of $\chi^2$-divergence. We adopt the regularized loss $\ell_\xi(\mathbf{w}) = \frac{1}{2}(y_\xi - \mathbf{x}_\xi^\top \mathbf{w})^2 + 0.1 \sum_{j=1}^{34} \ln(1 + |w^{(j)}|)$.

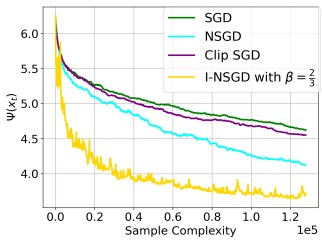 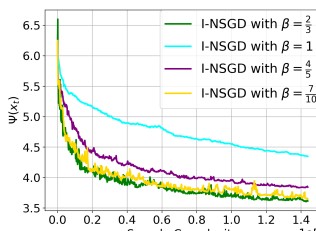 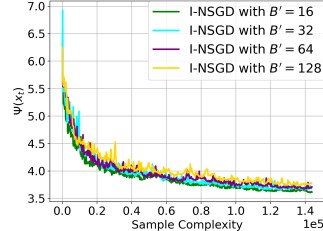

Figure 3: Left: Comparison of I-NSGD and stochastic algorithms. Middle: Performance of I-NSGD with different normalization parameters. Right: Performance of I-NSGD with different independent batch sizes.

We implement all the aforementioned stochastic algorithms with batch size $|B| = 128$, and we choose $|B'| = 16$ for I-NSGD. We use fine-tuned learning rates for all algorithms, i.e., $\gamma = 4e{-}5$ for SGD, $5e{-}2$ for normalized SGD, $0.18$ for Clipped SGD and $0.28$ for I-NSGD. We set the maximal gradient clipping constant $60$ and $\delta = 45$ for both Clipped SGD and I-NSGD.

Figure 3 (left) shows the comparison of objective value versus sample complexity. It can be seen that $\frac{2}{3}$-I-NSGD consistently converges faster than other methods. This indicates independent normalization and clipped updates is also more adapted to function geometry of equation 19. In Figure 3 (middle), we test the performance of I-NSGD under different choices of the normalization parameter $\beta = 1, \frac{4}{5}, \frac{7}{10}, \frac{2}{3}$. It can be seen that $\beta = \frac{2}{3}$ outperforms all other choices in terms of both convergence speed and stability. In Figure 3 (right), we explore the effect of the batch size for I-NSGD's independent batch samples $B'$. We test batch sizes $|B'| = 16, 32, 64, 128$ and keeping all other hyper-parameters unchanged. The plot shows the loss function as a function of sample complexity. We found the batch size of $|B'|$ has little effect the convergence speed. $|B'| = 16$ is sufficient to guarantee fast and stable convergence. This indicates I-NSGD doesn't require a large batch size to ensure convergence, and is more suitable for large-scale problems. To further demonstrate the effectiveness of I-NSGD on problem characterized by generalized smooth condition, we compare our algorithm with additional baseline methods, normalized SGD with momentum (Cutkosky & Mehta, 2020) and SPIDER (Fang et al., 2018) in Phase retrieval and DRO problems. We then conduct ablation study to unify the normalization parameter $\beta$ for all normalization method. In addition, to verify whether I-NSGD can be extended to deep networks characterized by generalized smooth properties, we train ResNet18, ResNet50(He et al., 2016) over CIFAR-10 data (Krizhevsky, 2009) using I-NSGD and other baseline methods, including SGD, Adam (Kingma, 2014), Adagrad (Duchi et al., 2011a), Normalized SGD, Normalized SGD with momentum (Cutkosky & Mehta, 2020) and Clipp-SGD (Zhang et al., 2019). Experiment results show the effectiveness of our proposed I-NSGD framework, which combines independent sampling with clipping updates, and normalization parameter $\beta$. We refer readers to check Section I in appendix for more details about experiments settings and corresponding results.

## 5 CONCLUSION

In this work, we study convergence of normalized gradient descent under generalized smooth and generalized PŁ condition. We propose independent normalized stochastic gradient descent for stochastic setting, achieving same sample complexity under relaxed assumptions. Our results extend the existing boundary of first-order nonconvex optimization and may inspire new developments in this direction. In the future, it is interesting to explore if the popular acceleration method such as stochastic momentum and variance reduction can be combined with independent sampling and normalization to improve the sample complexity.

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

# Appendix

## Table of Contents

## A  PROOF OF DESCENT LEMMA 1

**Lemma 1** *Under Assumption 1, function $f$ satisfies, for any $w, w' \in \mathbf{R}^d$,*

$$f(w) \leq f(w') + \langle \nabla f(w'), w - w' \rangle + \frac{1}{2}(L_0 + L_1 \|\nabla f(w')\|^\alpha)\|w - w'\|^2. \tag{3}$$

**Proof 1** *Use fundamental theorem of calculus, we have*

$$f(w') - f(w) - \langle \nabla f(w), w' - w \rangle$$
$$= \int_0^1 \langle \nabla f(w_\theta), w' - w \rangle \mathrm{d}\theta - \int_0^1 \langle \nabla f(w), w' - w \rangle \mathrm{d}\theta,$$

*where $w_\theta = \theta w' + (1-\theta)w$. Since the integration integrates over $w_\theta$, integrating second term doesn't affect the result. Now replacing above term by $\mathcal{L}_{asym}^*(\alpha)$ condition, we have*

$$f(w') - f(w) - \langle \nabla f(w), w' - w \rangle$$
$$= \int_0^1 \langle \nabla f(w_\theta), w' - w \rangle \mathrm{d}\theta - \int_0^1 \langle \nabla f(w), w' - w \rangle \mathrm{d}\theta$$

$$= \int_0^1 \langle \nabla f(w_\theta) - \nabla f(w), w' - w \rangle \mathrm{d}\theta$$

$$\leq \int_0^1 \big\| \nabla f(w_\theta) - \nabla f(w) \big\| \big\| w' - w \big\| \mathrm{d}\theta$$

$$\leq \int_0^1 \theta (L_0 + L_1 \big\| \nabla f(w_t) \big\|^\alpha) \big\| w' - w \big\|^2 \mathrm{d}\theta$$

$$= \frac{1}{2}(L_0 + L_1 \big\| \nabla f(w_t) \big\|^\alpha) \big\| w' - w \big\|^2, \tag{20}$$

*where the first inequality is due to Cauchy-schwarz inequality, the second inequality is due to Assumption 1 regarding on $\mathcal{L}_{asym}^*(\alpha)$ generalized smooth. Reorganize above inequality gives us the desired result.*

# B  PROOF OF DESCENT LEMMA UNDER GENERALIZED PŁ CONDITION

**Lemma 3** *For any $x \geq 0$, $C \in [0,1]$, $\Delta > 0$, and $0 \leq w \leq w'$ such that $\Delta \geq w' - w$, we have the following inequality hold*

$$Cx^w \leq x^{w'} + C^{\frac{w'}{\Delta}}. \tag{21}$$

The proof details for this lemma can be found at Chen et al. (2023), Lemma E.2 at Appendix.

**Lemma 4 (Descent Lemma under Generalized PL condition)** *Let Assumption 1 and 2 hold. Apply NGD, choose $\beta \in [\alpha, 1]$ or $\beta \in (\alpha, 1]$, when $\alpha \in (0,1]$ or $\alpha = 0$ respectively. Set the target accuracy $\epsilon$ satisfy $0 \leq \epsilon \leq \min\{1, 1/2\mu\}$. Define the step size $\gamma = \frac{(2\mu\epsilon)^{\beta/\rho}}{8(L_0+L_1)+1}$. Denote $\Delta_t = f(w_t) - f^*$, then we have descent lemma*

$$\Delta_{t+1} \leq \Delta_t - \frac{\gamma(2\mu)^{\frac{2-\beta}{\rho}}}{4} \Delta_t^{\frac{2-\beta}{\rho}}. \tag{22}$$

**Proof 2** *Start from descent lemma 1, we have*

$$f(w_{t+1}) - f(w_t)$$

$$\overset{(i)}{\leq} \nabla f(w_t)^\top (w_{t+1} - w_t) + \frac{1}{2}(L_0 + L_1 \big\| \nabla f(w_t) \big\|^\alpha) \big\| w_{t+1} - w_t \big\|^2$$

$$\overset{(ii)}{=} -\gamma \big\| \nabla f(w_t) \big\|^{2-\beta} + \frac{\gamma}{4} \big( 2L_0\gamma \cdot \big\| \nabla f(w_t) \big\|^{2-2\beta} + 2L_1\gamma \cdot \big\| \nabla f(w_t) \big\|^{2+\alpha-2\beta} \big)$$

$$\overset{(iii)}{\leq} -\gamma \big\| \nabla f(w_t) \big\|^{2-\beta} + \frac{\gamma}{4} \big( 2 \big\| \nabla f(w_t) \big\|^{2-\beta} + (2L_0\gamma)^{\frac{2}{\beta}-1} + (2L_1\gamma)^{\frac{2}{\beta}-1} \big)$$

$$\overset{(iv)}{\leq} -\frac{\gamma}{2} \big\| \nabla f(w_t) \big\|^{2-\beta} + \gamma^{\frac{2}{\beta}} (2L_0 + 2L_1)^{\frac{2}{\beta}-1}$$

$$\overset{(v)}{\leq} -\frac{\gamma}{2} \big\| \nabla f(w_t) \big\|^{2-\beta} + \frac{(2\mu\epsilon)^{\frac{2}{\rho}}}{(8(L_0+L_1)+1)^{\frac{2}{\beta}}} (\frac{1}{4})^{\frac{2}{\beta}-1} (8(L_0+L_1)+1)^{\frac{2}{\beta}-1}$$

$$\overset{(vi)}{\leq} -\frac{\gamma}{2} \big\| \nabla f(w_t) \big\|^{2-\beta} + \frac{1}{4} \frac{(2\mu\epsilon)^{\frac{\beta}{\rho}} (2\mu\epsilon)^{\frac{2-\beta}{\rho}}}{8(L_0+L_1)+1}$$

$$= -\frac{\gamma}{2} \big\| \nabla f(w_t) \big\|^{2-\beta} + \frac{\gamma}{4}(2\mu\epsilon)^{\frac{2-\beta}{\rho}}, \tag{23}$$

*where (i) follows from lemma 1; (ii) follows from update rule of NGD, namely replacing $w_{t+1} - w_t$ by $\frac{\nabla f(w_t)}{||\nabla f(w_t)||^\beta}$, (iii) follows from aggregates constant term by 6 and utilize technical lemma 3 by letting $\omega' = 2 - \beta$, $\Delta = \beta$ and applying it to $2L_0\gamma||\nabla f(w_t)||^{2-2\beta}$, $2L_1\gamma||\nabla f(w_t)||^{2+\alpha-2\beta}$ twice gives the desired result; (iv) follows from $a^\tau + b^\tau \leq (a+b)^\tau$ holds for $\tau = 2/\beta - 1 > 1$ and $a, b \geq 0$, (v) follows from the step size rule $\gamma = (2\mu\epsilon)^{\beta/\rho}/(8(L_0 + L_1) + 1)$, (vi) following from the fact $0 < \beta \leq 1$, thus $\frac{1}{4}^{(2/\beta)-1} < \frac{1}{4}$.*

*For function satisfying generalized PŁ-condition proposed in definition 5, we have*

$$\left\|\nabla f(w)\right\| \geq (2\mu)^{\frac{1}{\rho}}\left(f(w) - f^*\right)^{\frac{1}{\rho}}.$$

*This is equivalent as*

$$\left\|\nabla f(w)\right\|^{2-\beta} \geq (2\mu)^{\frac{2-\beta}{\rho}}\left(f(w) - f^*\right)^{\frac{2-\beta}{\rho}}. \tag{24}$$

*Substitute equation 24 into equation 25, we have*

$$f(w_{t+1}) - f(w_t) \leq -\frac{\gamma}{2}(2\mu)^{\frac{2-\beta}{\rho}}\left(f(w_t) - f^*\right)^{\frac{2-\beta}{\rho}} + \frac{\gamma}{4}(2\mu\epsilon)^{\frac{2-\beta}{\rho}}.$$

*Subtract $f^*$ on both sides, it is equivalent as*

$$f(w_{t+1}) - f^* \leq f(w_t) - f^* - \frac{\gamma}{2}(2\mu)^{\frac{2-\beta}{\rho}}\left(f(w) - f^*\right)^{\frac{2-\beta}{\rho}} + \frac{\gamma}{4}(2\mu\epsilon)^{\frac{2-\beta}{\rho}}.$$

*Now, denote $\Delta_t = f(w_t) - f^*$, we have the equivalent representation*

$$\Delta_{t+1} \leq \Delta_t - \frac{\gamma(2\mu)^{\frac{2-\beta}{\rho}}}{2}\Delta_t^{\frac{2-\beta}{\rho}} + \frac{\gamma}{4}(2\mu\epsilon)^{\frac{2-\beta}{\rho}}. \tag{25}$$

*By Choosing the stopping criterion as*

$$T = \inf\left\{t \mid \Delta_t = f(w_t) - f^* \leq \epsilon\right\}, \text{ where } 0 < \epsilon \leq \min\{1, \frac{1}{2\mu}\}.$$

*We conclude before algorithm terminates, $\Delta_t > \epsilon$ for all $t \leq T$, thus $-\frac{\gamma(2\mu)^{\frac{2-\beta}{\rho}}}{2}\Delta_t^{\frac{2-\beta}{\rho}}$ dominates $\frac{\gamma}{4}(2\mu\epsilon)^{\frac{2-\beta}{\rho}}$. Moreover, by definition of $\beta$, we have $\frac{2-\beta}{\rho} > 0$ and thus*

$$\Delta_t^{\frac{2-\beta}{\rho}} > \epsilon^{\frac{2-\beta}{\rho}},$$

*which is equivalent to claim*

$$\frac{\gamma}{4}(2\mu)^{\frac{2-\beta}{\rho}}\Delta_t^{\frac{2-\beta}{\rho}} > \frac{\gamma}{4}(2\mu\epsilon)^{\frac{2-\beta}{\rho}}.$$

*Thus, equation 25 reduces to relaxed descent inequality*

$$\Delta_{t+1} \leq \Delta_t - \frac{\gamma(2\mu)^{\frac{2-\beta}{\rho}}}{4}\Delta_t^{\frac{2-\beta}{\rho}}. \tag{26}$$

## C  PROOF OF THEOREM 1

**Theorem 1 (Convergence of NGD)** *Let Assumptions 1 and 2 hold. Choose learning rate $\gamma = \frac{(2\mu\epsilon)^{\beta/\rho}}{8(L_0+L_1)+1}$ where $\epsilon$ denotes the target accuracy, and set $\alpha \leq \beta \leq 1$. Then, the following statements hold.*

- *If $0 < \rho < 2 - \beta$, then we have*

$$\Delta_t = \mathcal{O}\Big(\big(\frac{\rho}{(2-\beta-\rho)\gamma t}\big)^{\frac{\rho}{2-\rho-\beta}}\Big). \tag{6}$$

*Furthermore, in order to achieve $\Delta_t \leq \epsilon$, the total number of iteration satisfies $T = \mathcal{O}\big((\frac{1}{\epsilon})^{\frac{\beta}{\rho}}\big)$ if $2 - 2\beta < \rho < 2 - \beta$, and $T = \mathcal{O}\big((\frac{1}{\epsilon})^{\frac{2-\rho-\beta}{\rho}}\big)$ if $0 < \rho \leq 2 - 2\beta$.*

- *If $\rho = 2 - \beta$ and choose $\epsilon$ such that $\gamma < \frac{2}{\mu}$, then we have*

$$\Delta_t = \mathcal{O}\Big(\big(1 - \frac{\gamma\mu}{2}\big)^t\Big). \tag{7}$$

*In order to achieve $\Delta_t \leq \epsilon$, the total number of iteration satisfies $T = \mathcal{O}\big((\frac{1}{\epsilon})^{\frac{\beta}{\rho}} \log \frac{1}{\epsilon}\big)$.*

- *If $\rho > 2 - \beta$, then there exists $T_0 \in \mathbf{N}$ such that for all $t \geq T_0$, we have*

$$\Delta_t = \mathcal{O}\Big(\big(\frac{\Delta_{T_0}}{\gamma^{\frac{\rho}{\rho+\beta-2}}}\big)^{\frac{\rho}{2-\beta}^{t-T_0}}\Big). \tag{8}$$

*In order to achieve $\Delta_t \leq \epsilon$, the total number of iterations after $T_0$ satisfies $T = \Omega\big(\log((\frac{1}{\epsilon})^{\frac{\beta}{\rho+\beta-2}})\big)$.*

**Proof 3** *We divide the convergence proof of theorem 1 into three cases depending on the value of $\beta$ and $\rho$.*

***Case I: When $\rho < 2 - \beta$***
*This is equivalent as $\frac{2-\beta}{\rho} > 1$. Now denote $\theta = \frac{2-\beta}{\rho}$. Since $\theta > 1$, we have following inequalities hold*

$$\Delta_{t+1} \leq \Delta_t$$
$$\Delta_{t+1}^{\theta} \leq \Delta_t^{\theta}$$
$$\Delta_{t+1}^{-\theta} \geq \Delta_t^{-\theta}. \tag{27}$$

*Now define an auxiliary function $\Phi(t) = \frac{1}{\theta-1}t^{1-\theta}$. Its derivative can be computed via $\Phi'(t) = -t^{-\theta}$ We now divide the last inequality at equation 27 into two different cases for analysis. One is the case where $\Delta_{t+1}^{-\theta} \leq 2\Delta_t^{-\theta}$, Another is the case where $\Delta_{t+1}^{-\theta} \geq 2\Delta_t^{-\theta}$.*

*When $\Delta_t^{-\theta} \leq \Delta_{t+1}^{-\theta} \leq 2\Delta_t^{-\theta}$, we have*

$$\begin{aligned}
\Phi(\Delta_{t+1}) - \Phi(\Delta_t) &= \int_{\Delta_t}^{\Delta_{t+1}} \Phi'(t)\mathrm{d}t = \int_{\Delta_{t+1}}^{\Delta_t} t^{-\theta}\mathrm{d}t \\
&\geq (\Delta_t - \Delta_{t+1})\Delta_t^{-\theta} \\
&\geq (\Delta_t - \Delta_{t+1})\frac{\Delta_{t+1}^{-\theta}}{2} \\
&\geq \frac{\gamma(2\mu)^{\theta}}{4}\Delta_t^{\theta}\frac{\Delta_{t+1}^{-\theta}}{2} \\
&\geq \frac{\gamma(2\mu)^{\theta}}{4}\Delta_{t+1}^{\theta}\frac{\Delta_{t+1}^{-\theta}}{2} = \frac{\gamma(2\mu)^{\theta}}{8}.
\end{aligned}$$

*The first inequality is using mean value theorem such that $\Phi(\Delta_{t+1}) - \Phi(\Delta_t) = |\Delta_{t+1} - \Delta_t||\Phi'(\xi)|$, where $\xi \in [\Delta_t, \Delta_{t+1}]$. Since $\Phi(\Delta_{t+1}) - \Phi(\Delta_t) \geq 0$, taking absolute value has no effect. Since $\theta > 0$,*

$|\Phi'(t)| = t^{-\theta}$ *is monotone decreasing. Thus, we always have* $\Delta_t^{-\theta} \le \Phi'(\xi) \le \Delta_{t+1}^{-\theta}$ *for any* $\xi \in [\Delta_t, \Delta_{t+1}]$; *The second inequality uses the fact* $\Delta_{t+1}^{-\theta} \le 2\Delta_t^{-\theta}$; *The third inequality is due to the recursion* $\Delta_t - \Delta_{t+1} \ge \frac{\gamma(2\mu)^\theta}{4}\Delta_t^\theta$; *The last inequality uses the fact that* $\Delta_t^\theta > \Delta_{t+1}^\theta$ *for all* $\theta > 0$.

*When* $\Delta_{t+1}^{-\theta} > 2\Delta_t^{-\theta}$, *it holds that* $\Delta_{t+1}^{1-\theta} = (\Delta_{t+1}^{-\theta})^{\frac{1-\theta}{-\theta}} > 2^{\frac{1-\theta}{-\theta}}\Delta_t^{1-\theta}$. *Then, we have*

$$
\begin{aligned}
\Phi(\Delta_{t+1}) - \Phi(\Delta_t) &= \frac{1}{\theta - 1}(\Delta_{t+1}^{1-\theta} - \Delta_t^{1-\theta}) \\
&\ge \frac{1}{\theta - 1}\big((2)^{\frac{\theta-1}{\theta}} - 1\big)\Delta_t^{1-\theta} \\
&\ge \frac{1}{\theta - 1}\big((2)^{\frac{\theta-1}{\theta}} - 1\big)\Delta_0^{1-\theta},
\end{aligned}
$$

*where the first inequality is due to the recursion* $\Delta_{t+1}^{1-\theta} = (\Delta_{t+1}^{-\theta})^{\frac{1-\theta}{-\theta}} > 2^{\frac{1-\theta}{-\theta}}\Delta_t^{1-\theta}$; *the last inequality is due to the fact the sequence* $\{\Delta_t\}_{t=1}^T$ *is non-increasing.*

*Now put the expression of* $\theta$ *in and denote*

$$
C = \min\Big\{ \frac{\gamma(2\mu)^{\frac{2-\beta}{\rho}}}{8}, \frac{\rho}{2-\beta-\rho}(2^{\frac{2-\beta-\rho}{2-\beta}} - 1)\Delta_0^{\frac{2-\beta-\rho}{\rho}} \Big\}.
$$

*We conclude for all* $t$, *we have*

$$
\Phi(\Delta_t) \ge \sum_{i=0}^{t-1} \Phi(\Delta_{i+1}) - \Phi(\Delta_i) \ge Ct,
$$

*Thus, we have*

$$
\Delta_t \le \Big( \frac{\rho}{(2-\beta-\rho)Ct} \Big)^{\frac{\rho}{2-\rho-\beta}} = \mathcal{O}\Big( \frac{\rho}{(2-\beta-\rho)\gamma t} \Big)^{\frac{\rho}{2-\rho-\beta}}, \tag{28}
$$

*When* $C = \frac{\rho}{2-\beta-\rho}(2^{(2-\beta-\rho)/(2-\beta)} - 1)\Delta_0^{(2-\beta-\rho)/\rho}$, *in order to make* $\Delta_t \le \epsilon$, *we have*

$$
\rho/(2-\rho-\beta)\log\big((2-\beta-\rho)Ct/\rho\big) = \log\big(1/\epsilon\big),
$$

*which indicates* $T = \mathcal{O}\big((\frac{1}{\epsilon})^{\frac{2-\rho-\beta}{\rho}}\big)$.

*When* $C = \widetilde{C}\epsilon^{\beta/\rho} = \Theta(\epsilon^{\beta/\rho})$, *in order to make* $\Delta_t \le \epsilon$, *taking logarithm we have*

$$
\log\big(\frac{(2-\beta-\rho)\widetilde{C}\epsilon^{\beta/\rho}t}{\rho}\big) = \log\big((1/\epsilon)^{\frac{2-\beta-\rho}{\rho}}\big).
$$

*Re-arrange above equality, we have* $T = \mathcal{O}\big((\frac{1}{\epsilon})^{\frac{2-\rho-\beta}{\rho}} + (\frac{1}{\epsilon})^{\frac{\beta}{\rho}}\big)$. *Thus, when* $0 \le \rho \le 2 - 2\beta$, *we have* $T = \mathcal{O}(\frac{1}{\epsilon})^{\frac{2-\rho-\beta}{\rho}}$; *when* $2 - 2\beta < \rho \le 2 - \beta$, *we have* $T = \mathcal{O}\big((\frac{1}{\epsilon})^{\frac{\beta}{\rho}}\big)$.

**Case II: When** $\rho = 2 - \beta$, *It is equivalent to claim* $\beta, \rho$ *satisfies* $\frac{2-\beta}{\rho} = 1$, *descent inequality equation 26 reduces to*

$$
\Delta_{t+1} \le \Delta_t - \frac{\gamma\mu}{2}\Delta_t = (1 - \frac{\gamma\mu}{2})\Delta_t,
$$

As long as $\mu < \frac{2}{\gamma}$, the $\Delta_t$ converges to 0.

$$\Delta_t \leq (1 - \frac{\gamma\mu}{2})^t \Delta_0 = \mathcal{O}\Big((1 - \frac{\gamma\mu}{2})^t\Big).$$

However, since the step-size rule of $\gamma$ includes target accuracy $\epsilon$. The convergence rate is not a standard linear convergence. To obtain a $\epsilon$-stationary point, we have

$$\Delta_t \leq (1 - \frac{\gamma\mu}{2})^t \Delta_0 \leq \exp(-\frac{\gamma\mu t}{2})\Delta_0 \leq \epsilon, \tag{29}$$

which gives us iteration complexity

$$T = \frac{2}{\gamma\mu} \log(\frac{\Delta_0}{\epsilon}) = \mathcal{O}\big((\frac{1}{\epsilon})^{\frac{\beta}{\rho}} \log(\frac{1}{\epsilon})\big).$$

***Case II: When $\rho > 2 - \beta$***

This case is equivalent to $\frac{2-\beta}{\rho} < 1$. For simplicity, denote $C = \frac{(2\mu)^{(2-\beta)/\rho}}{4}$ and $\omega = \frac{\rho}{2-\beta}$. The sequence generated by recursion equation 26 is guaranteed to converge to 0 when $\epsilon \downarrow 0$.

For simplicity, rewriting equation 26 as $\Delta_{t+1} \leq \Delta_t - C\gamma\Delta_t^{1/\omega}$. Notice $\Delta_t \geq 0$, $C > 0$, $\{\Delta\}_t$ is non-increasing. Now suppose the sequence $\{\Delta\}_t$ converge to a positive constant, denoted as $D$. There must exists $0 < \tilde{\varepsilon} < D$ such that $\Delta_t > \tilde{\varepsilon}$ for all $t$. Then we have

$$\Delta_{t+1} \leq \Delta_t - C\gamma\Delta_t^{\frac{1}{\omega}} \leq \Delta_t - C\gamma\tilde{\varepsilon}^{\frac{1}{\omega}}.$$

Re-organize above recursion, we have $TC\gamma\tilde{\varepsilon}^{1/\omega} \leq \sum_{t=0}^{T-1} \Delta_t - \Delta_{t+1} \leq \Delta_0$, which is equivalent as $T \leq \frac{\Delta_0}{C\gamma\tilde{\varepsilon}^{1/\omega}} < \infty$. This fact contradicts to $\Delta > \tilde{\varepsilon}$ for arbitrary $t$. In conclusion, as long as equation 26 holds, the sequence $\{\Delta_t\}$ converges to 0 as $\epsilon \downarrow 0$.

Next, we determine the local convergence rate. When $\Delta_t$ is small enough, $\Delta_{t+1}^{1/\omega}$ will dominate $\Delta_{t+1}$ order-wisely since $1/\omega < 1$. This leads to refined recursion

$$C\gamma\Delta_{t+1}^{\frac{1}{\omega}} \leq \Delta_{t+1} + C\gamma\Delta_{t+1}^{\frac{1}{\omega}} \leq \Delta_{t+1} + C\gamma\Delta_t^{\frac{1}{\omega}} \leq \Delta_t.$$

The first inequality is due to non-negativity of $\Delta_{t+1}$, the second inequality is due to $\Delta_{t+1} \leq \Delta_t$, the third inequality is a re-organization of equation 26. Denote $T_0 = \inf\{t \in \mathbf{N}|\Delta_t/(C\gamma)^{\omega/\omega-1} < 1\}$, then we have

$$\Delta_{t+1} \leq (C\gamma)^{-\omega}\Delta_t^\omega = (C\gamma)^{-\omega-\omega^2-...-\omega^{t-T_0}}\Delta_{T_0}^{\omega^{t-T_0}}$$

$$= (C\gamma)^{\frac{\omega(1-\omega^{t-T_0})}{\omega-1}}\Delta_{T_0}^{\omega^{t-T_0}}$$

$$= (C\gamma)^{\omega/\omega-1}\big((C\gamma)^{\omega/\omega-1}\big)^{\omega^{-t-T_0}}\Delta_{T_0}^{\omega^{t-T_0}}. \tag{30}$$

Since $(C\gamma)^{\omega/\omega-1}$ only effects order of convergence up to a constant. To simplify analysis, denote $\hat{C} = (\frac{C(2\mu)^{\beta/\rho}}{8(L_0+L_1)+1})^{\omega/\omega-1}$ and then we have $(C\gamma)^{\omega/\omega-1} = \hat{C}\epsilon^{\beta/\rho+\beta-2} \leq \hat{C}$. since $0 \leq \epsilon \leq \min\{1, 1/2\mu\}$, we further reduce the recursion to

$$\Delta_{t+1} \leq (C\gamma)^{\omega/\omega-1}\big((C\gamma)^{\omega/\omega-1}\big)^{\omega^{-t-T_0}}\Delta_{T_0}^{\omega^{t-T_0}}$$

$$\leq \hat{C}\Big((C\gamma)^{\frac{\omega}{\omega-1}}\Big)^{-\omega^{t-T_0}}\Delta_{T_0}^{\omega^{t-T_0}}$$

$$= \mathcal{O}\Big(\Big(\frac{\Delta_{T_0}}{\gamma^{\omega/\omega-1}}\Big)^{\omega^{t-T_0}}\Big). \tag{31}$$

*Taking logarithm and multiply negative sign on both sides of equation 31. We have*

$$\log(\frac{\hat{C}}{\epsilon}) = \omega^{t-T_0} \cdot \log\big(\frac{(C\gamma)^{\frac{\omega}{\omega-1}}}{\Delta_{T_0}}\big).$$

*Now, extract $\epsilon^{\beta/(\rho+\beta-2)}$ from $(C\gamma)^{\omega/\omega-1}/\Delta_{T_0}$. We have*

$$\log\left((C\gamma)^{\frac{\omega}{\omega-1}}/\Delta_{T_0}\right) = \log\left((\hat{C}/\Delta_{T_0}) \cdot \epsilon^{\frac{\beta}{\rho+\beta-2}}\right) \leq (\hat{C}/\Delta_{T_0}) \cdot \epsilon^{\frac{\beta}{\rho+\beta-2}},$$

*where the last inequality is due to the fact $\log(x) \leq x, \forall x > 0$. Taking logarithm again, we have*

$$t - T_0 = \Omega\big(\log\big((\frac{1}{\epsilon})^{\frac{\beta}{\rho+\beta-2}}\big)\big).$$

## D  PROOF OF THEOREM 2

**Theorem 2 (Convergence of I-NSGD)** *Let Assumptions 1, 3 and 4 hold. For the I-NSGD algorithm, choose learning rate $\gamma = \min\{\frac{1}{4L_0}, \frac{1}{4L_1}, \frac{1}{\sqrt{T}}, \frac{1}{8L_1(3\tau_2/\tau_1)^\beta}\}$, batch sizes $B = 2\tau_1^2$, $B' = 16\tau_1^2$ and $\delta = \frac{\tau_2}{\tau_1}$. Denote $\Lambda := F(w_0) - F^* + \frac{1}{2}(L_0 + L_1)(1 + \tau_2^2/\tau_1^2)^2$. Then, with probability at least $\frac{1}{2}$, I-NSGD produces a sequence satisfying $\min_{t \leq T} \|\nabla F(w_t)\| \leq \epsilon$ if the total number of iteration $T$ satisfies*

$$T \geq \Lambda \max\left\{\frac{256\Lambda}{\epsilon^4}, \frac{640L_1}{\epsilon^{2-\beta}}, \frac{64(L_0 + L_1) + 128L_1(3\tau_2/\tau_1)^\beta}{\epsilon^2}\right\}. \tag{16}$$

**Proof 4** *Start from descent lemma equation 3 and put the update rule of I-NSGD, equation 13 in, we have*

$$F(w_{t+1}) - F(w_t) \leq \nabla F(w_t)^\top(w_{t+1} - w_t) + \frac{1}{2}(L_0 + L_1\|\nabla F(w_t)\|^\alpha)\|w_{t+1} - w_t\|^2$$

$$= -\gamma\frac{\nabla F(w_t)^\top \nabla f_{\xi_B}(w_t)}{h_t^\beta} + \frac{1}{2}\gamma^2(L_0 + L_1\|\nabla F(w_t)\|^\alpha)\frac{\|\nabla f_{\xi_B}(w_t)\|^2}{h_t^{2\beta}}. \tag{32}$$

*Since the update rule using I-NSGD formulates a random trajectory in terms of $w_t$, taking expectation over $\xi_B$ and $w_t$, using condition expectation rule, we have*

$$\mathbb{E}_{w_t}\big[\big[\mathbb{E}_{\xi_B}\big[F(w_{t+1}) - F(w_t)|w_t\big]\big] \leq \mathbb{E}_{w_t}\Big[\frac{-\gamma\mathbb{E}_{\xi_B}[\|\nabla F(w_t)\|^2|w_t]}{h_t^\beta}$$

$$+ \frac{1}{2}\gamma^2(L_0 + L_1\|\nabla F(w_t)\|^\alpha)\frac{\mathbb{E}_{\xi_B}[\|\nabla f_{\xi_B}(w_t)\|^2|w_t]}{h_t^{2\beta}}\Big].$$

*When the expectation is conditioned on $w_t$, we can simplify $\mathbb{E}_{\xi_B}\big[\|\nabla F(w_t)\|^2|w_t\big]$ into $\|\nabla F(w_t)\|^2$ since $\nabla F(w_t)$ is deterministic over $\xi_B$. Additionally, by remarks induced by assumption 4, when conditioned over $w_t$, randomness only comes from $\xi_{B'}$, thus we have*

$$\mathbb{E}_{\xi_B}\big[\|\nabla f_{\xi_B}(w_t)\|^2|w_t\big] \leq \underbrace{(\frac{2\tau_1^2}{B} + 1)\|\nabla F(w_t)\|^2}_{\text{See equation 46}} + \frac{2\tau_2^2}{B}.$$

*Let $B = 2\tau_1^2$, above inequality reduces to*

$$\mathbb{E}_{\xi_B}\big[\|\nabla f_{\xi_B}(w_t)\|^2|w_t\big] \leq 2\|\nabla F(w)\|^2 + \frac{\tau_2^2}{\tau_1^2}. \tag{33}$$

*Put equation 33 into above descent lemma, we have*

$$\mathbb{E}_{w_t}\left[\mathbb{E}_{\xi_B}\left[F(w_{t+1}) - F(w_t)|w_t\right]\right]$$

$$\leq \mathbb{E}_{w_t}\left[-\gamma\frac{\left\|\nabla F(w_t)\right\|^2}{h_t^\beta} + \frac{1}{2}\gamma^2(L_0 + L_1\|\nabla F(w_t)\|^\alpha)\frac{\mathbb{E}_{\xi_B}\left[\left\|\nabla f_{\xi_B}(w_t)\right\|^2|w_t\right]}{h_t^{2\beta}}\right]$$

$$\leq \mathbb{E}_{w_t}\left[-\gamma\frac{\left\|\nabla F(w_t)\right\|^2}{h_t^\beta} + \frac{1}{2}\gamma^2(L_0 + L_1\|\nabla F(w_t)\|^\alpha)\frac{2\left\|\nabla F(w_t)\right\|^2 + \tau_2^2/\tau_1^2}{h_t^{2\beta}}\right].$$

$$= \mathbb{E}_{w_t}\left[\left(\frac{\gamma}{h_t^\beta}\left(-1 + \gamma\frac{L_0 + L_1\left\|\nabla F(w_t)\right\|^\alpha}{h_t^\beta}\right)\right)\|\nabla F(w_t)\|^2 + \frac{1}{2}\gamma^2\frac{L_0 + L_1\left\|\nabla F(w_t)\right\|^\alpha}{h_t^{2\beta}}\frac{\tau_2^2}{\tau_1^2}\right]. \quad (34)$$

*By clipping structure and step size rule, from where we know $\frac{1}{h_t^\beta} = \min\left\{1, \frac{1}{4L_1\gamma(2\|\nabla f_{\xi_{B'}}(w_t)\| + \frac{\tau_2}{\tau_1})^\beta}\right\} < 1$ and $\gamma \leq \frac{1}{4L_0}$, we have*

$$\frac{\gamma L_0}{h_t^\beta} < \gamma L_0 \leq \frac{1}{4}, \quad (35)$$

$$\frac{\gamma L_1\left\|\nabla F(w_t)\right\|^\alpha}{h_t^\beta} \leq \frac{1}{4}. \quad (36)$$

*The last inequality in equation 36 utilizes lemma 2, from where we know*

$$\frac{1}{4}h_t^\beta \overset{(i)}{\geq} \frac{1}{4}h_t^\alpha$$

$$\overset{(ii)}{=} \frac{1}{4}(h_t^\beta)^{\frac{\alpha}{\beta}}$$

$$\overset{(iii)}{\geq} \frac{1}{4}(4\gamma L_1)^{\frac{\alpha}{\beta}}\left(2\left\|\nabla f_{\xi_{B'}}(w_t)\right\| + \frac{\tau_2}{\tau_1}\right)^\alpha$$

$$\overset{(iv)}{\geq} \frac{1}{4}(4\gamma L_1)\left(2\left\|\nabla f_{\xi_{B'}}(w_t)\right\| + \frac{\tau_2}{\tau_1}\right)^\alpha$$

$$\overset{(v)}{\geq} \gamma L_1\left\|\nabla F(w_t)\right\|^\alpha, \quad (37)$$

*where (i) utilizes the fact $h_t \geq 1$ and $\beta \geq \alpha$; (iii) utilize the fact that $h_t^\beta \geq 4L_1\gamma(2\left\|\nabla f_{\xi_{B'}}(w_t)\right\| + \frac{\tau_2}{\tau_1})^\beta$; (iv) utilizes the fact that $\gamma \leq \frac{1}{4L_1}$, thus $(4\gamma L_1) \leq (4\gamma L_1)^{\alpha/\beta}$ since $\beta \in [\alpha, 1]$; (v) utilizes the fact stated fact in Lemma 2.*

*Combining equation 36, above descent lemma further reduces to*

$$\mathbb{E}_{w_t}\left[\mathbb{E}_{\xi_B}\left[F(w_{t+1}) - F(w_t)|w_t\right]\right]$$

$$\leq \mathbb{E}_{w_t}\left[-\frac{\gamma}{2h_t^\beta}\left\|\nabla F(w_t)\right\|^2 + \underbrace{\frac{1}{2}\gamma^2\frac{L_0 + L_1\left\|\nabla F(w_t)\right\|^\alpha}{h_t^{2\beta}}\frac{\tau_2^2}{\tau_1^2}}_{\text{Term 1, See Lemma 6}}\right]$$

$$\leq \mathbb{E}_{w_t}\left[-\frac{\gamma}{2h_t^\beta}\left\|\nabla F(w_t)\right\|^2 + \frac{1}{2}\gamma^2(L_0 + L_1)(1 + \frac{\tau_2^2}{\tau_1^2})^2 + \frac{\gamma}{4h_t^\beta}\left\|\nabla F(w_t)\right\|^2\right]$$

$$= \mathbb{E}_{w_t}\Big[ - \frac{\gamma}{4h_t^\beta}\big\|\nabla F(w_t)\big\|^2 + \frac{1}{2}\gamma^2(L_0+L_1)(1+\frac{\tau_2^2}{\tau_1^2})^2\Big], \tag{38}$$

*where the last inequality utilize the inequality stated in Lemma 6, equation equation 48.*
*Re-organize the inequality by putting the negative term to LHS*

$$\mathbb{E}_{w_t}\Big[ \frac{\gamma}{4h_t^\beta}\big\|\nabla F(w_t)\big\|^2\Big] \leq \mathbb{E}_{w_t}\Big[\mathbb{E}_{\xi_B}\big[F(w_t)-F(w_{t+1})|w_t\big]\Big] + \frac{1}{2}(L_0+L_1)\gamma^2(1+\frac{\tau_2^2}{\tau_1^2})^2. \quad \forall t\in[T] \tag{39}$$

*In order to express the LHS into a more tractable form, we want to express $\frac{\|\nabla F(w_t)\|^2}{h_t^\beta}$ explicitly in a simpler form. Using the fact $\frac{1}{(a+b)^\beta}\geq \min\{\frac{1}{(2a)^\beta},\frac{1}{(2b)^\beta}\}$. We have*

$$\gamma\frac{\big\|\nabla F(w_t)\big\|^2}{h_t^\beta} \stackrel{(i)}{=} \gamma\min\Big\{1, \frac{1}{4L_1\gamma(2\|\nabla f_{\xi_{B'}}(w_t)\|+\frac{\tau_2}{\tau_1})^\beta}\Big\}\big\|\nabla F(w_t)\big\|^2$$

$$\stackrel{(ii)}{\geq} \gamma\min\Big\{1, \frac{1}{4L_1\gamma\big(\frac{5}{2}\|\nabla F(w_t)\|+\frac{3\tau_2}{2\tau_1}\big)^\beta}\Big\}\big\|\nabla F(w_t)\big\|^2$$

$$\stackrel{(iii)}{\geq} \gamma\min\Big\{1, \frac{1}{4L_1\gamma\big(5\|\nabla F(w_t)\|\big)^\beta}, \frac{1}{4L_1\gamma\big(\frac{3\tau_2}{\tau_1}\big)^\beta}\Big\}\big\|\nabla F(w_t)\big\|^2$$

$$\stackrel{(iv)}{=} \min\Big\{\gamma, \frac{1}{4L_1\big(5\|\nabla F(w_t)\|\big)^\beta}, \frac{1}{4L_1\big(\frac{3\tau_2}{\tau_1}\big)^\beta}\Big\}\big\|\nabla F(w_t)\big\|^2$$

$$\stackrel{(v)}{=} \min\Big\{\gamma, \frac{1}{4L_1\big(5\|\nabla F(w_t)\|\big)^\beta}\Big\}\big\|\nabla F(w_t)\big\|^2$$

$$\stackrel{(vi)}{\geq} \min\Big\{\gamma\big\|\nabla F(w_t)\big\|^2, \frac{\big\|\nabla F(w_t)\big\|^{2-\beta}}{20L_1}\Big\}, \tag{40}$$

*where (i) expands the expression of $\frac{1}{h_t^\beta}$; (ii) utilizes the equation 45 to upper bounds $\|\nabla f_{\xi_{B'}}(w_t)\|$ by $(\tau_1/\sqrt{16\tau_1^2}+1)\|\nabla F(w_t)\|+\tau_2/\sqrt{16\tau_1^2}$ by setting $B'=16\tau_1^2$; (iii) utilizes the fact $\frac{1}{a+b}\geq\min\{\frac{1}{2a},\frac{1}{2b}\}$ where $a=\frac{5}{2}\|\nabla F(w_t)\|$, $b=\frac{3\tau_2}{\tau_1}$; (iv) puts $\gamma$ inside the minimum operator. From step size rule, $\gamma\leq \frac{1}{8L_1(3\tau_2/\tau_1)^\beta}$, we can directly delete the third term $\frac{1}{4L_1(3\tau_2/\tau_1)^\beta}$, which reduces expressions in (v); (vi) further replaces $5^\beta$ by 5 in denominator.*

*Since now equation 40 has no randomness induced from $\xi_{B'}$. Summing the above descent lemma from 0 to $T-1$, we have*

$$\sum_{t=0}^{T-1}\mathbb{E}_{w_t}\Big[ \min\Big\{\frac{\gamma}{4}\big\|\nabla F(w_t)\big\|^2, \frac{\big\|\nabla F(w_t)\big\|^{2-\beta}}{80L_1}\Big\}\Big]$$

$$\leq \sum_{t=0}^{T-1}\mathbb{E}_{w_t}\Big[\frac{\gamma}{4h_t^\beta}\big\|\nabla F(w_t)\big\|^2\Big]$$

$$\leq \sum_{i=1}^{T}\mathbb{E}_{w_t}\Big[\mathbb{E}_{\xi_B}\big[F(w_t)-F(w_{t+1})|w_t\big]\Big] + T\frac{1}{2}\gamma^2(L_0+L_1)(1+\frac{\tau_2^2}{\tau_1^2})^2. \tag{41}$$

By step size rule, from where we know $\gamma \le \frac{1}{\sqrt{T}}$, we have

$$\sum_{t=0}^{T-1} \mathbb{E}_{w_t}\Big[\min\Big\{\frac{\gamma}{4}\big\|\nabla F(w_t)\big\|^2, \frac{\big\|\nabla F(w_t)\big\|^{2-\beta}}{80L_1}\Big\}\Big] \le F(w_0) - F^* + \frac{1}{2}(L_0 + L_1)(1 + \frac{\tau_2^2}{\tau_1^2})^2. \quad (42)$$

Denote $K = \{t | t \in [T] \text{ such that } \gamma\big\|\nabla F(w_t)\big\|^2 \le \frac{\|\nabla F(w_t)\|^{2-\beta}}{20L_1}\}$, then above descent lemma can be reduced to

$$\sum_{t\in K} \mathbb{E}_{w_t}\Big[\frac{\gamma}{4}\big\|\nabla F(w_t)\big\|^2\Big] \le F(w_0) - F^* + \frac{1}{2}(L_0 + L_1)(1 + \frac{\tau_2^2}{\tau_1^2})^2,$$

and

$$\sum_{t\in K^c} \mathbb{E}_{w_t}\Big[\frac{\big\|\nabla F(w_t)\big\|^{2-\beta}}{80L_1}\Big] \le F(w_0) - F^* + \frac{1}{2}(L_0 + L_1)(1 + \frac{\tau_2^2}{\tau_1^2})^2.$$

Now denote RHS by $\Lambda = F(w_0) - F^* + \frac{1}{2}(L_0 + L_1)(1 + \frac{\tau_2^2}{\tau_1^2})^2$, then we have

$$\mathbb{E}_{w_t}\Big[\min_{t\in T}\big\|\nabla F(w_t)\big\|\Big] \le \mathbb{E}_{w_t}\Big[\min\Big\{\frac{1}{|K|}\sum_{t\in K}\big\|\nabla F(w_t)\big\|, \frac{1}{|K^c|}\sum_{t\in K^c}\big\|\nabla F(w_t)\big\|\Big\}\Big]$$

$$\overset{(i)}{\le} \mathbb{E}_{w_t}\Big[\min\Big\{\sqrt{\frac{1}{|K|}\sum_{i=1}^{|K|}\big\|\nabla F(w_t)\big\|^2}, \Big(\frac{1}{|K^c|}\sum_{i=1}^{|K^c|}\big\|\nabla F(w_t)\big\|^{2-\beta}\Big)^{\frac{1}{2-\beta}}\Big\}\Big]$$

$$\overset{(ii)}{\le} \max\Big\{\sqrt{(4\Lambda)\frac{4(L_0 + L_1) + \sqrt{T} + 8L_1(3\tau_2/\tau_1)^\beta}{T}}, \Big(\Lambda\frac{160L_1}{T}\Big)^{\frac{1}{2-\beta}}\Big\},$$

where (i) comes from the concavity $y^{\frac{1}{2}}$ and $y^{\frac{1}{2-\beta}}$ and inverse Jensen's inequality for concave function, and the last inequality follows from descent lemma as well as either $K > \frac{T}{2}$ or $K^c > \frac{T}{2}$. This implies, in order to find a point satisfies

$$Pr(\min_{t\in[T]}||\nabla F(w_t)|| \ge \epsilon) \le \frac{1}{2}.$$

By Markov inequality, we must have $\mathbb{E}_{w_t}[\min_{t\in[T]}||\nabla F(w_t)||] \le \frac{\epsilon}{2}$ when $T$ satisfies

$$T \ge \Lambda\max\Big\{\frac{256\Lambda}{\epsilon^4}, \frac{640L_1}{\epsilon^{2-\beta}}, \frac{64(L_0 + L_1) + 128L_1(3\tau_2/\tau_1)^\beta}{\epsilon^2}\Big\}. \quad (43)$$

# E  PROOF OF LEMMA 2

Before proving Lemma 2, let us proof the technical lemma to determine the upper bound of mini-batch stochastic gradient estimators given assumption 4

**Lemma 5** *For mini-batch stochastic gradient estimator satisfying assumption 4, denote $\delta_B(w)$ as the approximation error $\delta_B(w) = \frac{1}{B}\sum_{i=1}^{B}\nabla f_i(w) - \nabla F(w)$, we have the upper bound*

$$\big\|\delta_B(w)\big\| \le \frac{1}{\sqrt{B}}(\tau_1\big\|\nabla F(w)\big\| + \tau_2). \quad (44)$$

**Proof 5** *The proof follows from applying Jensen's inequality for L2 norm.*

$$\left\|\delta_B(w)\right\| = \left\|\frac{1}{B}\sum_{i=1}^{B}\delta_i(w)\right\|$$

$$= \frac{1}{B}\Big(\big\|\sum_{i=1}^{B}\delta_i(w)\big\|_2^2\Big)^{\frac{1}{2}}$$

$$\leq \frac{1}{B}\Big(\sum_{i=1}^{B}\big\|\delta_i(w)\big\|_2^2\Big)^{\frac{1}{2}}$$

$$\leq \frac{1}{B}\Big(\sum_{i=1}^{B}(\tau_1\|\nabla F(w)\| + \tau_2)^2\Big)^{\frac{1}{2}}$$

$$= \frac{1}{\sqrt{B}}(\tau_1\|\nabla F(w)\| + \tau_2).$$

*where the first inequality uses Jensen's inequality and convexity of squared L2-norm; the second inequality uses the assumption equation 14.*

This fact leads to

$$\left\|\nabla f_{\xi_B}(w)\right\| \leq (\frac{\tau_1}{\sqrt{B}} + 1)\left\|\nabla F(w)\right\| + \frac{\tau_2}{\sqrt{B}}. \tag{45}$$

Similarly, for variance of $\delta_B(w)$, we have the remark stated as following.

**Remark 1 (Variance bound for mini-batch $\delta_B(w)$)** *For variance of $\delta(w)$, following the same logic above, we have*

$$Var(\left\|\delta_B(w)\right\|) = \mathbb{E}\left\|\delta_B(w)\right\|^2$$

$$= \mathbb{E}\Big((\frac{1}{B}\sum_{i=1}^{B}\delta_i(w))^T(\frac{1}{B}\sum_{i=1}^{B}\delta_j(w))\Big)$$

$$= \frac{1}{B^2}\mathbb{E}\Big[\sum_{i=1}^{B}\left\|\delta_i(w)\right\|^2\Big]$$

$$\leq \frac{1}{B}(\tau_1\left\|\nabla F(w)\right\| + \tau_2)^2$$

$$\leq \frac{1}{B}(2\tau_1^2\left\|\nabla F(w)\right\| + 2\tau_2^2),$$

*where the first inequality is due to equation 14. Thus, it is equivalent as*

$$\mathbb{E}_{\xi_B}\big[\left\|\nabla f_{\xi_B}(w)\right\|^2\big] \leq (\frac{2\tau_1^2}{B} + 1)\left\|\nabla F(w)\right\|^2 + \frac{2\tau_2^2}{B}. \tag{46}$$

**Lemma 2** *Let Assumptions 3 and 4 hold. Consider the mini-batch stochastic gradient $\nabla f_{\xi_B}$ with batch size $B = 16\tau_1^2$, then for all $w \in \mathbf{R}^d$ we have*

$$\left\|\nabla f_{\xi_B}(w)\right\| \geq \frac{1}{2}\left\|\nabla F(w)\right\| - \frac{\tau_2}{2\tau_1}. \tag{15}$$

**Proof 6 (Proof of Lemma 2)** *When $\left\|\nabla F(w)\right\|$ is large such that $\left\|\nabla F(w)\right\| \geq \frac{\tau_2}{\tau_1}$, then equation 44 indicates*

$$\left\|\delta_B(w)\right\| \leq \frac{2\tau_1 \left\|\nabla F(w)\right\|}{\sqrt{B}}.$$

*In this case, if we choose $B = 16\tau_1^2$, we have $\left\|\delta_B(w)\right\| \leq \frac{1}{2}\left\|\nabla F(w)\right\|$. Since in this case, we assume, $\left\|\nabla F(w)\right\| \geq \frac{\tau_2}{\tau_1} \geq \frac{\tau_2}{2\tau_1}$, we have*

$$\left| \left\|\nabla f_{\xi_B}(w_t)\right\| - \left\|\nabla F(w)\right\| \right| \leq \frac{1}{2}\left\|F(w)\right\|,$$

*which is equivalent as*

$$\left\|\nabla f_{\xi_B}(w)\right\| - \left\|\nabla F(w)\right\| \geq -\frac{1}{2}\left\|\nabla F(w)\right\|.$$

*And this fact leads to*

$$\frac{1}{2}\left\|\nabla F(w)\right\| \leq \left\|\nabla f_{\xi_B}(w)\right\| \leq \left\|\nabla f_{\xi_B}(w)\right\| + \frac{\tau_2}{2\tau_1}.$$

*Re-organize the term gives us*

$$\left\|\nabla f_{\xi_B}(w)\right\| \geq \frac{1}{2}\left\|\nabla F(w)\right\| - \frac{\tau_2}{2\tau_1}.$$

*Similarly, when $\left\|\nabla F(w)\right\| \leq \frac{\tau_2}{\tau_1}$, for single stochastic sample, by assumption 4, we have $\left\|\delta(w)\right\| \leq 2\tau_2$, from equation 44, for mini-batch stochastic gradient estimator, we have*

$$\left\|\delta_B(w)\right\| \leq \frac{2\tau_2}{\sqrt{B}}.$$

*By setting $B = 16\tau_1^2$, we have $\left\|\delta_B(w)\right\| \leq \frac{\tau_2}{2\tau_1}$. This fact leads to*

$$\left| \left\|\nabla f_{\xi_B}(w)\right\| - \left\|\nabla F(w)\right\| \right| \leq \frac{\tau_2}{2\tau_1},$$

*which is equivalent as*

$$\left\|\nabla f_{\xi_B}(w)\right\| - \left\|\nabla F(w)\right\| \geq -\frac{\tau_2}{2\tau_1}.$$

*Thus, we have*

$$\frac{1}{2}\left\|\nabla F(w)\right\| \leq \left\|\nabla F(w)\right\| = \left\|\nabla F(w)\right\| + \frac{\tau_2}{2\tau_1} - \frac{\tau_2}{2\tau_1} \leq \left\|\nabla f_{\xi_B}(w)\right\| + \frac{\tau_2}{2\tau_1},$$

*which leads to*

$$\left\|\nabla f_{\xi_B}(w)\right\| \geq \frac{1}{2}\left\|\nabla F(w)\right\| - \frac{\tau_2}{2\tau_1}.$$

*Combine above, we conclude by choosing $B = 16\tau_1^2$, we always have*

$$\left\|\nabla f_{\xi_B}(w)\right\| \geq \frac{1}{2}\left\|\nabla F(w)\right\| - \frac{\tau_2}{2\tau_1}. \tag{47}$$

## F   LEMMA 6 AND PROOF

**Lemma 6** *For the "Term 1" defined in equation 38, we have upper bound*

$$\frac{1}{2}\gamma^2 \frac{\left(L_0 + L_1 \left\|\nabla F(w_t)\right\|^\alpha\right)}{h_t^{2\beta}} \frac{\tau_2^2}{\tau_1^2} \leq \frac{1}{2}\gamma^2(L_0 + L_1)(1 + \frac{\tau_2^2}{\tau_1^2})^2 + \frac{\gamma}{4h_t^\beta}\left\|\nabla F(w_t)\right\|^2. \tag{48}$$

**Proof 7** *When $\left\|\nabla F(w_t)\right\| \leq \sqrt{1 + \tau_2^2/\tau_1^2}$, we have $\left\|\nabla F(w_t)\right\|^\alpha \leq (1 + \tau_2^2/\tau_1^2)^{\frac{\alpha}{2}}$ for any $\alpha > 0$. Since $(1 + \tau_2^2/\tau_1^2) > 1$ and $(1 + \tau_2^2/\tau_1^2) > \tau_2^2/\tau_1^2$. These facts lead to*

$$\frac{1}{2}\gamma^2 \frac{\left(L_0 + L_1 \left\|\nabla F(w_t)\right\|^\alpha\right)}{h_t^{2\beta}} \frac{\tau_2^2}{\tau_1^2}$$

$$\leq \frac{1}{2}\gamma^2(1 + \frac{\tau_2^2}{\tau_1^2})^{\frac{\alpha}{2}} \frac{(L_0 + L_1)}{h_t^{2\beta}} \frac{\tau_2^2}{\tau_1^2}$$

$$\leq \frac{1}{2}\gamma^2(L_0 + L_1)(1 + \frac{\tau_2^2}{\tau_1^2})^{\frac{\alpha}{2}}(1 + \frac{\tau_2^2}{\tau_1^2})$$

$$\leq \frac{1}{2}\gamma^2(L_0 + L_1)(1 + \frac{\tau_2^2}{\tau_1^2})^2, \tag{49}$$

*where the first inequality comes from $\|\nabla F(w_t)\| \leq \sqrt{1 + \tau_2^2/\tau_1^2}$ and $1 + \tau_2^2/\tau_1^2 > 1$; the second inequality comes from the fact that $\frac{1}{h_t} < 1$, so does $\frac{1}{h_t^{2\beta}}$, and upper bound $\tau_2^2/\tau_1^2$ by $(1 + \tau_2^2/\tau_1^2)$; the last inequality uses the fact that $0 \leq \alpha \leq 1$ and $(1 + \tau_2^2/\tau_1^2)^{1+\alpha/2} \leq (1 + \tau_2^2/\tau_1^2)^2$.*

*When $\left\|\nabla F(w_t)\right\| \geq \sqrt{1 + \tau_2^2/\tau_1^2}$, we must have $\left\|\nabla F(w_t)\right\|^2 \geq (1 + \tau_2^2/\tau_1^2) \geq \tau_2^2/\tau_1^2$ for any $\alpha > 0$, Thus, we conclude*

$$\frac{\gamma^2}{2} \frac{L_1\left\|\nabla F(w_t)\right\|^\alpha}{h_t^{2\beta}} \cdot \frac{\tau_2^2}{\tau_1^2}$$

$$= \frac{\gamma^2}{2} \frac{L_1\left\|\nabla F(w_t)\right\|^\alpha}{h_t^\beta} \frac{1}{h_t^\beta} \cdot \frac{\tau_2^2}{\tau_1^2}$$

$$= \frac{\gamma^2}{2h_t^\beta} \frac{L_1\left\|\nabla F(w_t)\right\|^\alpha}{h_t^\beta} \cdot \frac{\tau_2^2}{\tau_1^2}$$

$$\overset{(i)}{\leq} \frac{\gamma^2}{2h_t^\beta} \frac{L_1\left\|\nabla F(w_t)\right\|^\alpha}{h_t^\beta} \cdot \left\|\nabla F(w_t)\right\|^2$$

$$\overset{(ii)}{\leq} \frac{\gamma^2}{2h_t^\beta} \frac{L_1\left\|\nabla F(w_t)\right\|^\alpha}{(4L_1\gamma)(2\left\|\nabla f_{\xi_{B'}}(w_t)\right\| + \frac{\tau_2}{\tau_1})^\beta} \cdot \left\|\nabla F(w_t)\right\|^2$$

$$\overset{(iii)}{\leq} \frac{\gamma^2}{2h_t^\beta} \frac{L_1\left\|\nabla F(w_t)\right\|^\alpha}{(4L_1\gamma)\left\|\nabla F(w_t)\right\|^\beta} \left\|\nabla F(w_t)\right\|^2$$

$$= \frac{\gamma^2}{2h_t^\beta} \frac{L_1}{4L_1\gamma\left\|\nabla F(w_t)\right\|^{\beta-\alpha}} \left\|\nabla F(w_t)\right\|^2$$

$$\overset{(iv)}{\leq} \frac{\gamma^2}{2h_t^\beta} \frac{L_1}{4L_1\gamma} \left\|\nabla F(w_t)\right\|^2$$

$$= \frac{\gamma}{8h_t^{\beta}} \left\| \nabla F(w_t) \right\|^2, \tag{50}$$

where (i) comes from the fact that $||\nabla F(w_t)|| \geq \sqrt{1 + \tau_2^2/\tau_1^2}$; (ii) comes from the fact $\frac{1}{h_t^{\beta}} \leq \frac{1}{4L_1\gamma(2||\nabla f_{\xi_{B'}}(w_t)|| + \tau_2/\tau_1)^{\beta}}$; (iii) comes to the fact $||\nabla F(w_t)|| \leq 2||\nabla f_{\xi_{B'}}(w_t)|| + \tau_2/\tau_1$; (iv) comes from the fact that now $||\nabla F(w_t)|| > 1$, thus $\frac{1}{||\nabla F(w_t)||^{\beta-\alpha}} < 1$. Similarly, when $\left\| \nabla F(w_t) \right\| \geq \sqrt{1 + \tau_2^2/\tau_1^2}$, we can upper bound $\frac{1}{2}\gamma^2 L_0 \frac{\tau_2^2}{\tau_1^2}$ by

$$\frac{1}{2h_t^{2\beta}}\gamma^2 L_0 \cdot \frac{\tau_2^2}{\tau_1^2}$$

$$\leq \frac{\gamma}{2h_t^{\beta}}\gamma L_0 \left\| \nabla F(w_t) \right\|^2$$

$$\leq \frac{\gamma}{8h_t^{\beta}} \left\| \nabla F(w_t) \right\|^2, \tag{51}$$

where the first inequality uses the fact $\left\| \nabla F(w_t) \right\| \geq \sqrt{(1 + \tau_2^2/\tau_1^2)}$ and $\frac{1}{h_t^{\beta}} \leq 1$, second inequality uses the fact $\gamma L_0 \leq \frac{1}{4}$.

Combine equation 49, equation 50, equation 51 give us desired result.

# G    CONVERGENCE RESULT OF I-NSGD UNDER GENERALIZED PŁ CONDITION

**Theorem 3 (Convergence of I-NSGD under generalized PŁ condition)** *Let Assumptions 1, 3, 4 hold. For I-NSGD algorithm, choose $\epsilon$ to make $\gamma$ satisfying $\gamma = \frac{L_1(2\mu\epsilon)^{(4-2\beta)/\rho}}{16(L_0+2L_1)^2(1+\tau_2^2/\tau_1^2)^4} \leq \min\{\frac{1}{4L_0}, \frac{1}{8L_1(3\tau_2/\tau_1)^{\beta}}, \frac{1}{(20)^{2/3}L_1}\}$, $\alpha \leq \beta \leq 1$, batch size $B = 2\tau_1^2$, $B' = 16\tau_1^2$ and denote $\Delta_t = F(w_t) - F^*$. Depending on the choice of $\rho + \beta$, the following statements for I-NSGD's convergence under generalized PŁ condition hold.*

- *If $0 < \rho < 2 - \beta$, we have*

$$\mathbb{E}\left[\Delta_t\right] = \mathcal{O}\left(\left(\frac{\rho}{(2-\beta-\rho)\gamma^{3/2}t}\right)^{\frac{\rho}{2-\rho-\beta}}\right). \tag{52}$$

  *I-NSGD converges with $T = \mathcal{O}\left(\left(\frac{1}{\epsilon}\right)^{\frac{3(2-\beta)}{\rho}}\right)$ to attain $\mathbb{E}[\Delta_t] \leq \epsilon$.*

- *If $\rho = 2 - \beta$, and choose $\gamma \leq (4/\mu\sqrt{L_1})^{2/3}$, then we have*

$$\mathbb{E}\left[\Delta_{t+1}\right] = \mathcal{O}\left((1 - \frac{\mu\sqrt{L_1}\gamma^{3/2}}{4})^t\right). \tag{53}$$

  *I-NSGD converges with $\mathcal{O}\left(\left(\frac{1}{\epsilon}\right)^3 \log(\frac{1}{\epsilon})\right)$ to attain $\mathbb{E}[\Delta_t] \leq \epsilon$.*

- *If $\rho > 2 - \beta$, there exists $T_0 \in \mathbf{N}$ such that for all $t \geq T_0$, we have recursion*

$$\mathbb{E}\left[\Delta_t\right] = \mathcal{O}\left(\mathbb{E}\left[\left(\frac{\Delta_{T_0}}{\gamma^{3\rho/2(\rho+\beta-2)}}\right)^{\frac{\rho}{2-\beta}t}\right]\right). \tag{54}$$

  *After $T_0$, I-NSGD converges with $T = \Omega\left(\log\left(\left(\frac{1}{\epsilon}\right)^{3(2-\beta)/(\rho+\beta-2)}\right)\right)$ to attain $\mathbb{E}[\Delta_t] \leq \epsilon$.*

**Proof 8** *Similarly, starting From descent lemma and taking expectation over $\xi_B$ and conditioned over $w_t$, we have*

$$\mathbb{E}_{\xi_B}\big[F(w_{t+1}) - F(w_t)|w_t\big] \leq \frac{-\gamma\mathbb{E}_{\xi_B}[\|\nabla F(w_t)\|^2|w_t]}{h_t^\beta}$$

$$+ \frac{1}{2}\gamma^2(L_0 + L_1\|\nabla F(w_t)\|^\alpha)\frac{\mathbb{E}_{\xi_B}\big[\|\nabla f_{\xi_B}(w_t)\|^2|w_t\big]}{h_t^{2\beta}}.$$

*Let $B = 2\tau_1^2$, we must have $\frac{1}{B} = \frac{1}{2\tau_1^2}$, Put equation 33 into above descent lemma, we have*

$$\mathbb{E}_{\xi_B}\big[F(w_{t+1}) - F(w_t)|w_t\big]$$

$$\leq -\gamma\frac{\|\nabla F(w_t)\|^2}{h_t^\beta} + \frac{1}{2}\gamma^2(L_0 + L_1\|\nabla F(w_t)\|^\alpha)\frac{2\|\nabla F(w_t)\|^2 + \tau_2^2/\tau_1^2}{h_t^{2\beta}}$$

$$= \frac{\gamma}{h_t^\beta}\big(-1 + \gamma\frac{L_0 + L_1\|\nabla F(w_t)\|^\alpha}{h_t^\beta}\big)\|\nabla F(w_t)\|^2 + \frac{\gamma^2}{2h_t^{2\beta}}(L_0 + L_1\|\nabla F(w_t)\|^\alpha)\frac{\tau_2^2}{\tau_1^2}. \qquad (55)$$

*Similarly as above, since $(20)^{\frac{2}{3}} \approx 7.37$, we still have $\gamma \leq \frac{1}{4L_1}$, $\frac{\gamma L_0}{h_t^\beta} \leq \frac{1}{4}$, and $\frac{L_1\gamma\|\nabla F(w_t)\|^\alpha}{h_t^\beta} \leq \frac{1}{4}$ holds. We omit the proof for these upper bounds, which are the same as proof for Theorem 2*

*Additionally, for $\frac{\gamma^2}{2h_t^{2\beta}}(L_0 + L_1\|\nabla F(w)\|^\alpha)\frac{\tau_2^2}{\tau_1^2}$, lemma 6 still holds. These arguments leads to the same descent inequality as above*

$$\frac{\gamma}{4h_t^\beta}\|\nabla F(w_t)\|^2 \leq \mathbb{E}_{\xi_B}\big[F(w_{t+1}) - F(w_t)|w_t\big] + \frac{1}{2}(L_0 + L_1)\gamma^2(1 + \frac{\tau_2^2}{\tau_1^2})^2.$$

*To create proper expression to induce generalized PŁ condition, by leveraging $\frac{1}{(a+b)^\beta} \geq \min\{\frac{1}{(2a)^\beta}, \frac{1}{(2b)^\beta}\}$ we have*

$$\gamma\frac{\|\nabla F(w_t)\|^2}{h_t^\beta} = \gamma\min\Big\{1, \frac{1}{(4L_1\gamma)(2\|\nabla f_{\xi_{B'}}(w_t)\| + \frac{\tau_2}{\tau_1})^\beta}\Big\}\|\nabla F(w_t)\|^2$$

$$\overset{(i)}{\geq} \gamma\min\Big\{1, \frac{1}{(4L_1\gamma)(\frac{5}{2}\|\nabla F(w_t)\| + \frac{3\tau_2}{2\tau_1})^\beta}\Big\}\|\nabla F(w_t)\|^2$$

$$\overset{(ii)}{\geq} \gamma\min\Big\{1, \frac{1}{(4L_1\gamma)(5\|\nabla F(w_t)\|)^\beta}, \frac{1}{(4L_1\gamma)(\frac{3\tau_2}{\tau_1})^\beta}\Big\}\|\nabla F(w_t)\|^2$$

$$\overset{(iii)}{\geq} \min\Big\{\gamma, \frac{1}{20L_1\|\nabla F(w_t)\|^\beta}\Big\}\|\nabla F(w_t)\|^2$$

$$\overset{(iv)}{\geq} \min\Big\{\gamma(L_1\gamma)^{\frac{\beta}{2}}\|\nabla F(w_t)\|^{2-\beta} - L_1\gamma^2, \frac{1}{20L_1}\|\nabla F(w_t)\|^{2-\beta}\Big\}$$

$$\overset{(v)}{\geq} \min\Big\{\gamma(L_1\gamma)^{\frac{\beta}{2}}\|\nabla F(w_t)\|^{2-\beta}, \frac{1}{20L_1}\|\nabla F(w_t)\|^{2-\beta}\Big\} - L_1\gamma^2$$

$$\overset{(vi)}{\geq} L_1^{\frac{1}{2}}\gamma^{\frac{3}{2}}\|\nabla F(w_t)\|^{2-\beta} - L_1\gamma^2$$

$$\overset{(vii)}{\geq} L_1^{\frac{1}{2}}\gamma^{\frac{3}{2}}\big(2\mu(F(w_t) - F^*)\big)^{\frac{2-\beta}{\rho}} - L_1\gamma^2, \qquad (56)$$

where (i) comes from equation 44 by setting $B' = 16\tau_1^2$; (ii) comes from the fact $\min\{\frac{1}{(a+b)^\beta}\} \geq \min\{\frac{1}{(2a)^\beta}, \frac{1}{(2b)^\beta}\}$; (iii) comes from the fact that $\gamma \leq \frac{1}{8L_1(3\tau_2/\tau_1)^\beta}$; (iv) utilizes Lemma 3 by setting $C = (L_1\gamma)^{\beta/2}$, $w = 2 - \beta$ and $w' = 2$, $\Delta = \beta$, which leads to $||\nabla F(w_t)||^2 \geq (L_1\gamma)^{\beta/2}||\nabla F(w_t)||^{2-\beta} - L_1\gamma$ ; (v) is due to the fact $\min\{a - c, b\} \geq \min\{a, b\} - c$ holds for $a, b, c > 0$; (vi) is due to $\gamma L_1 < 1$, thus we have $\gamma(L_1\gamma)^{\frac{\beta}{2}} \geq L_1^{\frac{1}{2}}(\gamma)^{\frac{3}{2}}$ and $\gamma \leq \frac{1}{(20)^{2/3}L_1}$; (vii) is from the assumption 5 generalized PŁ condition, i.e., $||\nabla F(w_t)||^{2-\beta} \geq (2\mu(F(w_t) - F^*))^{\frac{2-\beta}{\rho}}$. Put this fact into above descent lemma and re-organize it, we have

$$\Delta_t \leq \Delta_t - \frac{L_1^{\frac{1}{2}}\gamma^{3/2}}{4}(2\mu\Delta_t)^{\frac{2-\beta}{\rho}} + \frac{1}{2}(L_0 + L_1)\gamma^2(1 + \frac{\tau_2^2}{\tau_1^2})^2 + L_1\gamma^2$$

$$\leq \Delta_t - \frac{L_1^{\frac{1}{2}}\gamma^{3/2}}{4}(2\mu\Delta_t)^{\frac{2-\beta}{\rho}} + \frac{1}{2}(L_0 + 2L_1)\gamma^2(1 + \frac{\tau_2^2}{\tau_1^2})^2. \tag{57}$$

By defining $\gamma = \frac{L_1(2\mu\epsilon)^{\frac{4-2\beta}{\rho}}}{16(L_0+2L_1)^2(1+\tau_2^2/\tau_1^2)^4}$ into descent inequality, we have

$$\Delta_{t+1} \leq \Delta_t - \frac{L_1^{\frac{1}{2}}\gamma^{3/2}}{4}(2\mu\Delta_t)^{\frac{2-\beta}{\rho}} + \frac{L_1^{\frac{1}{2}}\gamma^{3/2}}{8}(2\mu\epsilon)^{\frac{2-\beta}{\rho}}. \tag{58}$$

Since convergence rate of equation 58 is depending on $\rho + \beta$, we divide it into 3 cases for analysis. The analysis in the rest is similar compared with Theorem 1, we highlight the key steps to yield the convergence rate.

**Case I, when** $0 < \rho < 2 - \beta$ This equivalent as $\frac{2-\beta}{\rho} > 1$. Taking expectation over equation 58, for any $t \leq T = \inf\{t | \Delta_t \leq \epsilon\}$ we have

$$\mathbb{E}[\Delta_{t+1}] \leq \mathbb{E}\left[\Delta_t - \frac{L_1^{\frac{1}{2}}\gamma^{3/2}}{4}(2\mu\Delta_t)^{\frac{2-\beta}{\rho}}\right] + \frac{L_1^{\frac{1}{2}}\gamma^{3/2}}{8}(2\mu\epsilon)^{\frac{2-\beta}{\rho}}$$

$$\leq \mathbb{E}\left[\Delta_t - \frac{L_1^{\frac{1}{2}}\gamma^{3/2}}{4}(2\mu\Delta_t)^{\frac{2-\beta}{\rho}}\right] + \frac{L_1^{\frac{1}{2}}\gamma^{3/2}}{8}(2\mu\mathbb{E}[\Delta_t])^{\frac{2-\beta}{\rho}}$$

$$\leq \mathbb{E}[\Delta_t] - \frac{L_1^{\frac{1}{2}}\gamma^{3/2}}{4}(2\mu\mathbb{E}[\Delta_t])^{\frac{2-\beta}{\rho}} + \frac{L_1^{\frac{1}{2}}\gamma^{3/2}}{8}(2\mu\mathbb{E}[\Delta_t])^{\frac{2-\beta}{\rho}}$$

$$= \mathbb{E}[\Delta_t] - \frac{L_1^{\frac{1}{2}}\gamma^{3/2}}{8}(2\mu\mathbb{E}[\Delta_t])^{\frac{2-\beta}{\rho}}, \tag{59}$$

where the second inequality is due to the argument for any $t \leq T$, we have $\mathbb{E}[\Delta_t] \geq \epsilon$ holds; third inequality is due to Jensen's inequality since $2 - \beta/\rho > 1$.

After constructing equation 59, the following proof is same as proof in Theorem equation 1, we present key steps to determinie convergence rate. For simplicity of notation, now we denote $\theta = \frac{2-\beta}{\rho}$ and $\overline{\Delta}_t = \mathbb{E}[\Delta_t]$. Since $\theta > 1$, we have following inequalities hold

$$\overline{\Delta}_{t+1} \leq \overline{\Delta}_t$$
$$\overline{\Delta}_{t+1}^\theta \leq \overline{\Delta}_t^\theta$$
$$(\overline{\Delta}_{t+1})^{-\theta} \geq (\overline{\Delta}_t)^{-\theta}. \tag{60}$$

Now define an auxiliary function $\Phi(t) = \frac{1}{\theta-1}t^{1-\theta}$. Its derivative can be computed via $\Phi'(t) = -t^{-\theta}$ Divide the last inequality at equation 60 into two different cases.

When $(\overline{\Delta}_t)^{-\theta} \leq (\overline{\Delta}_{t+1})^{-\theta} \leq 2(\overline{\Delta}_t)^{-\theta}$, we have

$$\Phi(\overline{\Delta}_{t+1}) - \Phi(\overline{\Delta}_t) = \int_{\overline{\Delta}_t}^{\overline{\Delta}_{t+1}} \Phi'(t)\mathrm{d}t = \int_{\overline{\Delta}_{t+1}}^{\overline{\Delta}_t} t^{-\theta}\mathrm{d}t$$

$$\geq (\overline{\Delta}_t - \overline{\Delta}_{t+1})\overline{\Delta}_t^{-\theta}$$

$$\geq (\overline{\Delta}_t - \overline{\Delta}_{t+1})\frac{\overline{\Delta}_{t+1}^{-\theta}}{2}$$

$$\geq \frac{L_1^{\frac{1}{2}}\gamma^{\frac{3}{2}}(2\mu)^\theta}{8}\overline{\Delta}_t^{-\theta}\frac{\overline{\Delta}_{t+1}^{-\theta}}{2}$$

$$\geq \frac{L_1^{\frac{1}{2}}\gamma^{\frac{3}{2}}(2\mu)^\theta}{8}\overline{\Delta}_{t+1}^{-\theta}\frac{\overline{\Delta}_{t+1}^{-\theta}}{2} = \frac{L_1^{\frac{1}{2}}\gamma^{\frac{3}{2}}(2\mu)^\theta}{16},$$

where the first inequality is using mean value theorem; The second inequality uses fact $\overline{\Delta}_{t+1}^{-\theta} \leq 2\overline{\Delta}_t^{-\theta}$. The third inequality is due to the recursion $\overline{\Delta}_t - \overline{\Delta}_{t+1} \geq \frac{L_1^{\frac{1}{2}}\gamma^{\frac{3}{2}}(2\mu)^\theta}{8}\overline{\Delta}_t^{\theta}$. The last inequality uses the fact that $\overline{\Delta}_t^{\theta} > \overline{\Delta}_{t+1}^{\theta}$ for all $\theta > 0$.

When $\overline{\Delta}_{t+1}^{-\theta} > 2\overline{\Delta}_t^{-\theta}$, it holds that $\overline{\Delta}_{t+1}^{1-\theta} = (\overline{\Delta}_{t+1}^{-\theta})^{\frac{1-\theta}{-\theta}} > 2^{\frac{1-\theta}{-\theta}}\overline{\Delta}_t^{1-\theta}$. Then, we have

$$\Phi(\overline{\Delta}_{t+1}) - \Phi(\overline{\Delta}_t) = \frac{1}{\theta-1}(\overline{\Delta}_{t+1}^{1-\theta} - \overline{\Delta}_t^{1-\theta})$$

$$\geq \frac{1}{\theta-1}\big((2)^{\frac{\theta-1}{\theta}} - 1\big)\overline{\Delta}_t^{1-\theta}$$

$$\geq \frac{1}{\theta-1}\big((2)^{\frac{\theta-1}{\theta}} - 1\big)\Delta_0^{1-\theta},$$

where the first inequality is from recursion $\overline{\Delta}_{t+1}^{1-\theta} = (\overline{\Delta}_{t+1}^{-\theta})^{\frac{1-\theta}{-\theta}} > 2^{\frac{1-\theta}{-\theta}}\overline{\Delta}_t^{1-\theta}$, the last inequality is due to the fact the sequence $\{\overline{\Delta}_t\}_{t=1}^{T}$ is non-increasing.

Now put the expression of $\theta$ in and denote

$$C = \min\Big\{\frac{L_1^{\frac{1}{2}}\gamma^{\frac{3}{2}}(2\mu)^{\frac{2-\beta}{\rho}}}{16}, \frac{\rho}{2-\beta-\rho}(2^{\frac{2-\beta-\rho}{2-\beta}} - 1)\Delta_0^{\frac{2-\beta-\rho}{\rho}}\Big\}.$$

We conclude for all $t$, we have

$$\Phi(\overline{\Delta}_t) \geq \sum_{i=0}^{t-1}\Phi(\overline{\Delta}_{i+1}) - \Phi(\overline{\Delta}_i) \geq Ct,$$

Thus, re-organize the inequality and taking expectation, we have

$$\mathbb{E}\big[\Delta_t\big] \leq \Big(\frac{\rho}{(2-\beta-\rho)Ct}\Big)^{\frac{\rho}{2-\beta-\rho}} = \mathcal{O}\Big(\Big(\frac{\rho}{(2-\beta-\rho)\gamma^{3/2}t}\Big)^{\frac{\rho}{2-\beta-\rho}}\Big).$$

When $C = \frac{\rho}{2-\beta-\rho}(2^{2-\beta-\rho/2-\beta} - 1)\Delta_0^{2-\beta-\rho/\rho}$, taking logarithm leads to $T = \mathcal{O}((\frac{1}{\epsilon})^{\frac{2-\rho-\beta}{\rho}})$.

When $C = \Theta(\epsilon^{3(2-\rho)/\beta})$, we have $T = \mathcal{O}\big((\frac{1}{\epsilon})^{\frac{2-\rho-\beta}{\rho}} + (\frac{1}{\epsilon})^{\frac{3(2-\beta)}{\rho}}\big)$. Since $\mathcal{O}\big((\frac{1}{\epsilon})^{\frac{3(2-\beta)}{\rho}}\big)$ dominates order-wisely, we conclude $T = \mathcal{O}\big((\frac{1}{\epsilon})^{\frac{3(2-\beta)}{\rho}}\big)$.

**Case II, when $\rho = 2 - \beta$** In this case, above recursion is equivalent as

$$\Delta_{t+1} \leq \Delta_t - \frac{\sqrt{L_1}\gamma^{3/2}}{4}(2\mu\Delta_t) + \frac{\sqrt{L_1}\gamma^{3/2}}{8}2\mu\epsilon.$$

*Taking expectation on both sides, we have*

$$\mathbb{E}\big[\Delta_t\big] \leq (1 - \frac{\sqrt{L_1}\gamma^{3/2}2\mu}{4})\mathbb{E}\big[\Delta_t\big] + \frac{\sqrt{L_1}\gamma^{3/2}}{8}2\mu\epsilon.$$

*By choosing the stopping criterion as $T = \inf\{t|\mathbb{E}[\Delta_t] \leq \epsilon\}$, we are guaranteed before $t \leq T$, $\mathbb{E}[\Delta_t] \geq \epsilon$. Utilizing this argument, we can further relax descent lemma by replacing $\Delta_t$ with $\epsilon$, which yields*

$$\mathbb{E}[\Delta_{t+1}] \leq (1 - \frac{2\sqrt{L_1}\gamma^{3/2}\mu}{8})\mathbb{E}[\Delta_t] = \mathcal{O}\Big((1 - \frac{\sqrt{L_1}\gamma^{3/2}\mu}{4})^t\Big).$$

*Thus, we have*

$$\mathbb{E}[\Delta_{t+1}] \leq (1 - \frac{2\mu\sqrt{L_1}\gamma^{3/2}}{8})^t\Delta_0 \leq \exp(-t \cdot \frac{2\mu\sqrt{L_1}\gamma^{3/2}}{8})\Delta_0,$$

*This gives us the sample complexity $T = \mathcal{O}\big((\frac{1}{\epsilon})^3 \log(\frac{1}{\epsilon})\big)$.*

**Case III, when $\rho > 2 - \beta$** This is equivalent as $\frac{2-\beta}{\rho} < 1$. Define stopping time $\tau = T \wedge \inf\{t|\Delta_t \leq \epsilon\}$, where $\wedge$ represents minimize operation, and $T = \inf\{t|\mathbb{E}_{\Delta_t} \leq \epsilon\}$. Thus, by the definition of $\tau$, we have $\tau \leq T$ and for all $t \leq \tau$, $\Delta_t \geq \epsilon$ holds. In this case, equation 58 reduces to

$$\Delta_{t+1} \leq \Delta_t - \frac{L_1^{\frac{1}{2}}\gamma^{3/2}}{4}(2\mu\Delta_t)^{\frac{2-\beta}{\rho}} + \frac{L_1^{\frac{1}{2}}\gamma^{3/2}}{8}(2\mu\epsilon)^{\frac{2-\beta}{\rho}} \leq \Delta_t - \frac{L_1^{\frac{1}{2}}\gamma^{3/2}}{8}(2\mu\Delta_t)^{\frac{2-\beta}{\rho}}.$$

*Since we have for all $t \leq \tau$, $\Delta_t$ dominates $\epsilon$. For simplicity, denote $C = \frac{\sqrt{L_1}(2\mu)^{\frac{4-2\beta}{\rho}}}{8}$ and $\omega = \frac{\rho}{2-\beta}$. The recursion now reduces to*

$$\Delta_t^{\frac{1}{\omega}}C\gamma^{\frac{3}{2}} + \Delta_{t+1} \leq \Delta_t. \tag{61}$$

*Use the same argument in Theorem 1, we can argue that equation 61 is guaranteed to converge to the level $\Delta_t \leq \epsilon$ before algorithm terminates. As long as $\Delta_t$ decreases, $\Delta_t^{\frac{2-\beta}{\rho}}$ will dominate $\Delta_t$ order-wisely when $\Delta_t$ is small enough. Instead, there exists $T_0 = \inf\{t \in \mathbf{N}|\Delta_t/C\gamma^{3/2} < 1\}$ such that equation 61 reduces to*

$$\Delta_t \leq (C\gamma^{\frac{3}{2}})^{-\omega}\Delta_t^\omega = (C\gamma^{3/2})^{\omega(1-\omega^{t-T_0})/\omega-1}\Delta_{T_0}^{\omega^{t-T_0}}$$

$$= (C\gamma^{3/2})^{\omega/\omega-1}\Big(\frac{\Delta_{T_0}}{(C\gamma^{3/2})^{\omega/\omega-1}}\Big)^{\omega^{t-T_0}}. \tag{62}$$

*Taking expectation on both sides over $\tau, T_0$, we have recursion*

$$\mathbb{E}\big[\Delta_\tau\big] \leq (C\gamma^{3/2})^{\omega/\omega-1}\mathbb{E}\Big[\Big(\frac{\Delta_{T_0}}{(C\gamma^{3/2})^{\omega/\omega-1}}\Big)^{\omega^{t-T_0}}\Big] = \mathcal{O}\Big(\mathbb{E}\Big[\frac{\Delta_{T_0}^{\omega^{t-T_0}}}{\big((\gamma^{3/2})^{\omega/\omega-1}\big)^{\omega^{t-T_0}}}\Big]\Big). \tag{63}$$

*By setting LHS of equation 62 equals to $\epsilon$, this recursion implies for any $\tau - T_0$, the iteration complexity is $\Omega\big(\log\big((\frac{1}{\epsilon})^{3(2-\beta)/(\rho+\beta-2)}\big)\big)$. It indicates after $T_0$, I-NSGD needs $T = \Omega\big(\log\big((\frac{1}{\epsilon})^{3(2-\beta)/(\rho+\beta-2)}\big)\big)$ to reach $\mathbb{E}[\Delta_t] \leq \epsilon$.*

## H  Experiment Details

### H.1  More details about DRO Experiments

In this experiment, we evaluate our algorithm by solving the nonconvex DRO problem in equation 19 using the life expectancy dataset. This dataset contains the life expectancy and its influencing factors for 2,413 individuals for regression analysis, where life expectancy serves as the target variable and the corresponding influencing factors as the features.

We process the data by filling missing values with the median of the respective variables, censoring and standardizing all variables, removing the categorical variables 'country' and 'status,' and adding standard Gaussian noise to the target variable to enhance model robustness. From this dataset, we select the first 2,313 samples $\{\mathbf{x}_i, y_i\}_{i=1}^{2313}$ as the training set, where $\mathbf{x}_i \in \mathbf{R}^{34}$ represents the features and $y_i \in \mathbf{R}$ represents the target. The loss function we used is regularized mean square loss, i.e., $\ell_\xi(\mathbf{w}) = \frac{1}{2}(y_\xi - \mathbf{x}_\xi^\top \mathbf{w})^2 + 0.1 \sum_{j=1}^{34} \ln(1 + |w^{(j)}|)$ with parameter $\mathbf{w} \in \mathbf{R}^{34}$, and initialize $\eta_0 = 0.1$ and $\mathbf{w}_0 \sim \mathcal{N}(0, I)$.

## I  Additional Experiments

To make a thorough study of our proposed I-NSGD. In this section, we design additional experiments to compare our methods with two additional baseline algorithms, NSGD with momentum (Cutkosky & Mehta, 2020; Hübler et al., 2024) and SPIDER algorithm(Fang et al., 2018).

### I.1  Additional Experiments for Phase Retrieval

For Phase Retrieval Problem, we generate the synthic data and initialization of $z$ as we have introduced in main paragraph.

During the experiments, we fix the moving average parameter, batch size as following. For acceleration methods, NSGD with momentum and SPIDER, we set its momentum moving average parameter as 0.2 and 0.4 respectively. For stochastic algorithms without usage of multiple mini-batches, i.e., SGD, NSGD, NSGD with momentum and Clipped SGD, we set their batch sizes as $|B| = 64$. For SPIDER, we set $|B| = 64$ and $|B'| = 3000$, where the algorithm will conduct a full-gradient computation after every 20 iterations. For I-NSGD, we set $|B| = 64$ and $|B'| = 4$.

In first experiment, we implement all the stochastic algorithms in original form described in previous literatures. we use fine-tuned learning rate for all algorithms, i.e., $\gamma = 5e - 5$ for SGD, $\gamma = 0.2$ for NSGD and NSGD with momentum, $\gamma = 0.3$ for SPIDER, $\gamma = 0.6$ for clipped SGD and $\gamma = 0.3$ for I-NSGD. We set the maximal gradient clipping constant as 45 and $\delta = 15$ for both Clipped SGD and I-NSGD. And we set normalization parameter $\beta = \frac{2}{3}$. Figure 4 (left) shows the comparison of objective value versus sample complexity. It can be observed that I-NSGD consistently converges faster than other algorithms.

In second experiment, we unify the normalization parameter of all the normalized methods, i.e., NSGD, NSGD with momentum, Clipped SGD, SPIDER and I-NSGD to have $\beta = \frac{2}{3}$. To make sure algorithm converges, we adjust the learning rate accordingly, i.e., $\gamma = 5e - 5$ for SGD, $\gamma = 0.03$ for NSGD and NSGD with momentum, $\gamma = 0.05$ for SPIDER, $\gamma = 0.3$ for both Clipped SGD and I-NSGD. To make a fair comparison between I-NSGD and clipped SGD, we decrease I-NSGD's independent batch size $|B'| = 4$ and keep others unchanged. Figure4 (right) shows the comparison of objective value versus sample complexity. It can be observed that, by adjusting $\beta = \frac{2}{3}$, the objective value optimized by all normalization method decreases much faster compared with Figure 4 (Left), this verifies the effectiveness of inducing normalization parameter $\beta$. Moreover, even though I-NSGD requires additional sampling at each iteration, the training loss optimized by it still decrease much faster than Clipped SGD, NSGD and SGD with momentum. This

indicates that, inducing independent sampling for clipping updates makes I-NSGD more adapted to the underlying generalized smooth non-convex geometry.

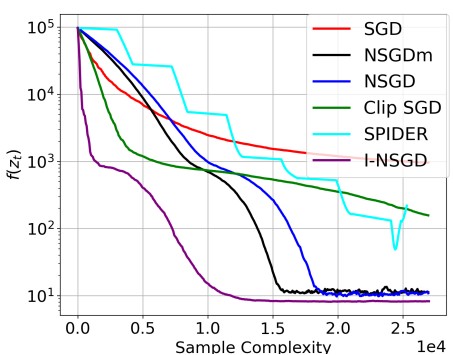 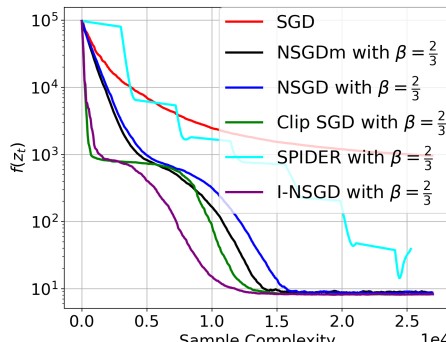

Figure 4: Additional Experiments for Phase Retrieval Problem

## I.2 ADDITIONAL EXPERIMENTS FOR DISTRIBUTIONALLY ROBUST OPTIMIZATION

For DRO problem, we reuse life Expectancy data(Arshi, 2017), pre-process the dataset and initialize model parameter in the same way as described in section H.

In first experiment, we implement all the stochastic algorithms in original form designed in previous literatures. The detailed algorithm settings are described as following. For moving average parameter used for acceleration method, we set it as 0.1 and 0.25 for NSGD with momentum and SPIDER respectively. For stochastic algorithms without usage of multiple mini-batches, i.e., SGD, NSGD, NSGD with momentum and Clipped SGD, we set their batch sizes as $|B| = 128$. For SPIDER, we set $|B| = 128$ and $|B'| = 2313$, where the algorithm will conduct a full-gradient computation after every 15 iterations. For I-NSGD, we set the batch size for two batch samples as $|B| = 128$ and $|B'| = 16$. We used fine-tuned learning rate for all algorithms, i.e., $\gamma = 4e-5$ for SGD, $\gamma = 5e-3$ for NSGD, NSGD with momentum and SPIDER, $\gamma = 0.25$ for clipped SGD and $\gamma = 0.15$ for I-NSGD. We set the maximal gradient clipping constant as 35 and $\delta = 25$ for both Clipped SGD and I-NSGD. And we set normalization parameter $\beta = \frac{2}{3}$. Figure 5(Left) shows the comparison of objective value versus sample complexity. It can be observed that objective value optimized by I-NSGD consistently converges faster than other baselines algorithms.

In second experiment, we further unify the normalization parameter of all the normalized methods, i.e., NSGD, NSGD with momentum, Clipped SGD, SPIDER and I-NSGD to have the same $\beta = \frac{2}{3}$. We adjust the learning rate, i.e., $\gamma = 4e-5$ for SGD, $\gamma = 5e-3$ for NSGD and NSGD with momentum, $\gamma = 3e-3$ for SPIDER, $\gamma = 0.14$ for both Clipped SGD and I-NSGD. To make a fair comparison with Clipped SGD, we also decrease I-NSGD's independent batch size to $|B'| = 8$ and keep others unchanged. Figure5 (Right) shows the comparison of objective value versus sample complexity. It can be observed that, by adjusting $\beta = \frac{2}{3}$, the objective value optimized by all normalization methods decreases much faster than Figure 5 (Left). This again verifies the effectiveness of inducing normalization parameter $\beta$. Even though our I-NSGD requires additional sampling at each iteration, the objective value optimized by I-NSGD decreases faster than SPIDER, NSGD and SGD with momentum and keeps the similar behavior as Clipped SGD. This indicates that, independent sampling for clipp-updates doesn't increase the sample complexity, which justifies I-NSGD framework's effectiveness when dealing with geometry characterized by generalized smooth condition.

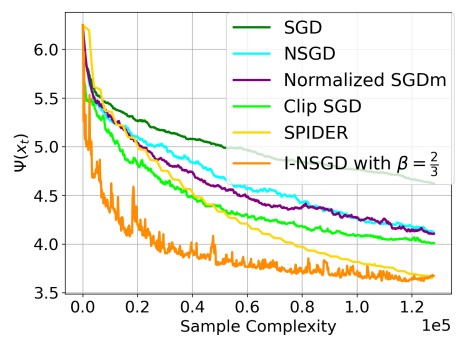 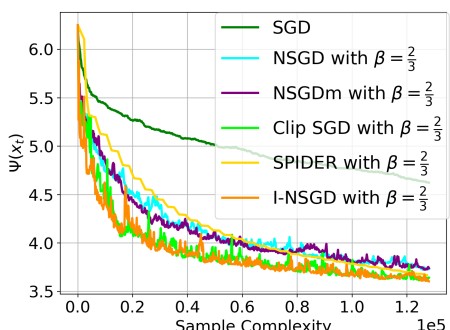

Figure 5: Additional Experiments for Distributionally Robust Optimization

## I.3 ADDITIONAL EXPERIMENTS ON TRAINING RESNET

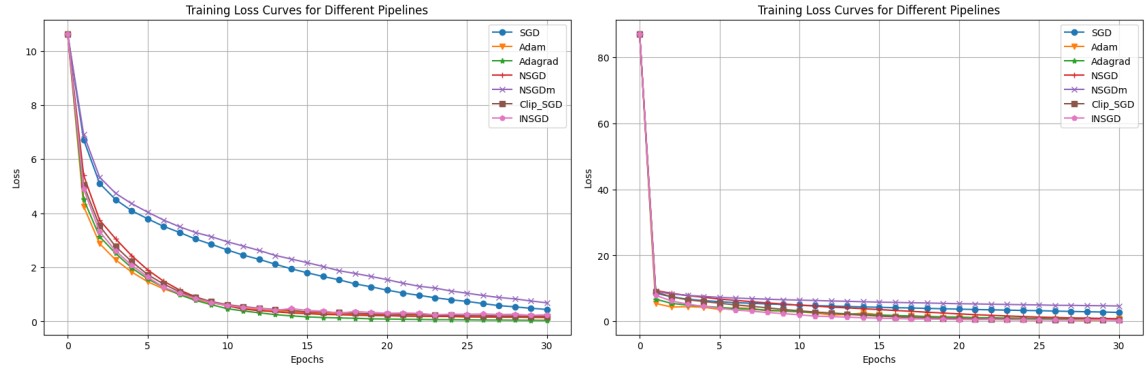

Figure 6: Additional Experiments for training ResNet18, ResNet50 on CIFAR10 Data

According to Zhang et al. (2019), generalized smooth has been observed to hold in deep neural networks. To further demonstrate the effectiveness of I-NSGD algorithm, we conduct experiments on training deep neural networks. To elaborate, we train ResNet18, ResNet50 (He et al., 2016) on CIFAR10 Dataset(Krizhevsky, 2009) from scratch. We resize images as $32 \times 32$ and normalize images with standard derivation equals to 0.5 on each dimension. At the beginning of each algorithm, we fix random seed and initialize model parameters using Kaiming initialization. We compare our algorithm with baseline methods, including SGD (Robbins & Monro, 1951), Adam (Kingma, 2014), Adagrad(Duchi et al., 2011a), NSGD, NSGD with momentum (Cutkosky & Mehta, 2020) and Clipp-SGD(Zhang et al., 2019).

To elaborate, we utilize pytorch built-in optimizer to implement training pipeline for SGD, Adam and Adagrad. Then we implement training pipeline for NSGD, NSGD with momentum, Clipp-SGD and I-NSGD. The normalization constant is computed through all model parameters at each iteration. The detailed algorithm settings are as following. For batch size, all algorithms use $B = 128$, and $B' = 32$ for I-NSGD. For moving average parameter, we use 0.9, 0.99 for Adam, and 0.25 for normalized SGD with momentum. For clipping threshold used in clipped SGD and I-NSGD, we set them as 2 and $\delta = 1e - 1$. The normalization power used for I-NSGD is $\beta = \frac{2}{3}$. We use fine-tuned learning rate for all algorithms, i.e., $\gamma = 1e - 3$ for SGD, Adam and Adagrad, $\gamma = 1e - 1$ for NSGD and NSGD with momentum, $\gamma = 2e - 1$ for Clipped SGD and I-NSGD. We trained ResNet18, ResNet50 on CIFAR10 dataset for 30 epochs and plot the training loss

in Figure 6. Figure 6 (left) shows the training loss of ResNet18, Figure6 (right) shows the training loss of ResNet50. As we can see from these figures, the (pink) loss curve optimized by I-NSGD indicates fast convergence rate complared several baselines, including SGD, NSGD NSGDm, Clip-SGD, which demonstrate the effectiveness of I-NSGD framework.