# OpenReview forum: "Independently-Normalized SGD for Generalized-Smooth Nonconvex Optimization"
_ICLR.cc/2025/Conference — Submitted to ICLR 2025_

### Official Review · Reviewer_AeAr · 2024-10-24

**Soundness:** 2
**Presentation:** 2
**Contribution:** 1
**Rating:** 3
**Confidence:** 4

**Summary:**

In this work the authors study the convergence of Normalized Gradient Descent and Normalized Stochastic Gradient Descent under generalized smoothness assumption and PL-condition.

**Strengths:**

This is a theoretical work on optimization. Assumptions are presented, theorem statements are written. The Appendix contains the proofs of the theorems.

**Weaknesses:**

I think that the contribution of this paper is not good enough for the standards of the ICLR conference.

NGD was already proposed and analyzed in the nonconvex setting in [1]. Extending their analysis to the PL-case does not seem as a strong contribution to me.

The second question in the paper is: "Can we develop a novel algorithm tailored for stochastic generalized-smooth optimization that
achieves fast convergence in practice while providing convergence guarantee under relaxed assumptions?"

There are papers with the analysis of NSGD with momentum [2], Adam [3,4], AdaGrad-Norm/AdaGrad [5,6]. All are tailored for the stochastic optimization in the generalized smooth setting and seem more advanced. All have convergence guarantees. Is the newly proposed I-NSGD faster in practice than any of them? As far as I can see, I-NSGD is compared to SGD, Clipped-SGD, NSGD in the experimental section only. The original paper [7], which introduced the generalized smoothness assumption, explains that this assumption is based on empirical findings from language model training and image classification tasks. Does I-NSGD perform well on them?

There is a recent paper [8], which contains a similar to NSGD method with smoothed clipping (Algorithm 1: (L0,L1)-GD). You mention this work in your paper. I can see that the authors develop the adaptive versions of the algorithm, the accelerated version. How does your theory compares to theirs? As far as I can tell from their contribution section, they improve upon many previous results.

[1] Ziyi Chen , Yi Zhou , Yingbin Liang , and Zhaosong Lu, Generalized-Smooth Nonconvex Optimization is As Efficient As Smooth Nonconvex Optimization.

[2] Hübler, F., Yang, J., Li, X., and He, N., Parameter-Agnostic Optimization under Relaxed Smoothness.

[3] Wang, B., Zhang, Y., Zhang, H., Meng, Q., Ma, Z.-M., Liu, T.-Y., and Chen, W., Provable adaptivity in adam.

[4] Li, H., Rakhlin, A., and Jadbabaie, A., Convergence of adam under relaxed assumptions.

[5] Faw, M., Rout, L., Caramanis, C., and Shakkottai, S., Beyond uniform smoothness: A stopped analysis of adaptive sgd.

[6] Wang, B., Zhang, H., Ma, Z., and Chen, W., Convergence of adagrad for non-convex objectives: Simple proofs and relaxed assumptions.

[7] Jingzhao Zhang, Tianxing He, Suvrit Sra, Ali Jadbabaie, Why gradient clipping accelerates training: A theoretical justification for adaptivity

[8] Eduard Gorbunov, Nazarii Tupitsa, Sayantan Choudhury, Alen Aliev, Peter Richtárik, Samuel Horváth, Martin Takáč, Methods for Convex (L0,L1)-Smooth Optimization: Clipping, Acceleration, and Adaptivity

**Questions:**

There is a standard sanity check procedure in optimization. The field is connected to physics (take the heavy-ball method of Polyak, for instance). And different quantities in physics have different units of measure. Take, e.g., the mechanical work: $W=|F|\cdot |s| \cdot cos\theta.$ We get that 1 Joule is 1 Newton multiplied by 1 meter.

I will try to clarify my next concern with the example from physics.

Let us consider the one-dimensional problem of minimization of the coordinate $x(t)$ of the material point, where $t$ is a time variable. Suppose $x(t)$ is some $(L_0,L_1)$-smooth twice-differentiable function. In the work [7] (see above) the $(L_0,L_1)$-smoothness reduces to the condition $|\nabla^2 x(t)|\le L_0 + L_1|\nabla x(t)|$ in this case. Let us observe that $|\nabla^2 x(t)|$ is the absolute value of the acceleration and is measured in m/s^2. Therefore, $L_0$ has the same unit of measure, m/s^2. In turn, $|\nabla x(t)|$ is the absolute value of the speed and is measured in $m/s.$ Therefore, $L_1$ has the unit of measure $s^{-1}.$

In Theorem 1, when choosing $\gamma,$ the constants $L_0$ and $L_1$ are added to each other. How two quantities with different units of measure can be added to each other? This happens over the whole paper. In Theorem 2 when choosing $\gamma$ in the minimum you compare $\frac{1}{4L_0}$ to $\frac{1}{4L_1\sqrt{T}};$ when you define $\Lambda$ you add them again.

In the original work [7] the stepsize is chosen as either $\frac{1}{L_0+L_1\|\nabla f(x_t)\|}$ or $\frac{1}{L_0+L_1M},$ where $M$ is an upper bound on the gradient, which has the same unit of measure. In the work [8] as well.

The constants $L_0$ and $L_1$ are not added to each other as in your paper. They do not have this problem with the units of measure. Therefore, there might be technical flaws in your proofs.

---

> ### Author Response · Authors · 2024-11-22
> **Detailed reply of your comments, question by question.**
>
> Thank you very much for reviewing our manuscript and providing valuable feedback. Below is a response to the review questions/comments. We have revised the manuscript accordingly and uploaded a revised manuscript. Please let us know if further clarifications are needed.
>
> **Q1:**  There are papers with the analysis of NSGD with momentum [2], Adam [3,4], AdaGrad-Norm/AdaGrad [5,6]. All are tailored for the stochastic optimization in the generalized smooth setting and seem more advanced. All have convergence guarantees. Is the newly proposed I-NSGD faster in practice than any of them?
>
> **A:** Thanks for your question. We agree that both our work and these achieve the optimal rate under similar generalized-smoothness conditions.  While these works focus on analyzing the existing adaptive-type algorithms, our work aims to propose a new and fundamental independent sampling scheme that can inspire new stochastic algorithm designs and hence is complementary to the existing studies on algorithms. We have conducted more experiments to show that our proposed I-NSGD is indeed much faster than SPIDER, Normalization SGD with momentum.
>
> For Phase Retrieval and DRO experiments, the training loss versus sample complexity and detailed description of algorithm settings can be found at Appendix I.
>
> the loss function evaluated every 100 epochs are
> | Method | 100 | 200 | 300 | 400| 500 |
> |---|---|---|---|---|---|
> | SGD | 4837.102334|1835.358158|1240.747515|991.178520|882.728049|
> | NSGD with momentum |3419.421198|237.020564|12.536876|11.795320|11.203911|
> | NSGD | 6345.978257 |579.544323|10.754439|10.820819|10.809943|
> | SPIDER | 165.234828 |253.344905 |222.983539 |199.988661| 196.264103|
> | Clipped SGD | 942.788701| 630.029406|378.531543|187.528112|75.678732|
> | I-NSGD |**150.054129** |**8.492338** |**8.257874** |**8.305479** |**8.305773** |
>
> For DRO, the detailed description of algorithm settings and training loss versus sample complexity can be found at our paper's Appendix I.
>
> The objective function evaluated at each 200 epochs are
>
> | Method | 200 | 400 | 600 | 800| 1000 |
> |---|---|---|---|---|---|
> | SGD |  13.980096 |11.792903 |10.343812 |9.370857 |8.550016 |
> | NSGD with momentum|10.655368 |7.589949 |6.060729 |5.322460 |4.770303 |
> | NSGD |11.015746 |8.015699 |6.468458 |5.478018 |4.947710 |
> | SPIDER |6.059500 |3.829211 |3.559534 |3.520244 |3.482132 |
> | Clipped SGD |8.699038 |6.449909 |5.615763 |5.003331 |4.675120 |
> | I-NSGD |**4.351718** |**3.859514** |**3.717275** |**3.636329** |**3.609229** |
> Experimental results from phase retrieval and DRO shows that I-NSGD is faster NSGD with momentum, SPIDER and Clipped SGD
>
> **Q2:**  The original paper [7], which introduced the generalized smoothness assumption, explains that this assumption is based on empirical findings from language model training and image classification tasks. Does I-NSGD perform well on them?
>
> **A:** Thank you for your question. The generalized smoothness condition we use belongs to $EL_{\text{asym}}^*(\alpha)$ proposed in [1], which is a generalized of $(L_0, L_1)$-smooth condition proposed in [7] by inducing problem-dependent constant $\alpha\in (0,1]$. This modification allows $EL_{\text{asym}}^*(\alpha)$ to characterize function geometry more tightly. However, per request, we conducted experiments to train ResNet on CIFAR 10 with algorithms SGD, Adam, Adagrad, NSGD with momentum, NSGD, Clipped SGD, and I-NSGD. Results are available in Appendix I in the submitted paper.
>
> **Q3:** There is a recent paper [8], which contains a similar to NSGD method with smoothed clipping (Algorithm 1: $(L_0,L_1)$-GD). You mention this work in your paper. I can see that the authors develop the adaptive versions of the algorithm, the accelerated version. How does your theory compares to theirs?
>
> **A:** Thank you for pointing out this reference. We will add more discussions in related work accordingly. We notice that the proposed $(L_0,L_1)$-GD and its acceleration is **deterministic** algorithm. Their analysis is conducted under **convex** settings. Our proposed I-NSGD algorithm's sample complexity, $\mathcal{O}(\epsilon^{-4})$ is obtained on **non-convex** **stochastic** settings.
> The stochastic extension $(L_0, L_1)$-SGD in [8] doesn't include the independent sampling scenario and clipping structure and their convergence analysis is also under **convex** setting. We appreciate their work very much, but this concurrent work[8] has a totally different setting from ours.

---

> ### Author Response · Authors · 2024-11-22
> **Detailed reply of your comments, question by question (continued).**
>
> **Q4a:** There is a standard sanity check procedure in optimization. The field is connected to physics (take the heavy-ball method of Polyak, for instance). And different quantities in physics have different units of measure. In Theorem 1, when choosing the constants $L_0$ and $L_1$ are added to each other. How two quantities with different units of measure can be added to each other?
>
> **A:** Great Question! We agree with your point that some optimization algorithms (such as heavy-ball method) have physical meaning in practice. And we understand some areas related with optimization, such as adaptive control, do assign physical measurement for parameters to make the system output has desired performance and reasonable interpretability. But in optimization, we do not agree with the sanity check that you proposed, i.e., assigning physical units to check learning rate's technical correctness. In generalized smooth condition, $L_0$ and $L_1$ are just scalar numbers that characterize the function's geometry, which is totally different from a real system's physical property.
>
> On the other hand, if $L_0$ and $L_1$ has $m/s^2$ and $m/s$ physical units, the resulting **descent lemma**
> $$
> f(w)\leq f(w')+ \langle \nabla f(w'), w - w' \rangle + \frac{1}{2} (L_0+L_1 || \nabla f(w')||^{\alpha})|| w - w' ||^2
> $$
> **contradicts** to the physical units you assumed since $f(w),\forall w$, and $w, \forall w$ should be measured in terms of $m$. As long as the learning rate can control the update so that the generated sequences from algorithm oracle, such as $f(w_t)$, $\nabla F(w_t)$, $w_t$ can converge, the learning rate is suitable to use for proof. In summary, the role of learning rate is to numerically avoid sequence divergence as algorithm progresses.
>
> **Q4b**: In Theorem 1, when choosing
>  the constants $L_0$
>  and $L_1$ are added to each other. How two quantities with different units of measure can be added to each other?
>
>  **A**: Thanks for raising question about step size rule. We will address your question by listing key steps resulting the design of step-size rule. The main derivation can be found at inequality (23) in appendix, from the descent lemma, by putting the NGD update rule, Lemma E.2 derived in [1], we have
>  $$
> f\big(w_{t+1}\big)-f\big(w_t\big) \leq \gamma ||\nabla f\big(w_t\big)||^{2-\beta}+\frac{\gamma}{4}\big(2 ||\nabla f\big(w_t\big)||^{2-\beta}+\big(2 L_0 \gamma\big)^{\frac{2}{\beta}-1}+\big(2 L_1 \gamma\big)^{\frac{2}{\beta}-1}\big),
>  $$ where the constant on RHS $(2L_0\gamma)^{\frac{2}{\beta}-1}$, $(2 L_1 \gamma\big)^{\frac{2}{\beta}-1}$ is obtained by utilizing **Lemma E.2.** from [1]. This lemma enables us to separate terms including $||\nabla f(w) ||^{2-\beta}$ and other constant.
>
> Then, from fact $a^{\tau}+b^{\tau}\leq (a+b)^{\tau}$ holds for any $a,b\geq 0$ and $\tau\geq 1$, we further combine $(2L_0\gamma)^{\frac{2}{\beta}-1}$, $(2 L_1 \gamma\big)^{\frac{2}{\beta}-1}$, which yields
> $$
>  f\big(w_{t+1}\big)-f\big(w_t\big)\leq -\frac{\gamma}{2}||\nabla f\left(w_t\right)||^{2-\beta}+\gamma^{\frac{2}{\beta}}\big(2 L_0+2 L_1\big)^{\frac{2}{\beta}-1}
> $$
> By designing $\gamma = \frac{(2\mu\epsilon)^{\beta/\rho}}{8(L_0+L_1)+1} $, we can control the constant on RHS to be small and easy to handle in notation during further deductions.

---

> ### Author Response · Authors · 2024-11-22
> **Detailed reply of your comments, question by question (continued).**
>
> **Q4c**In Theorem 2 when choosing
>  in the minimum you compare $\frac{1}{4L_0}$ to $\frac{1}{4L_1\sqrt{T}}$, you add them again.
>
>  **A**: Thanks you for pointing out this question. We notice that this is indeed a typo and we are sorry for the confusion. To clarify, the learning rate in theorem 2 should be
>  $$
>  \gamma = \min \big( \frac{1}{4L_0},\frac{1}{4L_1}, \frac{1}{\sqrt{T}}, \frac{1}{8L_1(3\tau_2/\tau_1)^{\beta}}\big),
>  $$
>  and
>  $$
>  \Lambda = F(w_0)-F^* + \frac{1}{2}(L_0+L_1)(1+ \tau_2^2/\tau_1^2)^2
>  $$
>  For the term $\frac{1}{2}(L_0+L_1)(1+ \tau_2^2/\tau_1^2)^2$, this is produced from Lemma 6 in Appendix F, where we claim
>  $$
>  \frac{1}{2}\gamma^2\frac{(L_0+L_1||\nabla F(w_t)  ||^{\alpha})}{h_t^{2\beta}}\frac{\tau_2^2}{\tau_1^2}\leq \frac{1}{2}\gamma^2(L_0+L_1)(1+\frac{\tau_2^2}{\tau_1^2})^2+ \frac{\gamma}{4h_t^{\beta}}||\nabla F(w_t) ||^2.
>  $$
>  When we prove this fact, we divide upper bound based on value of $||\nabla F(w_t) ||$. When $||\nabla F(w_t) ||$ is a small constant, we upper bound the LHS by a constant; When $||\nabla F(w_t) ||$ is large, we upper bound LHS $||\nabla F(w_t) ||^2/h_t^{\beta}$ associated with constant factor. Finally, we sum the two upper bounds together to yield RHS, as they are always positive.
>
>  To elaborate, when $||\nabla F(w_t) ||\leq \sqrt{1+{\tau_2^2}/{\tau_1^2}}$, we have $||\nabla F(w_t) ||^{\alpha}\leq (1+{\tau_2^2}/{\tau_1^2})^{\frac{\alpha}{2}}$ for $0<\alpha \leq 1$. Since $(1+{\tau_2^2}/{\tau_1^2}) >1$, so does $(1+{\tau_2^2}/{\tau_1^2})^{\frac{\alpha}{2}}$.
>     These facts lead to
>     $$
>         \frac{1}{2}\gamma^2\frac{(L_0+L_1||\nabla F(w_t) ||^{\alpha})}{h_t^{2\beta}}\frac{\tau_2^2}{\tau_1^2}
>         \leq \frac{1}{2}\gamma^2(1+\frac{\tau_2^2}{\tau_1^2})^{\frac{\alpha}{2}}\frac{(L_0+L_1)}{h_t^{2\beta}}\frac{\tau_2^2}{\tau_1^2}
>     $$
>     Here, we derive the upper bound by replacing $||\nabla F(w_t) ||^{\alpha}$ by $(1+{\tau_2^2}/{\tau_1^2})^{\frac{\alpha}{2}}$ and multiply $(1+{\tau_2^2}/{\tau_1^2})^{\frac{\alpha}{2}}$ in front of $L_0$.
>     Moreover, by using the clipping structure $h_t\geq 1$, and $0<\alpha\leq \beta\leq 1$, we have
>     $$
>         \frac{1}{2}\gamma^2(1+\frac{\tau_2^2}{\tau_1^2})^{\frac{\alpha}{2}}\frac{(L_0+L_1)}{h_t^{2\beta}}\frac{\tau_2^2}{\tau_1^2}
>         \leq \frac{1}{2}\gamma^2(L_0+L_1)(1+\frac{\tau_2^2}{\tau_1^2})^{2}.
>     $$
>     The second case where $||\nabla F(w_t) ||>\sqrt{1+\tau_2^2/\tau_1^2}$ will result $\frac{\gamma}{4h_t^{\beta}}||\nabla F(w_t) ||^2$, we omit the discussion here. For more details, you can check Appedix F, Proof of Lemma 6.
>
>
>
> References:
>
> [1] Chen, Ziyi, et al. Generalized-Smooth Nonconvex Optimization is As Efficient As Smooth Nonconvex Optimization.
>
> [2] Hübler, Florian, et al. Parameter-Agnostic Optimization under Relaxed Smoothness.
>
> [3] Wang, Bohan, et al. Provable adaptivity in adam.
>
> [4] Li, H., Rakhlin, A., and Jadbabaie, A., Convergence of adam under relaxed assumptions.
>
> [5] Faw, Matthew, et al Beyond uniform smoothness: A stopped analysis of adaptive sgd.
>
> [6] Wang, Bohan, et al. Convergence of adagrad for non-convex objectives: Simple proofs and relaxed assumptions.
>
> [7] Jingzhao Zhang, Tianxing He, Suvrit Sra, Ali Jadbabaie, Why gradient clipping accelerates training: A theoretical justification for adaptivity
>
> [8]Gorbunov,Eduard, et al. Methods for Convex (L0,L1)-Smooth Optimization: Clipping, Acceleration, and Adaptivity.
>
> [9] Arjevani, Yossi, et al. Lower bounds for non-convex stochastic optimization.
>
> [10]Fang, Cong, et al. "Spider: Near-optimal non-convex optimization via stochastic path-integrated differential estimator." Advances in neural information processing systems 31 (2018).
>
> [11] Cutkosky, Ashok, et al. "Momentum improves normalized sgd." International conference on machine learning. PMLR, 2020.

---

> ### Author Response · Authors · 2024-11-24
> **Update on experiments of training ResNet18 and ResNet50 over CIFAR10 Dataset per request from reviewer AeAr**
>
> We'd like to update our experiment results on training ResNet over CIFAR10 dataset.
> The baseline method includes SGD, Adagrad, Adam, NSGD, NSGD with momentum and Clipped SGD. The loss function we choose is cross-entropy loss. The initialization uses Kaiming initialization. The detailed algorithm settings can be found at Appendix I in our paper. We train all the baseline methods and I-NSGD on CIFAR10 Dataset for 30 epochs each. The resultant training loss recorded every 5 epochs are as following.
>
> **ResNet18** on CIFAR10
>
> |Method|5|10|15|20|25|30|
> |---|---|---|---|---|---|---|
> |SGD| 3.799652861431241|2.6444492801092565|1.8017334272153676|1.160722715780139|0.744824364897795|0.4474578978260979|
> |Adam|1.4727556260768324|0.6025010953308083|0.3352339585107984|0.21858624278684147|0.16716759049450047|0.14670390529863653|
> |Adagrad|1.5795468527358025|0.47506230941507965|0.17185649830207694|0.09319731855066493|0.061325946466240566|0.04338877404734376|
> |NSGD|1.9135879594832659|0.5788353047100827|0.292824345495319|0.21826101756596472|0.17621186005271738|0.15782649893662892|
> |NSGDm|4.045813534408808|2.9456085418350995|2.174826104193926|1.5469997769687325|1.0462557922583073|0.6905137724243104|
> |ClipSGD|1.737209850223735|0.6230987445160281|0.3534815391758457|0.2730324479562114|0.2233854775040527|0.19943709113067598|
> |INSGD|**1.6496218708343804**|**0.6025417874916457**|**0.4097965414985083**|**0.3055273554782616**|**0.26761460003763204**|**0.24892822335095843**|
>
> From the records, we can see that INSGD demonstrates fast convergence, which is comparable with **Adam**, **Clipped SGD** and better than **NSGDm**, **NSGD**, and **SGD**. This demonstrates the effectiveness of the I-NSGD algorithm.
>
> **ResNet50** on CIFAR10
>
> |Method|5|10|15|20|25|30|
> |---|---|---|---|---|---|---|
> |SGD|5.990726664662361|4.939687485806644|4.287755103781819|3.7324438001960516|3.25510926079005|2.729151414707303|
> |Adam|3.6487857876345515|2.9452997343614697|2.0379750484135|1.0010075609316118|0.4167641017556889|0.7922854552452918|
> |Adagrad|4.240041005425155|2.9372318675741553|1.9360805025789887|1.320955990231596|0.9049579487764277|0.634457179432502|
> |NSGD|6.778490765951574|4.921374089084566|3.6041279016062617|2.2895418484695256|1.3302382924593985|0.8264894502353854|
> |NSGDm|7.282688681036234|6.463156248442829|5.85003749653697|5.367515945807099|5.002681262791157|4.6777457147836685|
> |ClipSGD|5.527418973855674|3.230063008144498|1.6175740614999086|0.8517110171960667|0.5524751330958679|0.4216658901714254|
> |INSGD|**3.98776736529544**|**2.0011105102021247**|**0.9845839753979817**|**0.5838269208616111**|**0.4778438211797038**|**0.3469473898294382**|
>
> From the above records, we can see that INSGD indeed demonstrates fast convergence, which is comparable with **Adam**, **Adagrad** and better than **ClipSGD**,**NSGDm**, **NSGD**, **SGD**. This demonstrates the effectiveness of the I-NSGD algorithm.

---

> ### Comment · Reviewer_AeAr · 2024-11-26
>
> Thank you for your response.
>
> I repeat that this sanity check procedure is **standard** in the optimization field. I do not see why the descent lemma contradicts to the procedure. The descent Lemma does not depend on your algorithm, it is based on Calculus. Physical laws do not depend on your algorithm either. By saying that the descent Lemma contradicts the physical units you essentially mean that Physical laws do not merge with Calculus. But Calculus is the language of Physics, all Physical laws are expressed through it. Both of the theories are built so that they merge.
>
> I think I have explained myself very clearly in the previous review and made no mistake. It is a simple procedure, I will repeat it. I will set $\alpha=1.$
>
> $x(t_{n+1}) \le x(t_n) + \langle x'(t_n), t_{n+1} - t_n\rangle + \frac{1}{2}\left(L_0 + L_1|x'(t_n)|\right)|t_{n+1} - t_n|^2.$
>
> Let us go term by term. $x(t_{n+1})$ and $x(t_n)$ are measured in meters $m$ since they are coordinates of the material point at times $t_{n+1}$ and $t_n.$ Next, $x'(t)$ is $m/s$ (the speed), $t_{n+1} - t_n$ is $s$ (seconds, time). Then, $\langle x'(t_n), t_{n+1} - t_n\rangle$ gives $m/s\cdot s = m.$ No contradiction so far! Let's go further, $|t_{n+1} - t_n|^2$ is $s^2,$ therefore, $L_0$ has to be $m/s^2$ as $L_0 \cdot |t_{n+1} - t_n|^2$ has to be measured in meters: $m/s^2 \cdot s^2 = m.$ Next, $L_1$ needs to be $s^{-1}$ since $L_1|x'(t_n)|\cdot |t_{n+1} - t_n|^2 $ has to be in meters: $s^{-1} \cdot m/s \cdot s^2 = m.$
>
> Once again, $L_0$ and $L_1$ have **different** units of measure. These are not just simple constants.
>
> The primary area of your paper is chosen as optimization. Imagine I want to optimize a simple function from Physics with your algorithms (why not, it is optimization, it should work on functions from Physics). I look at your theoretical guarantees, try to use them and suddenly I fail because I can not add $m/s^2$ to $s^{-1}.$
>
> The problem is in the further analysis, I believe.
>
> The paper is theoretical. However, the theory fails on a simple example. I will keep my score.

---

> > ### Author Response · Authors · 2024-11-27
> > **Thanks for your comment and several remarks.**
> >
> > Thank you for raising your concerns regarding the compatibility of units in our analysis. We greatly appreciate the effort you put into articulating this potential issue and recognize the importance of ensuring consistency in physical measurements. While we respectfully disagree with your proposal to replace $||w_t||$ with t (time), we value your perspective and are eager to engage in further discussion to review the key steps that led to these terms. To foster constructive dialogue, we have highlighted the critical steps that resulted in what may appear to be a unit-inconsistent step size. Your view is important, and we hope our work can improve its applicability to physics.
> >
> > **Discussion on $\gamma$**, similarly with [1], by applying NGD update rule and Young's inequality, we are able to obtain
> > $$
> >  f\big(w_{t+1}\big)-f\big(w_t\big)\leq -\frac{\gamma}{2}||\nabla f\left(w_t\right)||^{2-\beta}+\gamma^{\frac{2}{\beta}}\big(2 L_0+2 L_1\big)^{\frac{2}{\beta}-1}
> > $$
> > To further simplify $\gamma^{\frac{2}{\beta}}\big(2 L_0+2 L_1\big)^{\frac{2}{\beta}-1}$, we design $\gamma = \frac{(2\mu\epsilon)^{\beta/\rho}}{8(L_0+L_1)+1} $. Because by extracting $\frac{1}{4}$ out,
> > we have
> > $$
> > (\gamma)^{\frac{2}{\beta}}(\frac{1}{4})^{\frac{2}{\beta}-1}(8L_0+8L_1)^{\frac{2}{\beta}-1}\leq (\gamma)^{\frac{2}{\beta}} (\frac{1}{4})^{\frac{2}{\beta}-1}((8L_0+8L_1)+1)^{\frac{2}{\beta}-1},
> > $$
> > which yields
> > $$
> > \frac{(2\mu\epsilon)^{\frac{2}{\rho}}}{(8(L_0+L_1)+1)^{\frac{2}{\beta}}}(\frac{1}{4})^{\frac{2}{\beta}-1}(8(L_0+L_1)+1)^{\frac{2}{\beta}-1}.
> > $$
> > Since $\frac{1}{4}<1$, $\frac{2}{\beta}-1>1$, $\frac{1}{4}^{\frac{2}{\beta}-1}\leq \frac{1}{4}$.It leads to
> > $$
> > \frac{1}{4}\frac{(2\mu\epsilon)^{\frac{\beta}{\rho}}(2\mu\epsilon)^{\frac{2-\beta}{\rho}}}{8(L_0+L_1)+1},
> > $$ which is equivalent as
> > $$
> > \frac{\gamma}{4} (2\mu\epsilon)^{\frac{2-\beta}{\rho}}.
> > $$
> >
> > **Discussion on $\Lambda$** TL;DR, $\Lambda$ is derived when full gradient is small than certain constant $||\nabla F(w_t) ||\leq \sqrt{1+\frac{\tau_2^2}{\tau_1^2}}$. Under this case, it's nature to upper bound $L_1 ||\nabla F(w_t) ||^{\alpha}$ by constant $L_1 (1+\frac{\tau_2^2}{\tau_1^2})^{\frac{\alpha}{2}}\leq L_1 (1+\frac{\tau_2^2}{\tau_1^2})^{\frac{1}{2}} $. And the rest of derivation aggregates $L_0$ to $L_0 (1+\frac{\tau_2^2}{\tau_1^2})^{\frac{1}{2}}$ and add together with  $L_1 (1+\frac{\tau_2^2}{\tau_1^2})^{\frac{1}{2}}$, yielding
> > $$
> > \frac{1}{2h_t^{2\beta}}\gamma^2(L_0+L_1)(1+\frac{\tau_2^2}{\tau_1^2})\frac{\tau_2^2}{\tau_1^2}\leq \frac{1}{2h_t^{2\beta}}\gamma^2(L_0+L_1)(1+\frac{\tau_2^2}{\tau_1^2})(1+\frac{\tau_2^2}{\tau_1^2})
> > $$
> > Since $h_t = \max\Big(1, (4L_1\gamma)^{\frac{1}{\beta}}\Big({2||\nabla f_{\xi_{B'}}(w_t)||+ \delta}\Big) \Big)\geq 1$, we have $\frac{1}{h_t^{2\beta}}<1$, This further yields exact expression of $\Lambda$.
> > The above derivation is grounded in established mathematical rules and inequalities. We understand you have concerns about potential technical flaws. Could you kindly specify which steps you believe may contain such flaws and which ones you find counterintuitive from a physical perspective? Your insights would be invaluable in helping us address these concerns effectively.
> >
> > [1] Chen, Ziyi, et al. Generalized-Smooth Nonconvex Optimization is As Efficient As Smooth Nonconvex Optimization.

---

### Official Review · Reviewer_LTZr · 2024-10-31

**Soundness:** 3
**Presentation:** 2
**Contribution:** 1
**Rating:** 3
**Confidence:** 5

**Summary:**

The paper analyzes stochastic normalized SGD with an extra sampling under generalized smoothness assumption and obtains known complexity rate. For the deterministic case paper additionally analyzes normalized SGD under generalized Polyak-Łojasiewicz condition.

**Strengths:**

- the convergence of normalized gradient descent under the generalized Polyak-Łojasiewicz (PŁ) condition
- generalized variance assumption

**Weaknesses:**

- only symmetric case, which is much easier to analyze; [Hubler etal, 2024] (mentioned by authors) obtains the same rate with non-symmetric, more general smoothness setup
- proposed I-NSGD requires additional sampling, while does not improve the known complexity rate. [Hubler etal, 2024] does not need additional sampling and has the same rate.
- having the above, [Hubler etal, 2024] is not experimentally compared

**Questions:**

- I would like to hear authors comment on the result with Polyak-Łojasiewicz condition. Despite I am not aware of such a result and it is clearly a contribution then, its value seems limited. I would like authors to explain in details why this result is important. I would also like an explanation of why NSGD is considered under this setup and not SGD for example. Does NSGD remarkably benefit from Polyak-Łojasiewicz setup?
- But my main concern is contribution of proposed I-NSGD. The most important factor is the focus of the paper on only symmetric generalized smoothness. Non-symmetric smoothness holds for a problem formulation appearing in Distributionally Robust Optimization (Jin et al., 2021). Chen et al. (2023) also show that exponential function satisfies non-symmetric case. Moreover, the rate is the same despite the less general assumption.
So, I want authors comment on why they think the case of less general smoothness overweights more general variance assumption. Can authors argue that the more general variance assumption is not compatible with a kind of [Hubler etal, 2024] analysis? Why haven't authors included [Hubler etal, 2024]'s NSGD with momentum to experimental comparison? It is highly related method to the best of my understanding.
By now, neither experiments nor theory tells me that I-NSGD is preferable in any scenario.

References:
Jikai Jin, Bohang Zhang, Haiyang Wang, and Liwei Wang. Non-convex distributionally robust optimization: Non-asymptotic analysis. Advances in Neural Information Processing Systems, 34: 2771–2782, 2021
Ziyi Chen, Yi Zhou, Yingbin Liang, and Zhaosong Lu. Generalized-smooth nonconvex optimization is as efficient as smooth nonconvex optimization. In International Conference on Machine Learning, pp. 5396–5427. PMLR, 2023.

---

> ### Author Response · Authors · 2024-11-22
> **Detailed reply of your comments, question by question.**
>
> Thank you very much for reviewing our manuscript and providing valuable feedback. Below is a response to the review questions/comments. We have revised the manuscript accordingly and uploaded a revised manuscript. Please let us know if further clarifications are needed.
>
> **Q1a:** Why (generalized) Polyak-Łojasiewicz condition is important? Its value seems to be limited.
>
> **A:** Great Question! we have updated our related work to discuss more about the motivation of studying generalized PL condition.
> Initially, the PL condition attracted researchers' attention because it relaxes the strong convexity requirement. Recently, from the deep learning perspective, [5] found that PL* -- a variant of PL condition widely exists in loss landscape of over-parametrized neural networks. They also provide explanation why SGD-type algorithm can converge fast when training over-parametrized neural network with mean-squared loss.
> Later, [6] further proposed a more general $\psi$-KL* condition to include more loss function such as the CE loss, hinge loss and also provide convergence guarantee of SGD. Interestingly, it turns out that the generalized PL condition is a sub-class of $\psi$-KL* condition if $F^*$ is non-negative. This is because if $F^*\geq 0$, we can rewrite the generalized PL condition as
> $$
> ||\nabla F(w) || \geq \big[(2\mu)\cdot (F(w)-F^*)\big]^{1/\rho}\geq \big[(2\mu)\cdot (F(w))\big]^{1/\rho},
> $$
> where the function $f(.)=(.)^{1/\rho}$ is non-decreasing for $\rho>0$. Based on these observations, we are motivated to analyze the convergence of stochastic algorithm under the generalized PL condition.
>
> **Q1b:** In deterministic case, why NGD is considered under this setup but not GD? Does NGD remarkably benefit from Polyak-Łojasiewicz setup?
>
> **A:** Great Question!
> In the deterministic setting, [2] showed that generalized-smooth functions have the following challenging local geometry
> $$
> f(w) \leq f(w')+ \langle \nabla f(w'), w - w' \rangle + \frac{1}{2} (L_0+L_1 || \nabla f(w')||^{\alpha}) || w - w' ||^2,
> $$
> where the term $L_1 ||\nabla f(w') ||^{\alpha}||w-w' ||^2$ induced by generalized-smoothness scales polynomially with the gradient norm. This motivates [8] to apply NGD update to control such an error term induced by generalized-smooth geometry.
>
> In [2], the authors also have conducted experiments showing that NGD outperforms GD in solving various nonconvex generalized-smooth machine learning problems. Moreover, the previous work [6] has studied GD under $\ell$-generalized smooth. The convergence result, and $\ell$-smooth requires strong assumptions on initial gradient bound $G$, which is impractical. These evidences indicate NGD is more adaptive to the generalized-smooth geometry than GD, and motivates us to continue studying NGD under the generalized PL condition, which has attracted much attention in optimization and deep learning.
>
> **Q2:** The most important factor is the focus of the paper only on **asymmetric** generalized smooth, **symmetric** smoothness holds for a problem appearing in [2]. Why the author picks the less general smoothness condition?  Why they think the case of less general smoothness overweights more than general variance assumption?
>
> **A:** Thanks for your question! We study the asymmetric generalized-smoothness just to simplify the calculations in the proof. We note that our analysis can be directly generalized to the **symmetric** generalized smooth case with minor modifications. Specifically, in the deterministic case, we can choose the following NGD learning rate instead.
> $$
> \gamma = \frac{(2\mu\epsilon)^{\beta/\rho}}{12(K_0+K_1+2K_2)+1} \text{ when }\alpha \in (0,1),
> $$
> and
> $$
> \gamma = \frac{(2\mu)^{\frac{1}{\rho}}}{4L_0+1} \text{ when } \alpha=1,
> $$
> where $ K_0:=L_0\big(2^{\frac{\alpha^2}{1-\alpha}}+1\big), K_1:=L_1 \cdot 2^{\frac{\alpha^2}{1-\alpha}} \cdot 3^\alpha, K_2:=L_1^{\frac{1}{1-\alpha}} \cdot 2^{\frac{\alpha^2}{1-\alpha}} \cdot 3^\alpha(1-\alpha)^{\frac{\alpha}{1-\alpha}}$.
> The same conclusion and proof logic still hold. The stochastic setting can be similarly addressed by using a slightly smaller learning rate.
>
> To provide a intuitive justification, note that the only difference between **asymmetric** and **symmetric** generalized-smoothness is that the definition of asymmetric smoothness depends on $||\nabla F(w) ||$ whereas the symmetric smoothness depends on $\max_{\theta \in [0,1]}||\nabla F(w_{\theta}) ||$ with $w_{\theta} = (1-\theta)w+\theta w'$. Such a difference is marginal since the distance between $w_{t+1}$ and $w_t$ is bounded very small in the proof (by using a small learning rate). Thus, we use the **asymmetric** version merely for simplicity of the proof. On the other hand, we note that the machine learning applications (DRO, deep networks, etc) mentioned in the paper satisfies asymmetric generalized smooth. The proof of DRO problem can be found at proposition 3.2 in [2].

---

> ### Author Response · Authors · 2024-11-22
> **Detailed reply of your comments, question by question (continued)**
>
> **Q3:** Can authors argue that the more general variance assumption is not compatible with analysis conducted in the work [3]?
>
> **A:** Great question! We have added some discussions in our related work accordingly. [3] studies normalized gradient with momentum under $(L_0, L_1)$-generalized smoothness with bounded variance, and the learning rate they adopt is in the parameter-free regime. We found that their Lemma 13 in appendix served as a key inequality under their variance assumptions, where they upper bound $||m_t-\nabla F(w_t) ||$ by some terms including
> $$
> \mathbb{E}\Big[||\sum_{\tau=1}^t \beta_{(\tau+1): t} \alpha_\tau (g_{\tau}-\nabla F(w_{\tau}))||\Big] \leq \sqrt{\sum_{\tau=1}^t \beta_{(\tau+1): t}^2 \alpha_t^2 \sigma^2},
> $$ where $\sigma^2$ represents their variance.
>
> If we consider the more general variance condition $||g_{\tau}-\nabla F(w_{\tau}) ||\leq \tau_1 ||\nabla F(w_{\tau}) ||+\tau_2$ used in our work for the above inequality, it will induce additional terms $||\nabla F(w_{\tau})||^2$ in order to bound $g_{\tau}-\nabla F(w_{\tau})$, which is no longer compatible with their original analysis.
> Moreover, our I-NSGD doesn't include momentum $m_t$ and parameter-agnostic step size rule in algorithm, Lemma 13 and their convergence analysis doesn't apply to our analysis. So our analysis is not compatible with [3].
>
> **Q4:** Why haven't authors included [Hubler etal, 2024]'s NSGD with momentum to experimental comparison? Can you add experiments with NSGD with momentum for comparison?
>
> **A** Thanks for your suggestion, in the new experiments, we add NSGD with momentum and SPIDER for comparison. The training loss versus sample complexity and detailed description of algorithms'setting can be found at Appendix I of our paper.
> For Phase retrieval experiments, the loss function evaluated every 100 epochs are
>
> | Method | 100 | 200 | 300 | 400| 500 |
> |---|---|---|---|---|---|
> | SGD |4837.102334|1835.358158|1240.747515|991.178520|882.728049|
> | NSGD with momentum |3419.421198|237.020564|12.536876|11.795320|11.203911|
> | NSGD | 6345.978257 |579.544323|10.754439|10.820819|10.809943|
> | SPIDER | 165.234828 |253.344905 |222.983539 |199.988661|196.264103|
> | Clipped SGD | 942.788701| 630.029406|378.531543|187.528112|75.678732|
> | I-NSGD |**150.054129**|**8.492338**|**8.257874**|**8.305479**|**8.305773**|
>
> For DRO, the detailed description of algorithm settings and the training loss versus sample complexity can be found at Appendix I. The objective function evaluated at each 200 epochs are
> | Method | 200 | 400 | 600 | 800| 1000 |
> |---|---|---|---|---|---|
> | SGD | 13.980096 |11.792903 |10.343812 |9.370857 |8.550016 |
> | NSGD with momentum|10.655368 |7.589949 |6.060729 |5.322460 |4.770303 |
> | NSGD |11.015746 |8.015699 |6.468458 |5.478018 |4.947710 |
> | SPIDER |6.059500 |3.829211 |3.559534 |3.520244 |3.482132 |
> | Clipped SGD |8.699038 |6.449909 |5.615763 |5.003331 |4.675120 |
> | I-NSGD |**4.351718**|**3.859514**|**3.717275**|**3.636329**|**3.609229**|
> From the recorded training loss for Phase Retrieval and DRO, we saw that I-NSGD is significantly faster than other baseline methods, Clipped SGD, SPIDER, NSGD, and NSGD with momentum. This verifies the effectiveness of proposed I-NSGD algorithm.
>
> References:
>
> [1]Jin, Jikai, et al. Non-convex distributionally robust optimization: Non-asymptotic analysis. Advances in Neural Information Processing Systems, 34: 2771–2782, 2021
>
> [2]Chen, Ziyi, et al. Generalized-smooth nonconvex optimization is as efficient as smooth nonconvex optimization. In International Conference on Machine Learning, pp. 5396–5427. PMLR, 2023.
>
> [3] Florian Hubler, et al. Parameter-agnostic optimization under relaxed smoothness. In Proceedings of The 27th International Conference on Artificial Intelligence and Statistics, 2024.
>
>
> [4] Liu, Chaoyue, et al. "Loss landscapes and optimization in over-parameterized non-linear systems and neural networks." Applied and Computational Harmonic Analysis 59 (2022): 85-116.
>
> [5] Scaman, Kevin, et al. "Convergence rates of non-convex stochastic gradient descent under a generic lojasiewicz condition and local smoothness." International conference on machine learning. PMLR, 2022.
>
> [6]Li, Haochuan, et al. "Convex and non-convex optimization under generalized smoothness." Advances in Neural Information Processing Systems 36 (2024).

---

> > ### Comment · Reviewer_LTZr · 2024-11-22
> > **Official Comment by Reviewer LTZr**
> >
> > Thank for your response.
> >
> > 1. I would like to thank authors explaining values of PL. But I wanted to say that PL simplifies analysis and theory and limits the contribution. That explains my score, just does not seem good enough to me. It would be grate to see an interesting unique problem satisfying your setup, or an advance in theory allowing to obtain much better bound, for example without the additional exponent (see below).
> >
> > 2. According to
> >
> > Chen, Ziyi, et al. "Generalized-smooth nonconvex optimization is as efficient as smooth nonconvex optimization." International Conference on Machine Learning. PMLR, 2023.
> >
> > the inequality in the authors' answer to Q1b has additional exponent in symmetric case. As a far as I understand, authors picked the less general assumption because they just can hide the this additional exponent to $\mathcal O$. Results
> > of
> >
> > Vankov, Daniil, et al. "Optimizing -Smooth Functions by Gradient Methods." arXiv preprint arXiv:2410.10800 (2024).
> >
> > give promises, that for GD the right rates should recover the standard one . The work is very recent, I cannot appeal to it, so it is just a comment, but it should explain why i think symmetric case is much trickier.

---

> ### Author Response · Authors · 2024-11-22
> **Explanation (continued)**
>
> Thanks for your quick reply. We'd like to add more discussions regarding on your claim.
>
> **Claim**: PL condition simplifies the proof.
>
> **Discussion**: In Chen et al, the authors have already studied NGD convergence in non-convex settings and reach $\mathcal{O}(\epsilon^{-2})$ iteration complexities. Our study continues contributing through this line under generalized PL condition. Unlike applying the original PL condition, which can directly replace $||\nabla f(w_t) ||^2$ by $2\mu \cdot (f(w_t)-f^*)$ to lead linear convergence, our generalized PL condition is more difficult to handle with because of $\rho$ and $\beta$. These difficulties finally lead us to the descent lemma
> $$
>   \Delta_{t+1} \leq \Delta_t - \frac{\gamma (2\mu)^{\frac{2-\beta}{\rho}}}{4} \Delta_t^{\frac{2-\beta}{\rho}},
> $$ where $\Delta_t = f(w_t)-f^*$.
> We have to discuss the convergence behavior of this sequence by different values of $\rho$ and $\beta$, which correspond to three different case states in our Theorem 1.
>
> **Claim**: We use asymmetric generalized smooth to hide mathematical tricks.
>
> **Discussion**:
> We agree with you that the symmetric generalized smooth condition will result additional terms on RHS as we wrote in Q1b.
> And the descent lemma is
> $$
> f\left(w^{\prime}\right) \leq f(w)+\nabla f(w)^{\top}\left(w^{\prime}-w\right)+\frac{1}{2}||w^{\prime}-w||^2\left(K_0+K_1||\nabla f(w)||^\alpha+2 K_2||w^{\prime}-w||^{\frac{\alpha}{1-\alpha}}\right) .
> $$
> when $\alpha\in (0,1]$.And,
> $$
> f\left(w^{\prime}\right) \leq f(w)+\nabla f(w)^{\top}\left(w^{\prime}-w\right)+\frac{1}{2}||w^{\prime}-w||^2\left(L_0+L_1||\nabla f(w)||\right) \exp \left(L_1 ||w^{\prime}-w||\right)
> $$
> when $\alpha=1$.
> As we have explained in **Q2**, in this scenario, by choosing $$
> \gamma = \frac{(2\mu\epsilon)^{\beta/\rho}}{12(K_0+K_1+2K_2)+1} \text{ when }\alpha \in (0,1),
> $$
> and
> $$
> \gamma = \frac{(2\mu)^{\frac{1}{\rho}}}{4L_0+1} \text{ when } \alpha=1.
> $$
> NGD will lead to the same convergence conclusion. The proof logic is same as the proof given in Appendix, except inducing another small component $K_2 \gamma^{\frac{1}{1-\alpha}} \cdot||\nabla f\left(w_t\right)||^{\frac{(2-\alpha)(1-\beta)}{1-\alpha}}$. To deal with terms including $K_2 \gamma^{\frac{1}{1-\alpha}} \cdot||\nabla f\left(w_t\right)||^{\frac{(2-\alpha)(1-\beta)}{1-\alpha}}$, $K_0 \gamma \cdot||\nabla f\left(w_t\right)||^{2-2 \beta}$, and $ K_1 \gamma \cdot||\nabla f\left(w_t\right)||^{2+\alpha-2 \beta}$, Chen et al. introduce **Lemma E.1**, where they utilize Young's inequality to unify the orders of all terms containing $||\nabla f(w_t) ||$ to $2-\beta$.  This technique is also used in our proof when dealing with $L_0 \gamma \cdot||\nabla f\left(w_t\right)||^{2-2 \beta}$,$ L_1 \gamma \cdot||\nabla f\left(w_t\right)||^{2+\alpha-2 \beta}$.
>
> We'd like to thank the reviewer for pointing out this excellent reference in your comment. We saw that their convergence rate indeed touched the desired bound in different cases. But conclusions are based on  $(L_0, L_1)$ generalized smooth, where Chen et al. classify such condition as $L_H^*$-generalized smooth. And it is also a sub-class of **symmetric** generalized smooth defined in the paper done by Chen et al. It's unclear regarding your statement, "Results from Vankov Daniil et al. give promises that for GD the right rates should recover the standard one." because the authors didn't analyze algorithms under **symmetric** generalized smooth with consideration of $\alpha$ and existence of $\max_{\theta \in [0,1]} ||\nabla F(w_{\theta}) ||$. We're not sure if their conclusions have direct implications for potential improvements in our results under symmetric generalization smooth conditions, as the designed learning rates are different and some of their convergence results are dependent on radius $|| x_0 - x^*||$, where we didn't assume such constant.  Could you explain it more explicitly? We are happy to respond to your concerns.
>
> References
>
> Vankov, Daniil, et al. "Optimizing $(L_0, L_1)$-Smooth Functions by Gradient Methods." arXiv preprint arXiv:2410.10800 (2024).
>
> Chen, Ziyi, et al. "Generalized-smooth nonconvex optimization is as efficient as smooth nonconvex optimization." International Conference on Machine Learning. PMLR, 2023.

---

### Official Review · Reviewer_8SN3 · 2024-11-04

**Soundness:** 3
**Presentation:** 3
**Contribution:** 2
**Rating:** 5
**Confidence:** 4

**Summary:**

This paper examines non-convex optimization under the well-known generalized smoothness condition and an extended (PL) condition. Under these relaxed assumptions, they establish the convergence rate of the Normalized Gradient Descent (NGD) method under both deterministic and stochastic regimes. In particular, for the stochastic setting, they address the limitations of NGD by using independent samples to compute the direction and normalization factor of the stochastic gradient. This method, referred to as Independently-Normalized SGD (I-NSGD), provably converges in a rate of $O(\epsilon^{-4})$.

**Strengths:**

Overall, this is a solid and well-written paper. The proofs have a good presentation and appear to be sound based on my reading. The rates are novel under the generalized PL condition. In particular, the idea of using different samples for the direction and scale of the normalized stochastic gradient estimator seems to be an interesting trick and might find its application in similar situations.

**Weaknesses:**

Nevertheless, the major concern comes from the novelty and the contribution perspective. This paper is clearly an extension of [Chen et al., 2023] and establishes new rates under the generalized PL condition. While some rates are indeed novel, no results really outperform the known results in comparable settings. Moreover, the PL condition seems to greatly restrict the universality of these rates: although PL does not imply convexity, it allows the objective to have favorable geometric properties that lead to fast rates, sometimes even comparable to the convex or strongly-convex setting. This makes the analysis under PL to be less general and less interesting.

A more serious concern is about the stochastic regime. The authors argue the analysis of I-NSGD outperforms [Zhang et al., 2021] by using an affine noise bound, as in Assumption 4. Nevertheless, similar assumptions have already been explored in other prior works like [1], [2], and [3], where convergence rates of $O(\epsilon^{-4})$ are established for adaptive gradient methods when the same general smooth assumption is satisfied. This reduces the contribution of this work, as the above results already achieve the optimal rate under similar conditions. A more thorough discussion of the connection to the literature would help the readers have a more complete understanding of the whole picture.

As a result, it seems that the current paper does not bring a lot of novel insights to the optimization community. I tend to reject this paper unless they are able to provide a more convincing argument to defend the broader impact of this work.

References

[1] Bohan Wang, Huishuai Zhang, Zhiming Ma, and Wei Chen. Convergence of AdaGrad for non-convex objectives: simple proofs and relaxed assumptions. COLT 2023.

[2] Matthew Faw, Litu Rout, Constantine Caramanis, and Sanjay Shakkottai. Beyond uniform smoothness: a stopped
analysis of adaptive SGD. COLT 2023.

[3] Yusu Hong, Junhong Lin. Revisiting Convergence of AdaGrad with Relaxed AssumptionsRevisiting Convergence of AdaGrad with Relaxed Assumptions.

**Questions:**

There is no further questions.

---

> ### Author Response · Authors · 2024-11-22
> **Detailed reply of your comments, question by question.**
>
> Thank you very much for reviewing our manuscript and providing valuable feedback. Below is a response to the review questions/comments. We have revised the manuscript accordingly and uploaded a revised manuscript. Please let us know if further clarifications are needed.
>
> **Q1** PL condition greatly restrict the universality of these rates. Although PL does not imply convexity, it allows the objective to have favorable geometric properties that lead to fast rates, sometimes even comparable to the convex or strongly-convex setting. Can author explain the value of inducing generalized PL condition?
>
> **A:** Great Question! we have updated our related work to discuss more about the motivation of studying generalized PL condition.
> Initially, the PL condition attracted researchers' attention because it relaxes the strong convexity requirement. Recently, from the deep learning perspective, [5] found that PL* -- a variant of PL condition widely exists in loss landscape of over-parametrized neural networks. They also provide explanation why SGD-type algorithm can converge fast when training over-parametrized neural network with mean-squared loss.
> Later, [6] further proposed a more general $\psi$-KL* condition to include more loss function such as the CE loss, hinge loss and also provide convergence guarantee of SGD. Interestingly, it turns out that the generalized PL condition is a sub-class of $\psi$-KL* condition if $F^*$ is non-negative. This is because if $F^*\geq 0$, we can rewrite the generalized PL condition as
> $$
> ||\nabla F(w) || \geq \big[(2\mu)\cdot (F(w)-F^*)\big]^{1/\rho}\geq \big[(2\mu)\cdot (F(w))\big]^{1/\rho},
> $$
> where the function $f(.)=(.)^{1/\rho}$ is non-decreasing for $\rho>0$. Based on these observations, we are motivated to analyze the convergence of stochastic algorithm under the generalized PL condition.
>
> **Q2:** The authors argue that the analysis of I-NSGD outperforms [7] by using an affine noise bound stated in assumption 4. Nevertheless, similar assumptions have already been explored in other prior works like [1], [2], and [3], where convergence rates of
> are established for adaptive gradient methods when the same general smooth assumption is satisfied. This reduces the contribution of this work, as the above results already achieve the optimal rate under similar conditions. Can author add a more thorough discussion of the connection to the literature would help the readers have a more complete understanding of the whole picture?
>
> **A:** Thanks for sharing these previous works. We have added a discussion in the related work to compare with these works, and we summarize it below for your reference. We agree that both our work and the works [1-3] achieve the optimal rate under similar generalized-smoothness conditions.
> While [1-3] focus on analyzing the existing adaptive-type algorithms, our work aims to propose a new and fundamental independent sampling scheme that can inspire new stochastic algorithm designs and hence is complementary to the existing studies on algorithms.
>
> **Discussion:** The references [1], [2], [3] focus on analyzing the existing algorithms from the AdaGrad family. The update rule of Adagrad leverages momentum and adaptive gradients, and hence is substantially different from the basic normalized SGD and clipped SGD. To elaborate, in Adagrad, the normalization term is accumulated as $v_{t} = v_{t-1}+||g_t ||^2$, the update rule is $w_{t+1} = w_{t}-\gamma\frac{1}{\sqrt{v_t}}g_t$. The normalized SGD and clipped SGD takes the form $w_{t+1} = w_t - \gamma \frac{1}{||g_t ||+\delta}g_t$ or $w_{t+1} = w_t - \gamma \frac{1}{\max( ||g_t ||+\delta, \eta)}g_t$, where $\eta$ and $\delta$ are some constant. The accumulation term $v_t$ will lead huge difference for convergence analysis.
>
> Instead of focusing on the existing algorithms, our work aims to develop a new and fundamental sampling scheme, and further apply it to Clipped SGD to solve generalized-smooth problems. Based on this new sampling scheme, we are able to relax the constant upper bound developed in [7] for controlling the stochastic gradient bias and achieve a comparable convergence rate with constant-level batch sizes.
> To elaborate the novelty of our independent sampling, as illustrated in Figure 1, normalized SGD/clipped SGD without independent sampling take biased updates in expectation. By leveraging independent sampling, our proposed I-NSGD can reduce bias in expectation especially when the stochastic gradients have diverse directions.
> Furthermore, in the design of our I-NSGD, we specifically adapt the hyper-parameters to the generalized-smoothness and affine stochastic gradient noise to improve the analysis under independent sampling. As a result, the technical proof of I-NSGD is substantially different from that of the existing algorithms.

---

> ### Author Response · Authors · 2024-11-22
> **References**
>
> References
>
> [1] Wang, Bohan, et al. Convergence of AdaGrad for non-convex objectives: simple proofs and relaxed assumptions. COLT 2023.
>
> [2] Faw, Matthew, et al. Beyond uniform smoothness: a stopped analysis of adaptive SGD. COLT 2023.
>
> [3] Hong, Yusong, et al. Revisiting Convergence of AdaGrad with Relaxed Assumptions.
>
> [4]Arjevani, Yossi, et al. Lower bounds for non-convex stochastic optimization.
>
> [5] Liu, Chaoyue, et al. "Loss landscapes and optimization in over-parameterized non-linear systems and neural networks." Applied and Computational Harmonic Analysis 59 (2022): 85-116.
>
> [6] Scaman, Kevin, et al. "Convergence rates of non-convex stochastic gradient descent under a generic lojasiewicz condition and local smoothness." International conference on machine learning. PMLR, 2022.
>
> [7] Zhang, Jingzhao, et al. "Why gradient clipping accelerates training: A theoretical justification for adaptivity." arXiv preprint arXiv:1905.11881 (2019).
>
> [8] Kingma, Diederik et al. "Adam: A method for stochastic optimization." arXiv preprint arXiv:1412.6980 (2014).

---

### Official Review · Reviewer_8dpS · 2024-11-04

**Soundness:** 3
**Presentation:** 3
**Contribution:** 3
**Rating:** 6
**Confidence:** 3

**Summary:**

This paper considers the problem of nonconvex optimization under the $(L_0, L_1)$ smoothness condition (also known as generalized smoothness) and specifically considers two problems: (a) adapting to the generalized Polyak-Łojasiewicz (PŁ) condition, a generalization of strong convexity, when it exists, and (b) solving the stochastic problem, with or without the PŁ condition. For the former problem, the authors show that the normalized gradient descent method can achieve different (faster) rates depending on the PŁ condition, recovering linear convergence in the classical case ($L_1 = 0$ in generalized smoothness and $\rho = 2$ in the generalized PŁ condition). For the stochastic problem, the authors consider a variant of normalized gradient descent where the normalization is done with a different, independently sampled point. Theorem 2 shows that this method can achieve the $1/\epsilon^4$ convergence rate (to achieve error $\epsilon$) without the need for large ($\epsilon$-dependent) batch sizes or assuming that the stochastic gradient noise norm is bounded almost surely. Finally, experiments show the proposed algorithm performs well compared to the baselines (SGD, Normalized SGD, SGD with clipping).

**Strengths:**

- The paper's idea is pretty nice, I'm quite surprised independent normalization has not been considered in the literature before-- it's such a simple and straightforward solution to the problem of correlated samples in normalization. That said, it might be the case that it works worse in practice than normalized SGD (see the weaknesses section), but we know that normalized SGD may be nonconvergent in general. Independent samples have been used before in the estimation of the smoothness constant in ordinary (smooth) optimization [1], but not in generalized smoothness optimization as far as I can see.
- The analysis removes the requirement for large batch sizes or almost surely bounded gradient noise present in prior work, although it still requires very specific choices of (constant) batch sizes.
- The paper is written quite clearly, and is easy to read.

[1] Malitsky and Mischenko, Adaptive Gradient Descent without Descent, arXiv 1910.09529 (2020).

**Weaknesses:**

- (Unrealistic requirement on the batch size) The batch sizes used, while constant, depend directly on the constants $\tau_1$ and $\tau_2$ in Assumption 4. These constants are in general not knowable, how are we to choose a batch size that depends on them when we can't know them? Can Theorem 2 be generalized to hold for any arbitrary batch size instead? For example, it seems that the rate explodes if $\tau_1 = 0$, which should be an easier case.
- (Unfair experimental evaluation) It seems that for I-NSGD, you tune the $\beta$ parameter while for NSGD you do not? For a fair comparison, it should be tuned as well. The same applies to gradient clipping: the text states you did tune the learning rate for the algorithm, but not the clipping constant. That should be tuned.
- (Limited technical novelty) The technical novelty in this paper is really quite limited, save for using independent normalization all the tools used are quite direct and similar to prior work.

Overall, despite the drawbacks, I lean towards recommending acceptance for the paper

**Questions:**

- Please address my concerns in the weaknesses sections.
- Why is the generalized PŁ condition interesting to study? I know you do give an example in lines 172-175 but how relevant is this in practice?

---

> ### Author Response · Authors · 2024-11-22
> **Detailed reply to your review comments question by question**
>
> Thank you very much for reviewing our manuscript and providing valuable feedback. Below is a response to the review questions/comments. We have revised the manuscript accordingly and uploaded a revised manuscript. Please let us know if further clarifications are needed.
>
> **Q1a** Can theorem 2 hold for any batch size ? When $\tau_1=0$, the learning rate explodes.
>
> **A**: Thanks for pointing this out. We set the batch size as $\mathcal{O}(\tau_1^2)$ just to simplify the calculations in the proof. In fact, our proof still holds for the more relaxed batch size choice $\mathcal{O}(\tau_1^2)+\mathcal{O}(1)$, which resolves the learning rate explosion issue. Since $\tau_1$ is normally a numerical constant, we can use any constant-level batch size. We have updated the theorem statement and its proof accordingly. In our experiments, we showed that for both Phase Retrieval and Distributionally Robust Optimization problems, constant-level batch sizes like $B=64,B' = 8$ and $B=128, B'=32$ always work well.
>
> **Q1b:** In theorem 2, the batch size is directly related with $\tau_1$ and $\tau_2$, which are generally unknown. How can we choose a batch size when we don't know $\tau_1$ and $\tau_2$?
>
> **A:** As we explained in the response to Q1, our proof still holds for the more relaxed batch size choice $\mathcal{O}(\tau_1^2)+\mathcal{O}(1)$, which allows to use any constant-level batch size since $\tau_1$ is normally a numerical constant. All our experiments use constant-level batch sizes.
>
> **Q2:** The experiment is not fair enough: 1. for I-NSGD, you tune $\beta$ while for NSGD you didn't; 2. For clipping constant, this hyper-parameter should be tuned as well. Can you report the performance of additional experiment results?
>
> **A:** Yes, we have conducted more experiments per the reviewer's request. We first want to clarify that the existing algorithms such as Clip-SGD and NSGD use the normalization hyper-parameter $\beta=1$ by default.
> In our new experiment,we add baseline methods: NSGD with momentum and SPIDER algorithm for comparison. And per request from you, we unify all the normalization algorithm's $\beta$ as $2/3$. The training loss curve versus sample compelxity, and the detailed algorithm settings can be found at our updated Appendix I.
> The objective value evaluated every 100 iterations are listed as following
>
> | Method | 100 | 200 | 300 | 400| 500 |
> |---|---|---|---|---|---|
> | SGD | 4837.102334 |1835.358158|1240.747515|991.178520|882.728049|
> | NSGD with momentum|619.606991|20.211925|9.082003|8.980433|8.633882|
> | NSGD |718.194839|65.204719|8.767983|8.818077|8.699032|
> | SPIDER |47.629694|48.652767|36.696431|45.175731|45.658158|
> | Clipped SGD |665.092092|10.367391|8.263630|8.250133|8.219856|
> | I-NSGD |**150.054129**|**8.492338**|**8.257874**|**8.305479**|**8.305773**|
>
> For the DRO problem, the result recorded every 200 iterations is
>
> | Method | 200 | 400 | 600 | 800| 1000 |
> |---|---|---|---|---|---|
> | SGD |13.980096 |11.792903 |10.343812 |9.370857 |8.550016|
> | NSGD with momentum|6.010543 |4.569586 |4.068003 |3.896011 |3.797351 |
> | NSGD |6.421145|4.610153|4.186758|3.916136|3.789031|
> | SPIDER |5.032559 |3.795189 |3.596386 |3.540992 |3.507888 |
> | Clipped SGD |4.633071 |3.900307 |3.943651 |3.662261 |3.642133 |
> | I-NSGD |**4.429903**|**3.892592**|**3.736907**|**3.648593**|**3.615151**|
> The experiment conducted on Phase retrieval and DRO provide evidence of I-NSGD faster convergence than baseline methods, such as SPIDER, NSGD with momentum, NSGD, and Clipped SGD

---

> > ### Comment · Reviewer_8dpS · 2024-11-24
> >
> > 1. I understand that $\tau_1$ is a constant but again if I need my batch size to be at least $\tau_1$, how can I find that? As far as I understand there's no good algorithm for estimating $\tau_1$ globally.
> >
> > 2. Thanks for the updated experiments. I'd also tune $\beta$ per-algorithm rather than fix it for everyone. That should be a more fair comparison.
> >
> > 3. Thanks for the discussion on novelty and more motivation for the generalized PL condition. I believe the latter should be included in the paper.

---

> ### Author Response · Authors · 2024-11-22
> **Detailed reply to your review comments (continued)**
>
> **Q3:** The technical novelty is limited except for using independent normalization. All other tools used are quite direct and similar to prior work. Can you explain explicitly about the technical novelty and technical tools?
>
> **A:** We would like to briefly compare our work and the previous work [1] and [2]. [1] is the first work to propose $(L_0, L_1)$-generalized smoothness and study Clipped SGD's convergence under bounded stochastic gradient bias assumption. Later, [2] proposes the $EL_{\text{asym}}^*(\alpha)$ generalized-smoothness and studied the convergence of SPIDER [3] under affine bounded variance assumption. Their batch size choices are $\Omega(\tau_2^2/\epsilon^2)$, which can be impractical. Our work continued on contributing to this line of research, we proposed the I-NSGD with newly designed learning rate and clipping rule tailored for optimizing  $EL_{\text{asym}}^*(\alpha)$ generalized-smooth functions under the relaxed gradient variance assumptions. Moreover, our algorithm design achieves convergence under constant-level batch size.
>
> We agree that independent normalization is a simple modification in the algorithm design, yet the resulting theoretical convergence analysis turns out to be highly different and more effective than the existing literature.
> In particular, the affine bounded gradient assumption and the normalization parameter $\beta$ bring new challenges into the stochastic setting. To be specific, we theoretically designed new learning rate $\gamma = \min ( \frac{1}{4L_0},\frac{1}{4L_1}, \frac{1}{\sqrt{T}}, \frac{1}{8L_1(3\tau_2/\tau_1)^{\beta}} )$,  and adaptive normalization term $h_t = \max \Big(1, (4L_1\gamma)^{\frac{1}{\beta}}\Big({2||\nabla f_{\xi_{B'}}(w_t)||+\frac{\tau_2}{\tau_1} }\Big) \Big) $ to guarantee convergence with constant level batch size choices $B, B' = \Theta (\tau_1^2)$ under generalized-smoothness. For example, in the main proof of Theorem 2 (in Appendix), the inequalities (36), (37) and (40) utilizes the proposed learning rate and adaptive normalization factor $h_t$ to bound the coefficient $\gamma (L_0+L_1||\nabla F(w) ||^{\alpha})/h_t^{\beta}$ by $1/2$ . In Lemma 6,  the $\gamma^2(L_0+L_1||\nabla F(w) ||^{\alpha})\frac{\tau_2^2}{\tau_1^2}$ induced from the generalized-smooth condition is upper bounded by combining two upper bounds together derived based on different values of $|| \nabla F(w)||$. These techniques and resultant inequalities are indirect to handle without rigorously mathematical reasoning.
>
> **Q**: Why is the generalized PL condition interesting to study? I know you do give an example in lines 172-175 but how relevant is this in practice?
>
> **A**: Great Question! we have updated our related work to include more discussions on the motivation of studying generalized PL condition. To elaborate, the PL condition initially caught researchers' attention because it relaxes the strong convexity requirement but yield linear convergence rate. Recently, [4] found PL*, a variation of PL condition widely exists in loss landscape of over-parametrized neural networks. They also provide explanation why SGD-type algorithm can converge fast when training over-parametrized neural network with mean-squared loss.
> Later, [5] further propose $\psi$-KL* condition to include a broader class of loss functions such as CE loss, hinge loss and also provide convergence guarantee of SGD algorithm. Interestingly, our generalized PL condition turn out to be a sub-class of $\psi$-KL* condition if $F^*$ is non-negative. Because if $F^*\geq 0$, we can rewritten generalized PL condition as
> $$
> ||\nabla F(w) || \geq \big[(2\mu)\cdot (F(w)-F^*)\big]^{1/\rho}\geq \big[(2\mu)\cdot (F(w))\big]^{1/\rho},
> $$
> where function $f(.)=(.)^{1/\rho}$ is non-decreasing for $\rho>0$.
>
> References
>
> [1] Zhang, Jingzhao, et al. "Why gradient clipping accelerates training: A theoretical justification for adaptivity." arXiv preprint arXiv:1905.11881 (2019)
>
> [2] Chen, Ziyi, et al. "Generalized-smooth nonconvex optimization is as efficient as smooth nonconvex optimization." International Conference on Machine Learning. PMLR, 2023
>
> [3] Fang, Cong, et al. "Spider: Near-optimal non-convex optimization via stochastic path-integrated differential estimator." Advances in neural information processing systems 31 (2018).
>
> [4] Liu, Chaoyue, et al. "Loss landscapes and optimization in over-parameterized non-linear systems and neural networks." Applied and Computational Harmonic Analysis 59 (2022): 85-116.
>
> [5] Scaman, Kevin, et al. "Convergence rates of non-convex stochastic gradient descent under a generic lojasiewicz condition and local smoothness." International conference on machine learning. PMLR, 2022.

---

> ### Author Response · Authors · 2024-11-24
> **Reply to comment from 8dpS**
>
> Thanks for your reply!
> Regarding your concern about our batch size choice, which is related to $\tau_1$ originating from assumption 4, we want to explain in more detail.
>
> In stochastic optimization literature, people assume either a constant bound or affine bound like our assumption 4, mainly to upper bound expectation of the second momentum of stochastic gradient. Mathematically, it can be expressed as
> $$
> E_{\xi}[||g_t||^2]\leq \mathcal{O}(1)+\mathcal{O}(||\nabla F(w_t) ||^2),
> $$
> where $||\nabla F(w_t) ||$ is the norm of full gradient.
>
> The motivation for doing this is because, in stochastic optimization, the theoretical evaluation metric for an algorithm is in terms of $||\nabla F(w_t) ||$ instead of stochastic estimator $||g_t||$, which can vary at each trial. To eliminate the effect of randomness, assumptions like bounded/affine bounded estimation bias or bounded/affine bounded variance are necessary when conducting convergence proof in terms of $||\nabla F(w_t) ||$. The usage of these assumptions can vary depending on specific algorithm structure.
>
> Back to the derived batch size $B, B'=\Theta(\tau_1^2)$ from assumption 4, we agree there is no intuitive way to estimate $\tau_1$. It's possible that for low dimension (2d, 1d)data, exploratory data analysis might help to determine the value of $\tau_1$ accurately. However, by tuning batch sizes from 16, 32, 64, and so on, users can find the appropriate batch size. Since in standard deep learning training pipeline, data pre-processing, which includes normalization (i.e., normalized image matrix or word embeddings), can control $\tau_1$, $\tau_2$ so that it wouldn't be a huge constant. As we have demonstrated in experiments of Phase retrieval, DRO, and training ResNet(See Appendix I), $B = 128, B'=32$ is enough to guarantee convergence for most problems using I-NSGD.
>
> Previous works, such as [1], [2],[3],[4], either require the batch size of their stochastic algorithms to be $\mathcal{O}(\tau_2^2\epsilon^{-2})$ to guarantee convergence under relaxed variance assumption, or they require strong assumption such that $||g_t - \nabla F(w_t) ||\leq \tau_2$ to guarantee convergence with $\mathcal{O}(1)$ batch size [**Note**: $\epsilon$ represents the desired algorithm accuracy, which is a small constant in (0,1)]. Our work balances the assumptions and resultant batch size, which said the newly proposed I-NSGD algorithm can converge with batch size $\Theta(\tau_1^2)$ under relaxed gradient bias assumption, $||g_t - \nabla F(w_t) ||\leq \tau_1|| \nabla F(w_t)||+\tau_2$.
>
> References
>
> [1] Chen, Ziyi, et al. "Generalized-smooth nonconvex optimization is as efficient as smooth nonconvex optimization." International Conference on Machine Learning. PMLR, 2023
>
> [2] Reisizadeh A, Li H, Das S, et al. Variance-reduced clipping for non-convex optimization[J]. arXiv preprint arXiv:2303.00883, 2023.
>
> [3] Zhang, Jingzhao, et al. "Why gradient clipping accelerates training: A theoretical justification for adaptivity." arXiv preprint arXiv:1905.11881 (2019)
>
> [4]Zhang B, Jin J, Fang C, et al. Improved analysis of clipping algorithms for non-convex optimization[J]. Advances in Neural Information Processing Systems, 2020, 33: 15511-15521.

---

### Meta-Review · Area_Chair_T2pL · 2024-12-22

**Metareview:**

This paper considers the problem of nonconvex optimization under the $(L_0,L_1)$-smoothness condition (also known as generalized smoothness) and specifically considers two problems: (a) adapting to the generalized Polyak-Łojasiewicz (PŁ) condition, a generalization of strong convexity, when it exists, and (b) solving the stochastic problem, with or without the PŁ condition. For the former problem, the authors show that the normalized gradient descent method can achieve different (faster) rates depending on the PŁ condition, recovering linear convergence in the classical case ($L_1=0$ in generalized smoothness and $\rho=2$ in the generalized PŁ condition). For the stochastic problem, the authors consider a variant of normalized gradient descent where the normalization is done with a different, independently sampled point. Theorem 2 shows that this method can achieve the $1/\epsilon^4$ convergence rate (to achieve error $\epsilon$) without the need for large ($\epsilon$-dependent) batch sizes or assuming that the stochastic gradient noise norm is bounded almost surely. Finally, experiments show the proposed algorithm performs well compared to the baselines (SGD, Normalized SGD, SGD with clipping).

The paper received for 4 reviews, with scores 6, 5, 3 and 3. So, three out of four reviewers recommend rejection. I've read the reviews, the rebuttals, and the ensuing discussion. I have decided to ignore several pieces of criticism, such as the "physical units" criticism by one of the reviewers. Some other comments by the same review were valid, however. I found the replies by the authors satisfactory. Overall, I like the paper, but believe it is a bit short of what I would expect from a theoretical ICLR paper.

My own view is mostly aligned with that of the positive reviewer:
- The paper's idea is nice, it's surprising that independent normalization has not been considered in the literature before -- it's such a simple and straightforward solution to the problem of correlated samples in normalization.
- That said, it might be the case that it works worse in practice than normalized SGD (see the weaknesses section), but we know that normalized SGD may be nonconvergent in general. Independent samples have been used before in the estimation of the smoothness constant in ordinary (smooth) optimization [1], but not in generalized smoothness optimization as far as I can see.
- The analysis removes the requirement for large batch sizes or almost surely bounded gradient noise present in prior work, although it still requires very specific choices of (constant) batch sizes.
- The paper is written quite clearly, and is easy to read.

The main issues include:
- Unrealistic requirement on the batch size
- Unfair experimental evaluation
- Limited technical novelty
- Add more emphasis on the novelty of the algorithm and not necessarily of the complexity results for the problem class considered, since other equally good or better methods exist for that (e.g., incorporate your reply to the reviewer who raised these concerns to the main body of the paper)

I believe that, after a major revision (focusing on the key meaningful criticism in the reviews), the paper would be in a good enough shape to be acceptable to a top AI/ML conference. I encourage the authors to perform such as revision.

**Additional Comments On Reviewer Discussion:**

I am ignoring the criticism raised from one reviewer about physical units. While the discussion was interesting, I believe that it unnecessarily side-tracked the focus on aspects of the paper that are not central to the main contributions. Also, I do not agree that physical units analysis is a standard tool in optimization.

---

### Decision · Program_Chairs · 2025-01-22

Reject